# Exploring the role of macromolecular crowding and TNFR1 in cell volume control

**Parijat Biswas[1], Priyanka Roy[1†], Subhamoy Jana[1†], Dipanjan Ray[1], Jibitesh Das[2], Bipasa Chaudhuri[1], Ridita Ray Basunia[1], Bidisha Sinha[2], Deepak Kumar Sinha[1]\***

[1]School of Biological Sciences, Indian Association for the Cultivation of Science, Kolkata, India; [2]Department of Biological Sciences, Indian Institute of Science Education and Research Kolkata, Kolkata, India

**\*For correspondence:** emaildks@gmail.com

[†]These authors contributed equally to this work

**Abstract** The excessive cosolute densities in the intracellular fluid create a physicochemical condition called macromolecular crowding (MMC). Intracellular MMC entropically maintains the biochemical thermodynamic equilibria by favoring associative reactions while hindering transport processes. Rapid cell volume shrinkage during extracellular hypertonicity elevates the MMC and disrupts the equilibria, potentially ushering cell death. Consequently, cells actively counter the hypertonic stress through regulatory volume increase (RVI) and restore the MMC homeostasis. Here, we establish fluorescence anisotropy of EGFP as a reliable tool for studying cellular MMC and explore the spatiotemporal dynamics of MMC during cell volume instabilities under multiple conditions. Our studies reveal that the actin cytoskeleton enforces spatially varying MMC levels inside adhered cells. Within cell populations, MMC is uncorrelated with nuclear DNA content but anti-correlated with the cell spread area. Although different cell lines have statistically similar MMC distributions, their responses to extracellular hypertonicity vary. The intensity of the extracellular hypertonicity determines a cell's ability for RVI, which correlates with nuclear factor kappa beta (NFkB) activation. Pharmacological inhibition and knockdown experiments reveal that tumor necrosis factor receptor 1 (TNFR1) initiates the hypertonicity-induced NFkB signaling and RVI. At severe hypertonicities, the elevated MMC amplifies cytoplasmic microviscosity and hinders receptor interacting protein kinase 1 (RIPK1) recruitment at the TNFR1 complex, incapacitating the TNFR1-NFkB signaling and consequently, RVI. Together, our studies unveil the involvement of TNFR1-NFkB signaling in modulating RVI and demonstrate the pivotal role of MMC in determining cellular osmoadaptability.

## Editor's evaluation

This study provides a useful real time technique utilising fluorescence emission anisotropy of cytoplasmically expressed mEGFP to measure macromolecular crowding in living cells. The authors use this technique to provide solid evidence for the role of macromolecular crowding in cell volume control in mammalian cells under different conditions and perturbations. This method is likely to be of general interest to cell biologists and biophysicists since macromolecular crowding has broad implications for cell biological phenomena such as in osmotic stress response, cell cycle, cell death, and phase separation to cite only a few.

## Introduction

The intracellular fluid is an aqueous milieu of multiple macromolecule species that include proteins, nucleic acids, lipids, polysaccharides, and numerous metabolites. Making up 56% of a cell's net dry mass, proteins are the most abundant macromolecules with intracellular concentrations ranging between 50 mg/mL and 400 mg/mL (*Kohata and Miyoshi, 2020*; *Neurohr and Amon, 2020*; *Model et al., 2021*). Such number densities within the confines of the intracellular fluid space create the macromolecular crowding (MMC) effect (*Minton, 1981*; *Ellis, 2001*; *Rivas and Minton, 2016*; *Delarue et al., 2018*). Individual macromolecules that operate a particular biochemical reaction cannot access the excluded volume of their cosolutes, thus their effective concentration increases while their average mobility decreases, resulting in a higher thermodynamic activity and lower entropy (*Minton, 1983*; *Garner and Burg, 1994*; *Rivas and Minton, 2016*). Consequently, MMC affects cellular microviscosity (*Rashid et al., 2015*), active transport processes (*Nettesheim et al., 2020*), protein-ligand binding kinetics (*Minton, 2001*; *Köhn et al., 2021*), enzyme-substrate reactivity (*Thoke et al., 2018*; *Wilcox et al., 2021*), macromolecular self-assembly (*André and Spruijt, 2020*; *Schreck et al., 2020*), protein folding (*Adén and Wittung-Stafshede, 2014*), and post-translational modifications (*Darling and Uversky, 2018*). Furthermore, since abrupt changes in cell volume affect MMC and in turn, the intracellular thermodynamic landscape, a hypothesis emerged that cells may utilize such shifts in biochemical reaction kinetics to 'sense' volume changes (*Minton et al., 1992*; *Burg, 2000*; *Al-Habori, 2001*; *Hoffmann et al., 2009*). Particularly, studies in dog erythrocytes have shown that MMC is a key determinant of the resting cell volume (*Colclasure and Parker, 1991*; *Colclasure and Parker, 1992*). Destabilizing the cell volume-MMC homeostasis through extracellular osmotic imbalances can be fatal, as persistent cell shrinkage precedes apoptosis while aberrant cell swelling leads to necrosis (*Kerr, 1971*; *DiBona and Powell, 1980*; *Roger et al., 1999*; *Maeno et al., 2000*; *Yu and Choi, 2000*; *Bortner and Cidlowski, 2002*; *Berghe et al., 2010*; *Okada et al., 2020*). Accordingly, cells initiate regulatory volume increase (RVI) or decrease (RVD) to avoid the lethal consequences of the osmotically altered volume, and concomitantly, MMC (*Burg, 1995*; *Antolic et al., 2007*; *Hall, 2019*; *Govindaraj et al., 2024*). The cellular ability of RVI/RVD and their molecular mechanisms vary widely among cell lines, source tissue, and organisms (*Garner and Burg, 1994*; *Pedersen et al., 2001*; *Lambert et al., 2008*; *Hoffmann et al., 2009*; *Jentsch, 2016*). In the particular case of RVI, the transcription factor TonEBP (NFAT5) has been well studied for its osmoprotective role (*Aramburu et al., 2006*; *Brocker et al., 2012*), but another prominent transcription factor of the same Rel-family, NFkB, has been implicated but relatively unexplored (*Hasler et al., 2008*; *Roth et al., 2010*). NFkB activity is involved in multiple cell survival pathways against a wide array of stressors, including apoptosis induction (*Taniguchi and Karin, 2018*; *Verzella et al., 2020*). As failure of cellular RVI also promotes apoptosis (*Bortner and Cidlowski, 1996*; *Gómez-Angelats and Cidlowski, 2002*; *Maeno et al., 2006*), it is interesting to see whether NFkB activity has a protective role by initiating the RVI process and if the altered MMC is involved in modulating NFkB activity.

Until recently, cellular MMC levels have been indirectly quantified through bulk viscosity measurements using fluorescence photobleaching techniques, correlation spectroscopy, polarization anisotropy, and single-particle tracking (*Luby-Phelps, 1999*; *Verkman, 2002*; *Zorrilla et al., 2004*; *Kuimova et al., 2008*; *Kuimova et al., 2009*; *Liu et al., 2013b*; *Miermont et al., 2013*; *Puchkov, 2013*; *Soleimaninejad et al., 2017*; *Delarue et al., 2018*; *Neurohr and Amon, 2020*). Other studies have used specialized FRET probes to directly investigate cellular MMC (*Boersma et al., 2015*; *Murade and Shubeita, 2019*; *Pittas et al., 2021*). Since solution refractive index generally scales linearly with macromolecule concentration, protoplasmic refractive index measurements can also serve as an estimate of MMC levels (*Charrière et al., 2006*; *Yanase et al., 2010*; *Bélanger et al., 2019*; *Aknoun et al., 2021*). Notably, the effect of refractive index on the fluorescence lifetime of EGFP-like proteins is a robust technique for quantifying cellular MMC at high spatial resolutions (*Sizaire et al., 2006*; *Pliss et al., 2012*; *Pliss et al., 2019*; *James et al., 2019*; *Pliss and Prasad, 2020*). In this manuscript, we propose that measuring the steady-state fluorescence anisotropy of EGFP ($r_{EGFP}$) is a more straightforward method of quantifying cellular MMC, with the equivalent spatial resolution of fluorescence lifetime measurements but faster temporal throughput. The rationale behind using $r_{EGFP}$ as a probe for MMC is explained in the 'Materials and methods' section. We demonstrate the high dynamic range, pH insensitivity, inertness to ionic and small-molecule crowding of $r_{EGFP}$ through in vitro studies, and then track the cell volume-MMC interplay during multiple isotonic and hypertonic

conditions using $r_{EGFP}$. Additionally, we unveil TNFR1-mediated NFkB signaling as a cellular RVI initiator and show that elevated cytosolic MMC levels at severe hypertonicities hinder TNFR1 molecular assembly and the RVI process.

## Results

### Fluorescence anisotropy of EGFP is a robust probe for MMC

MMC increases solution microviscosity (*Goins et al., 2008*; *Rashid et al., 2015*) and refractive index (*Khago et al., 2018*; *Levchenko et al., 2018*; *Pliss et al., 2012*; *Pliss et al., 2019*; *Sizaire et al., 2006*; *Zhao et al., 2011*), two physical parameters also influencing fluorescence anisotropy. To test the effects of MMC on the steady-state fluorescence anisotropy of EGFP ($r_{EGFP}$) in vitro, we purified EGFP from BL21-DE3 using anion exchange chromatography. We then measured the concentration of the purified EGFP using FCS (fluorescence correlation spectroscopy) after serially diluting the EGFP solution to FCS-compatible levels. Increasing the dilution of EGFP raised the autocorrelation amplitude ($G_0$) of the fluorescence intensity fluctuation (*Figure 1—figure supplement 1A*). The number density ($N = \frac{1}{G_0 - 1}$) of EGFP molecules in the confocal volume was linearly dependent on the dilution factor, and the same linearity prevailed while measuring the total fluorescence intensity of EGFP solutions (*Figure 1—figure supplement 1A*, inset) in our $r_{EGFP}$ measurement system (*Figure 1—figure supplement 1B*). We then measured the $r_{EGFP}$ of 50 nM EGFP in pH-adjusted buffer solutions containing different crowding agents with varying molecular weights and hydrodynamic radii. Raising crowder concentrations caused a linear increase of $r_{EGFP}$ (*Figure 1A*), and this linearity qualified $r_{EGFP}$ as a potential tool to quantify and compare MMC levels. Millimolar concentrations of the proteins - BSA (bovine serum albumin) and a-lactalbumin (alpha-lactalbumin) - caused a steep rise in $r_{EGFP}$. However, other crowder species common in the cytoplasm, like polysucrose (Ficoll), small organic molecules (L-arginine and glycine), and ions (NaCl), induced visible changes in $r_{EGFP}$ only at very high, non-physiological concentrations (*Figure 1A*). Two variants of polyethylene glycol (PEG-20000 and PEG-6000), having different molar masses (20 kDa and 6 kDa), increased $r_{EGFP}$ in the millimolar range as the proteins. However, PEG has limited biological relevance as it is not intrinsically present in cells. Among all the crowders tested by us, BSA with the highest molar mass had the most prominent impact on $r_{EGFP}$, even though the hydrodynamic radii of PEG, Ficoll, and proteins like BSA are comparable (~3.48 nm) (*Ikeda and Nishinari, 2000*; *Linegar et al., 2010*; *Sim et al., 2012*). In accordance with $r_{EGFP}$, increasing the crowder concentration monotonically decreased the fluorescence lifetime of EGFP ($\tau_{EGFP}$), and the effect of protein crowding (BSA) was much more pronounced than polysucrose (Ficoll) (*Figure 1B*). The changes in $\tau_{EGFP}$ are caused by a concentration-dependent increase in refractive index ($n$) (*Figure 1—figure supplement 1C*) because $\tau_{EGFP}$ scales linearly with $1/n^2$, as predicted by the Strickler-Berg equation (*Strickler and Berg, 1962*; *Figure 1—figure supplement 1D*). Time-resolved fluorescence anisotropy (TR-FA) measurements of EGFP in different BSA concentrations further revealed the effect of MMC on the intrinsic anisotropy ($r_0$) and rotational correlation time ($\theta_C$) (*Figure 1C-i*), both of which increased with crowder concentrations (*Figure 1C-ii*). We used the Perrin equation to reconstruct the steady-state $r_{EGFP}$ with the values of $r_0$, $\theta_C$, and $\tau$ obtained from the TR-FA measurements in different crowder concentrations. The reconstructed $r_{EGFP}$ agreed with the measured steady-state $r_{EGFP}$ (*Figure 1D*), albeit with a suitable instrumental correction factor to account for the differences between wide-field and confocal systems. The scattered light in wide-field microscopes can depolarize the net fluorescence emission, reducing the magnitude of the observed change in $r_{EGFP}$ and the overall values. Thus, we concluded that an increase in MMC affects the $r_0$, $\theta_C$, and $\tau$ of EGFP, such that $r_{EGFP}$ increases linearly with crowder concentrations.

To further estimate the relative contribution of the MMC-driven increase in $\eta$ and $n$ on the measured $r_{EGFP}$, we compared the steady-state fluorescence anisotropy values of EGFP ($r_{EGFP}$) with that of fluorescein ($r_{Fluorescein}$) in glycerol solutions (*Figure 1E*). In the range of 80–90% (vol/vol) glycerol, $\eta$ increases by 264%, but $n$ changes only by 1% (*Lide, 2004*). For these solutions, the $\tau/\theta_C$ approach 1 for fluorescein (*Devauges et al., 2012*). The linear nature of the $r_{EGFP}$ curve and the exponential nature of the $r_{Fluorescein}$ curve in *Figure 1E* showed that solution $\eta$ had a negligible effect on $r_{EGFP}$. Together, the results in *Figure 1A–C and E* established that changes in $r_0$ and $\tau$ played a greater role in elevating $r_{EGFP}$ than $\theta_C$. To verify the reliability of $r_{EGFP}$ as a probe of intracellular MMC, we further explored the dependence of $r_{EGFP}$ on EGFP concentration and pH. Fluorescence resonance energy

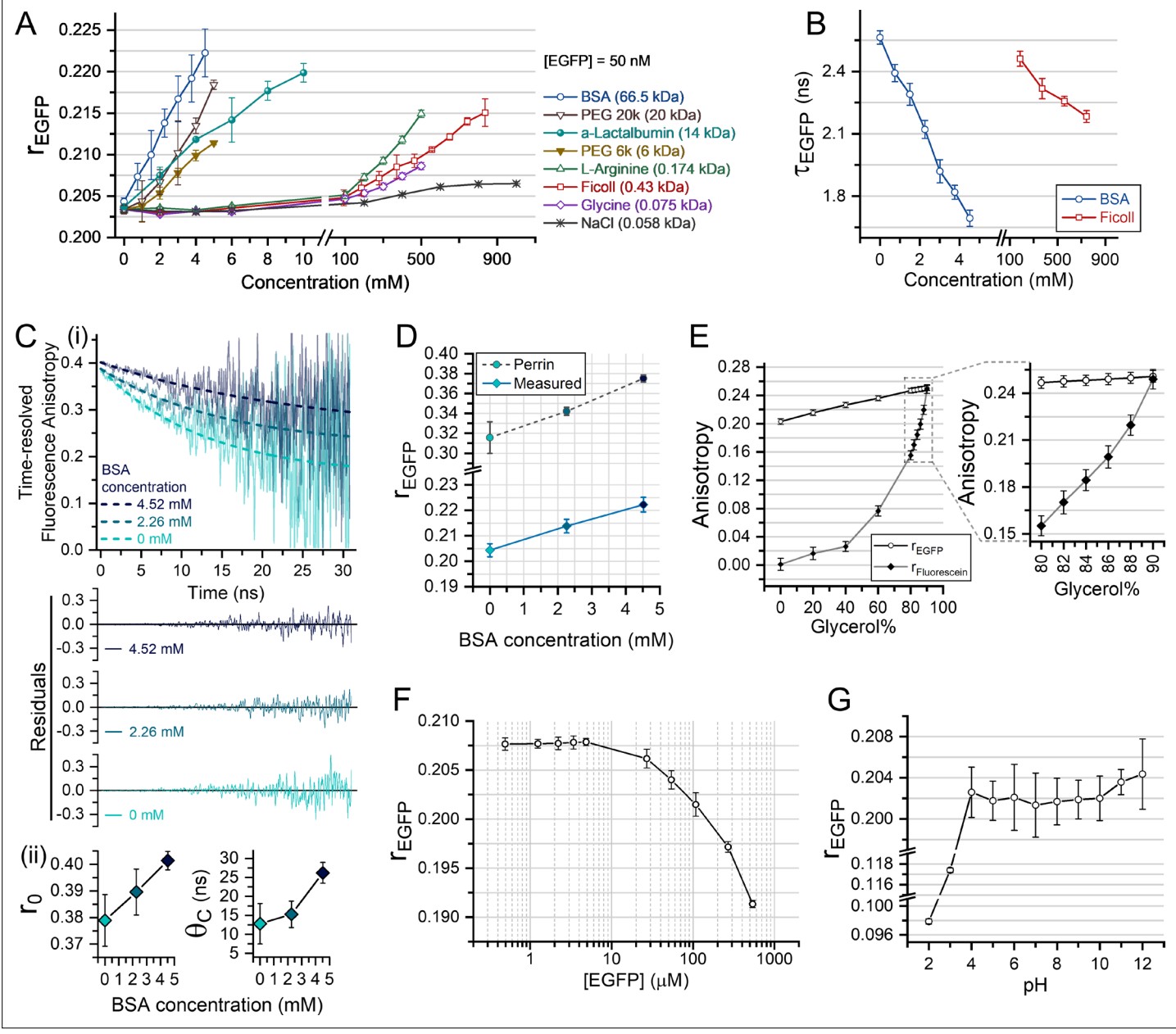

**Figure 1.** Fluorescence anisotropy of EGFP is a robust probe for macromolecular crowding. (**A**) Steady-state fluorescence anisotropy of EGFP ($r_{EGFP}$) progressively increases with crowder concentration and crowder molecular weight. (**B**) Fluorescence lifetime of EGFP ($\tau_{EGFP}$) steadily decreases with increasing crowder concentration - as shown for bovine serum albumin (BSA) (protein) and Ficoll (polysucrose). (**C-i**) Time-resolved fluorescence anisotropy of EGFP (continuous lines - representative data from one experiment) and their fit to mono-exponential decay (dashed lines) in three different BSA concentrations along with their residuals, (**C-ii**) $r_0$ (intrinsic anisotropy) and $\theta_C$ (rotational correlation time) of EGFP vs BSA concentration, as obtained from curve fitting in **C-i**. (**D**) Comparison of the reconstructed $r_{EGFP}$ (dashed line) using the Perrin equation with the $r_0$, $\tau_{EGFP}$, and $\theta_C$ values obtained from **B and C-ii**, and the measured $r_{EGFP}$ (solid line) for the same BSA concentrations. (**E**) Comparison of the steady-state fluorescence anisotropy of EGFP and fluorescein in solutions of varying glycerol content (zoomed-in glycerol content 80–90%), showing that the viscosity dependence of $r_{EGFP}$ is negligible. (**F**) $r_{EGFP}$ vs EGFP concentration in HEPES buffer (pH 7.4) reveals that at [EGFP]>10 µM, $r_{EGFP}$ enters the homo-FRET regime. (**G**) Dependence of $r_{EGFP}$ on the solution pH of HEPES buffers. All the plots show the mean values obtained from at least three individual experiments ($N≥3$) performed at 25°C, and the error bars represent the standard deviation (SD). Except (**F**), 50 nM EGFP was used for all experiments.

The online version of this article includes the following source data and figure supplement(s) for figure 1:

**Source data 1.** Data tables for *Figure 1C-i* and *Figure 1—figure supplement 1A*.

**Figure supplement 1.** Fluorescence anisotropy of EGFP is a robust probe for macromolecular crowding.

transfer between EGFP molecules (homo-FRET) could be an important artifact in $r_{EGFP}$ readouts at high EGFP concentrations. Measurements of $r_{EGFP}$ against EGFP concentrations showed that $r_{EGFP}$ is independent of [EGFP] variations at less than 10 μM concentrations, and the subsequent decrease of $r_{EGFP}$ at [EGFP]>10 μM is presumably due to homo-FRET (*Figure 1F*). Furthermore, $r_{EGFP}$ was also independent of pH at the physiological range (*Figure 1G*). Therefore, cell-to-cell variations in EGFP expression level (if [EGFP]<10 μM), cytosolic pH, ion concentrations, or the small-molecule crowder concentrations will not affect $r_{EGFP}$, making it a reliable probe of intracellular MMC. Given the enhanced sensitivity of $r_{EGFP}$ to proteins over other macromolecules, and proteins being the most abundant macromolecules in a cell, intracellular $r_{EGFP}$ values would primarily sense protein crowding.

## MMC levels do not significantly vary between individual cell lines

Next, we evaluated the reliability of $r_{EGFP}$ as a probe for intracellular MMC. We subjected NIH/3T3 fibroblasts expressing monomeric EGFP to extracellular hypertonicity (additional 600 mM mannitol in the isotonic culture media), such that the consequential water efflux increases the intracellular MMC (*Dmitrieva and Burg, 2005*). TR-FA measurements during isotonic conditions and 2 min after hypertonicity exposure showed that the elevated MMC decreased $\tau$ and increased $r_0$, $\theta_C$, and $r_{EGFP}$ (*Figure 2A-ii–iv*), analogous to our in vitro studies. We further reconstructed the $r_{EGFP}$ map from the $\tau$, $r_0$, and $\theta_C$ maps using the Perrin equation (*Figure 2A-v*) and found that the differences between the measured $r_{EGFP}$ (*Figure 2A-vi*) and the reconstructed $r_{EGFP}$ values were negligible (*Figure 2A-vii*). Therefore, the hypertonicity-induced changes in intracellular $r_{EGFP}$ could be ascribed to the Perrin equation. The intracellular $r_{EGFP}$ maps from the confocal TR-FA system contained spatial variability, and the variability was more prominent in the $r_{EGFP}$ maps obtained on a wide-field microscope (*Figure 2B*). Thus, to compare the MMC of different cells, we needed to assign a single $r_{EGFP}$ metric to each cell that represented its characteristic MMC. Judging from the intracellular $r_{EGFP}$ distributions in the representative examples with extremely dissimilar morphologies (*Figure 2B*), the modal $r_{EGFP}$ value corresponded to the predominant MMC condition in the cell while the mean $r_{EGFP}$ value included the influence of the spatial variability. Assuming that the hypertonicity-driven MMC increase should be equivalent across different cells irrespective of their morphology, we compared the spatial distributions of intracellular MMC during extracellular isotonicity and hypertonicity (+600 mM mannitol). In the two extreme examples of morphological dissimilarity (*Figure 2B*), the difference between the isotonic and hypertonic modal $r_{EGFP}$ was significantly lesser than the difference in the mean $r_{EGFP}$. The spread of the modal $r_{EGFP}$ distribution was also less than that of the mean $r_{EGFP}$ distribution for NIH/3T3 cells (*Figure 2—figure supplement 1A-i*), and the differences in $r_{EGFP}$ between isotonic and 10 min post hypertonicity induction were also more uniform for the modal values (*Figure 2—figure supplement 1A-ii*). Thus, the modal $r_{EGFP}$ values could be used as a robust metric for cell-to-cell MMC comparisons.

We then compared the cell-to-cell variations of the characteristic MMC of NIH/3T3 fibroblasts along with Hoechst co-staining to explore if the modal $r_{EGFP}$ per cell is correlated with the nuclear DNA content, which varies during the cell cycle phases of G1, S, or G2 (*Figure 2C*, *Figure 2—figure supplement 1B*). NIH/3T3 cells showed a broad distribution of $r_{EGFP}$ without any explicit correlation with DNA content, implying that the heterogeneity of intracellular MMC in the NIH/3T3 population is independent of the cell cycle stage during interphase. We then compared the modal $r_{EGFP}$ of four different cell lines - NIH/3T3 (fibroblasts), HeLa (epithelial cells from cervical tumor), MDA-MB-231 (mesenchymal subtype of triple negative breast cancer cells), and RAW 264.7 (macrophages) (*Figure 2D*). Although the characteristic MMC of NIH/3T3, HeLa, and MDA-MB-231 was statistically similar, RAW 264.7 macrophages had a higher MMC at the cell population level (*Figure 2E*). Given the substantial variability observed in the intracellular modal $r_{EGFP}$ values within a particular cell line, we questioned whether this heterogeneity might arise from genuine variations in the intracellular MMC or fluctuations in homo-FRET. In the homo-FRET regime (*Figure 1F*), $r_{EGFP}$ and the intracellular [EGFP] should be negatively correlated. Photobleaching is a well-established methodology for quantitatively assessing homo-FRET (*Ghosh, 2012*). To ascertain whether the intracellular [EGFP] distribution conforms to the homo-FRET regime, we conducted photobleaching experiments on NIH/3T3 cells expressing monomeric EGFP. As a positive control, we subjected NIH/3T3 cells expressing dimeric EGFP (2GFP) to similar degrees of photobleaching (*Figure 2—figure supplement 1C-i, ii*). The 2GFP molecules exhibited homo-FRET irrespective of their cellular expression levels due to the inherent

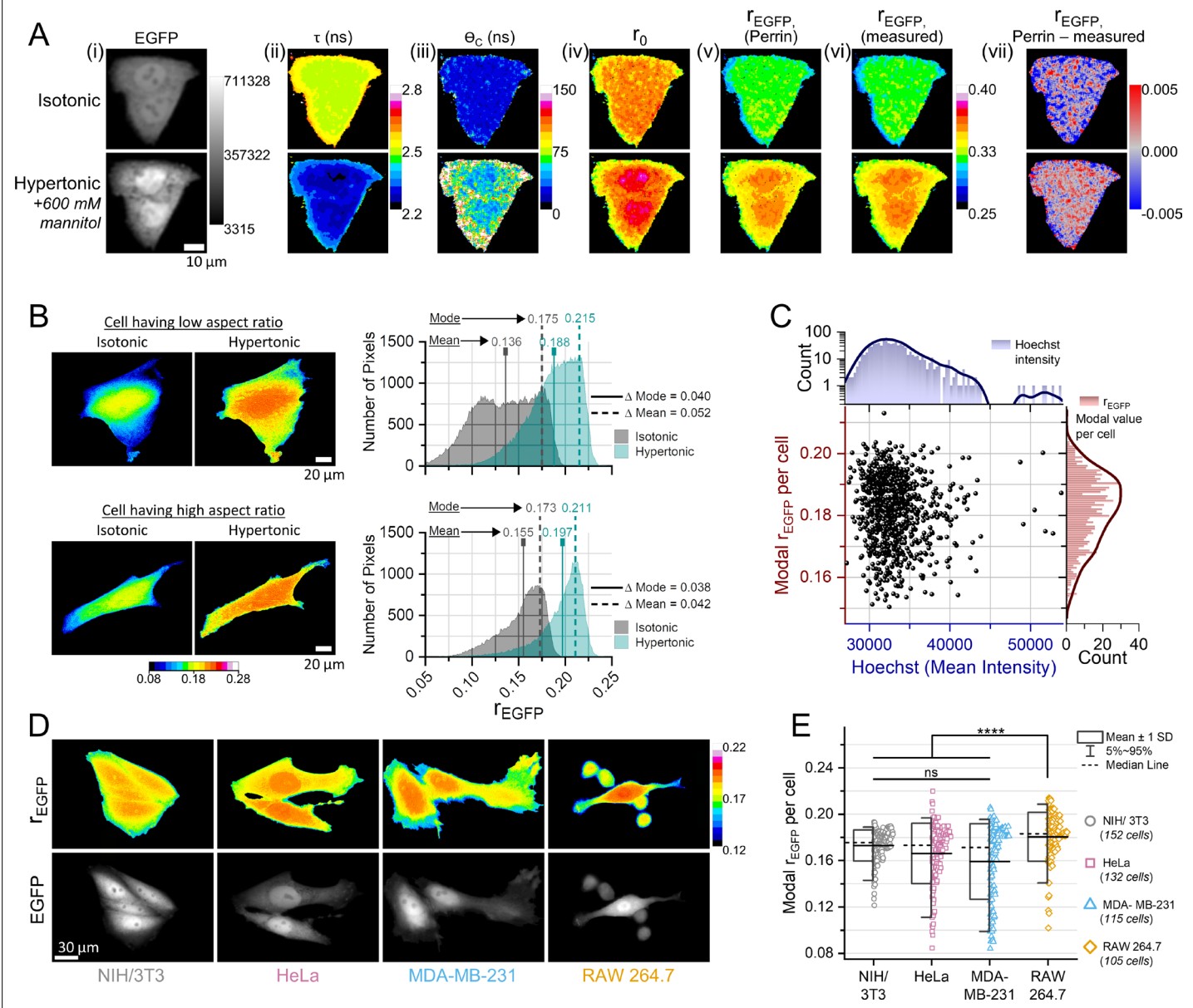

**Figure 2.** Macromolecular crowding (MMC) levels do not significantly vary between individual cell lines. (**A**) Time-resolved fluorescence micrographs of NIH/3T3 fibroblasts expressing EGFP in isotonic (top panel) and hypertonic (bottom panel) conditions. Representative images of EGFP's total intensity in (**i**), fluorescence lifetime ($\tau_{EGFP}$) in (**ii**), rotational correlation time ($\theta_C$) in (**iii**), intrinsic anisotropy ($r_0$) in (**iv**), steady-state anisotropy ($r_{EGFP}$) calculated using the Perrin equation in (**v**) with values from (i, ii, iii), measured steady-state anisotropy ($r_{EGFP}$) in (**vi**), and the difference between the anisotropy values obtained from the Perrin equation and direct measurements in (**vii**). Accompanying calibration bars indicate the colors representing the depicted quantities. (**B**) The $r_{EGFP}$ maps of two extreme examples of NIH/3T3-EGFP cells having dissimilar morphologies (aspect ratios) and the intracellular distribution of $r_{EGFP}$ values during isotonic and hypertonic conditions (+600 mM mannitol), highlighting the consistency of the modal value of $r_{EGFP}$ per cell in depicting MMC changes at different experimental conditions compared to the mean value of $r_{EGFP}$ per cell. (**C**) Cell-to-cell variability of MMC among NIH/3T3 fibroblasts (*n*=828 cells, *N*=3) imaged by a ×10 objective. The accompanying distributions depict kernel-smoothed histograms (modal $r_{EGFP}$, dark red) and DNA content (Hoechst intensity, dark blue). (**D**) $r_{EGFP}$ and total intensity maps of representative cells from different cell lines. (**E**) The modal $r_{EGFP}$ value per cell from different cell lines show that only RAW 264.7 cells have a statistically different distribution of cellular MMC. The boxes represent the distribution mean ± 1 SD, and the whiskers represent 5–95 percentiles. Number of biological replicates (cells) are provided alongside for at least four independent experiments for each cell line. Statistical analysis was performed using the non-parametric Kruskal-Wallis ANOVA after Bonferroni alpha-correction, followed by Mann-Whitney test for every group pair. **** indicates p<0.000025.

The online version of this article includes the following source data and figure supplement(s) for figure 2:

**Source data 1.** Data tables for *Figure 2B, C, and E* and *Figure 2—figure supplement 1A, C, and D*.

**Figure supplement 1.** Macromolecular crowding (MMC) levels do not significantly vary between individual cell lines.

proximity of the two GFP molecules. The $r_{EGFP}$ values observed in cells expressing monomeric EGFP did not exhibit a significant rise upon photobleaching, whereas cells expressing 2GFP displayed an approximate 8% increase in $r_{EGFP}$ when subjected to an equivalent ~30% photobleaching (*Figure 2—figure supplement 1C-iii*). Thus, we inferred that a significant majority of our experimental cell population did not belong in the homo-FRET regime. Consequently, the variability in modal $r_{EGFP}$ among different cells reflected genuine variability in the intracellular MMC. We further used FCS to measure the intracellular [EGFP] in NIH/3T3 fibroblasts. FCS measurements require low fluorophore concentrations, so the cells were photobleached until the fluorescence count rate decreased to suitable levels. The intracellular [EGFP] in the representative photobleached cell (*Figure 2—figure supplement 1D*) was estimated to be ~1.7 μM and scaling up the concentration according to the ratio of cellular EGFP intensities pre and post bleaching implied that the cell had ~8 μM EGFP before photobleaching. Comparing the average cellular EGFP intensity values of the same cells in the $r_{EGFP}$ measurement setup and the FCS measurement setup, we found that the intracellular [EGFP] in the total NIH/3T3 cell population varied between 3 μM and 18 μM (*Figure 2—figure supplement 1D*, inset). Monomeric EGFP exhibits homo-FRET at concentrations greater than 10 μM (*Figure 1F*), and in the experimental cell population, only ~12% of cells had [EGFP]>10 μM. Therefore, for studying intracellular MMC using $r_{EGFP}$, we selected cells whose fluorescence intensities corresponded to [EGFP]<10 μM. However, a potential caveat may arise while measuring $r_{EGFP}$ in cells under severe hypertonic conditions, where local EGFP concentrations might crossover to the homo-FRET regime. Hence, we photobleached randomly selected NIH/3T3-mEGFP cells at 10 min after inducing 600 mM hypertonicity (*Figure 2—figure supplement 1E*). We did not find a noticeable increase of modal $r_{EGFP}$ after photobleaching, and thus, concluded that 600 mM hypertonicity was not sufficient to induce homo-FRET in NIH/3T3 cells.

## The actin cytoskeleton enforces spatially varying MMC levels

The representative $r_{EGFP}$ maps in *Figure 2B and D* showed that cellular MMC is non-uniform at a few microns' length scales. Time-lapse videos of $r_{EGFP}$ in cells that generate new lamellipodial extensions further showed that the MMC in the lamellar cytoplasm was lower than the rest of the cell body (*Video 1*), which agreed with previous microviscosity measurements in the lamellar and near-lamellipodial regions (*Laurent et al., 2005*). Cellular lamellipodial dynamics are primarily regulated by actomyosin activity (*Ridley, 2011*; *Tojkander et al., 2012*), so we investigated if the actin cytoskeleton had a role in generating spatially heterogeneous intracellular MMC. Simultaneous imaging of actin filaments and $r_{EGFP}$ in NIH/3T3 fibroblasts revealed that regions of lower MMC within the lamellar areas were demarcated from the perinuclear areas by filamentous actin structures (*Figure 3A*). The different MMC levels in the lamellar and perinuclear regions should manifest in the microviscosity of the cytoplasm. Hence, we compared the translational mobility of EGFP in the two regions using FRAP (fluorescence recovery after photobleaching) (*Figure 3B-i*). However, the diffusion coefficient of EGFP did not vary appreciably between the two regions, possibly because the decelerating effect of the local MMC on the translational mobility of EGFP is not sufficient to resolve the meso-scale microviscosity (*Fabry et al., 2001*; *Goins et al., 2008*; *Wong et al., 2004*). Identical to the cytoplasm, using FRAP to resolve the differential microviscosity of different BSA concentrations was also unachievable, confirming our assumption (*Figure 3—figure supplement 1A-i*). Therefore, we performed single-particle tracking of fluorescent microspheres having 200 nm diameter, which are significantly larger than the local intracellular crowding agents. The mean-squared displacement (MSD) curves of the microspheres pronouncedly shifted with increasing BSA concentrations (*Figure 3—figure supplement 1A-ii*), indicating that single-particle tracking is better at resolving crowding-mediated microviscosity than FRAP. At the timescale of 1 s, the average diffusion rates of the 200 nm microspheres amounted to ~0.29, 0.14, and 0.016 μm²/s at the BSA concentrations of 0, 2.26, and 4.52 mM, respectively. Similarly, the MSD of the same microspheres in the lamellar regions of NIH/3T3 fibroblasts showed higher diffusivity

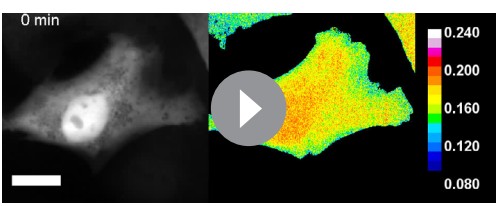

**Video 1.** 8 hr time-lapse (5 frames per second) of an NIH/3T3-EGFP, showing EGFP intensity on the left and $r_{EGFP}$ on the right. Scale bar 15 μm.

https://elifesciences.org/articles/92719/figures#video1

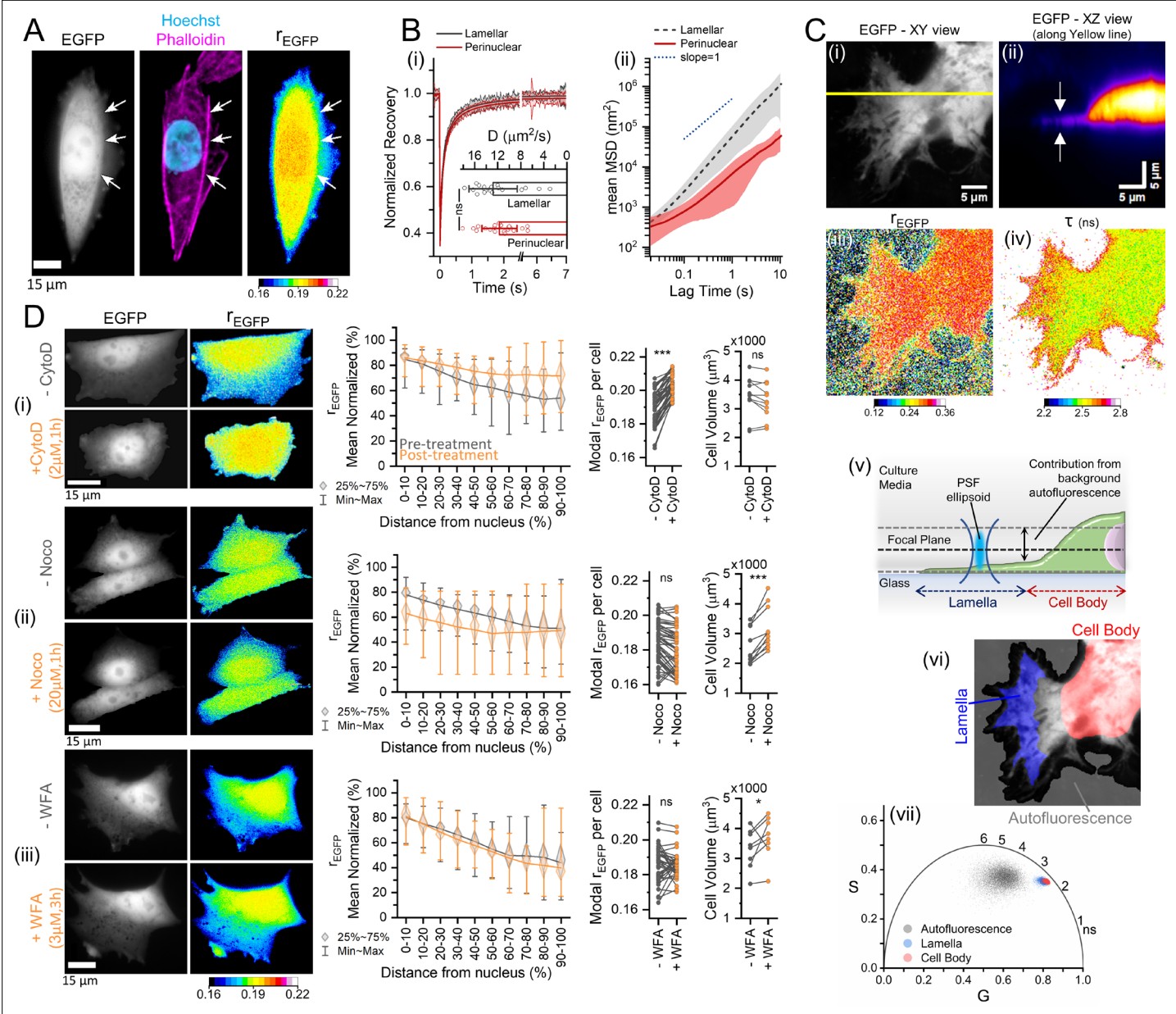

**Figure 3.** The actin cytoskeleton enforces spatially varying macromolecular crowding (MMC) levels. (**A**) EGFP intensity, Hoechst-stained DNA (in cyan), and phalloidin Alexa Fluor 546-stained actin (in magenta), and $r_{EGFP}$ map of an NIH/3T3 cell shows that the spatial heterogeneity of intracellular MMC is demarcated by actin stress fibers (arrows). (**B-i**) Average fluorescence recovery after photobleaching (FRAP) recovery curves and diffusion rates of EGFP in the lamellar (in black) and perinuclear (in red) regions of NIH/3T3 cells ($n$=21 cells; $N$=2), with the error bars representing standard deviation. Statistical significance was evaluated by unpaired t-test. (**B-ii**) The average MSD (mean-squared displacement) and their standard deviations obtained from tracking 200 nm fluorescent beads in the lamellar (dashed line in black) and perinuclear (solid line in red) regions of NIH/3T3 cells ($n$=17 cells; $N$=2). (**C**) The total intensity map of an EGFP expressing HeLa cell's lamellar region, viewed laterally (*XY*) in panel (**i**), or its cross-section (*XZ*) in panel (**ii**) along the yellow line in (**i**), with the arrows indicating the thin lamella. Panels (**iii**) and (**iv**) show the corresponding $r_{EGFP}$ and fluorescence lifetime ($\tau$) maps of panel (**i**). Panel (**v**) shows the graphical explanation of the influence of cell height on $r_{EGFP}$ values. Panel (**vi**) shows the different regions used to calculate the contributions of autofluorescence (gray), lamellar regions (blue), and the cell body (red) to the $\tau$ map in the phasor plot of panel (**vii**). In the phasor plot, the pixels corresponding to the $\tau$ of the thin lamellar region (blue dots) are slightly shifted above the $\tau$ of the cell body (red dots), revealing that $\tau$ in the lamellar region is slightly greater, and thus MMC is slightly lower than the cell body. (**D**) Representative images of NIH/3T3-EGFP, quantifications of the spatial heterogeneity of cytoplasmic $r_{EGFP}$, modal $r_{EGFP}$ ($n$=49, 51, 40 cells; $N$=2), and cell volume ($n$=11, 10, 8 cells; $N$=3) for individual cells. Black and orange colors represent pre- and post-treatment with (**i**) cytochalasin D (2 μM, 1 hr), (**ii**) nocodazole (20 μM, 1 hr), and (**iii**) withaferin A (3 μM, 3 hr). Statistical analysis performed by paired sample t-test. **** indicates p<0.0001, *** indicates p<0.001, * indicates p<0.05.

The online version of this article includes the following source data and figure supplement(s) for figure 3:

*Figure 3 continued on next page*

*Figure 3 continued*

**Source data 1.** Data tables for *Figure 3B, C-vi, and D* and *Figure 3—figure supplement 1A and E*.

**Figure supplement 1.** The actin cytoskeleton enforces spatially varying macromolecular crowding (MMC) levels.

(~0.056 μm$^2$/s at 1 s) compared to the perinuclear cytoplasm (~0.007 μm$^2$/s at 1 s) (*Figure 3B-ii*), which also agreed with previously reported observations (*Tseng et al., 2002*). Thus, we concluded that MMC levels in an individual cell are spatially heterogeneous with the lamellar regions being less crowded than the perinuclear regions. There was a small extent of super-diffusive motion of the microspheres in the lamellar regions (logarithmic MSD slope ≅ 1.29) compared to the perinuclear regions (logarithmic MSD slope ≅ 0.92), which is presumably due to the actin retrograde flows characteristic to the lamellar regions (*Anderson et al., 2008*).

However, it was still possible that the observed intracellular spatial variations in the $r_{EGFP}$ were a consequence of the imaging artifacts associated with wide-field epifluorescence microscopy. The thickness of the lamellar regions often falls below the vertical resolution limit of optical microscopy (*Atilgan et al., 2005*). Consequently, it is plausible that the $r_{EGFP}$ values within the lamellar regions were susceptible to the excitation geometry (due to a more significant focus uncertainty) and the background autofluorescence, potentially leading to an overestimation of the reduction in MMC levels within the lamellar regions. To rule out focus uncertainties, we measured the $r_{EGFP}$ and $\tau$ with confocal TR-FA with HeLa cells kept in low-autofluorescence serum-free media (*Figure 3C*). Measurements of $\tau$ are free of focus uncertainties, and the serum-free media reduces the contribution of background autofluorescence. HeLa cells showed prominent lamellar structures, and the representative cell's vertical cross-section showed that the lamellar region's thickness is in the submicron range (*Figure 3C-i and ii*, white arrows). The $r_{EGFP}$ values were marginally lower in the thinner sections (*Figure 3C-iii*), and the $\tau$ values were noticeably higher (*Figure 3C-iv*), confirming the spatial heterogeneity of cellular MMC. The contribution of the background autofluorescence is significantly higher in wide-field microscopes because of the large PSF (point spread function) (*Laasmaa et al., 2011*), causing an underestimation of the lamellar $r_{EGFP}$ (*Figure 3C-v*). We further confirmed the differential

$\tau$ values using phasor analysis, which graphically projects the chemical species having different fluorescence lifetimes in the phase space without the artifacts arising from fitting fluorescence decay curves (*Figure 3C-vi and vii*). The lamellar regions (marked blue) had slightly longer $\tau$ values than the cell body (marked red), while the autofluorescence (marked gray) had longer and noisier fluorescence lifetimes (*Figure 3C-vi*), further verifying the reduced MMC in the lamellar regions. The autofluorescence imaging artifact can be reduced by employing two different strategies: (i) using a confocal system with a narrow pinhole (*Figure 3—figure supplement 1B*) for better Z-resolution, and (ii) using imaging media without serum to reduce the autofluorescence (*Figure 3—figure supplement 1C*).

To investigate if F-actin structures genuinely barricade areas of spatially varying MMC, we induced actin depolymerization with cytochalasin D treatment in NIH/3T3 cells and estimated the spatial heterogeneity of the intracellular MMC. The spatial heterogeneity estimation was performed by creating sectorized geodesic distance maps (GDMs) between the cell and nucleus boundaries (*Figure 3—figure supplement 1D*), and then comparing the means of normalized $r_{EGFP}$ values in the different distance sectors. Comparing the

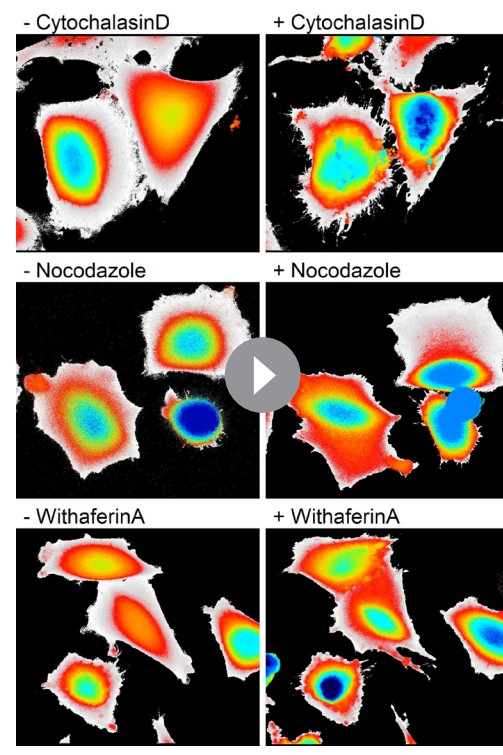

**Video 2.** 3D projections of NIH/3T3 cells after cytoskeletal depolymerization.

https://elifesciences.org/articles/92719/figures#video2

representative images and normalized $r_{EGFP}$ quantifications, the $r_{EGFP}$ in the 60–100% distance range is closer to the 0–20% distance range after actin depolymerization (*Figure 3D-i*). Surprisingly, depolymerization of microtubules or the intermediate filament vimentin led to an insignificant change in the spatial distribution of cytoplasmic MMC (*Figure 3D-ii and iii*), implying that the actin cytoskeleton segregates the intracellular regions of varying MMC. Actin disassembly also led to a significant rise of the intracellular MMC compared to the disassembly of microtubules and vimentin (*Figure 3D-i–iii*, associated graphs). Interestingly, the cell volumes pre- and post-actin disassembly were similar, but microtubule or vimentin disassembly led to a significant increase in cell volume (*Figure 3D-i–iii*, associated graphs). The probable cause for the elevation of cellular MMC upon F-actin disassembly is due to the generation of a significantly larger number of actin monomers in the constant cell volume as compared to microtubules and intermediate filaments (*Liebermeister et al., 2014*; *Pegoraro et al., 2017*; *Loiodice et al., 2019*). As cytoskeletal depolymerization also severely altered the cell morphologies, we suspected that the spatial heterogeneity estimations could be artifactual due to changes in the local cell height profile, as local cell height variations could change the autofluorescence contribution and affect the resultant $r_{EGFP}$. So, we created cell height maps from the 3D scans of the NIH/3T3 cells used for volume measurements (*Figure 3—figure supplement 1E*, *Video 2*). Comparing the relative heights between the cell and nucleus boundaries, actin depolymerization showed a higher local cell height increase in the 40–60% distance sectors compared to microtubule or vimentin depolymerization (*Figure 3—figure supplement 1E-i–iii*). Connecting the cell volume information (*Figure 3D*) and the cell height map images (*Figure 3—figure supplement 1E*), cell swelling upon microtubule and vimentin depolymerization increased the cell height uniformly while maintaining the height profile. Conversely, actin depolymerization altered the height profile while the cell volume was unchanged. As all three cases increase the local cell height, the contribution of background autofluorescence upon cytoskeletal depolymerization should be consistent. Thus, we concluded that the actin cytoskeleton genuinely enforces a spatially varying intracellular MMC.

## The characteristic cellular MMC is linked to cell spreading and adhesion

Cell spreading on fibronectin is driven by actin polymerization and actomyosin activity (*Choi et al., 2008*; *Fardin et al., 2010*; *Nisenholz et al., 2016*; *Reinhart-King et al., 2005*; *Wakatsuki et al., 2003*). As the polymerized state of actin is crucial for maintaining cellular MMC, we measured $r_{EGFP}$ of cells spreading on fibronectin-coated glass to investigate the cellular MMC during stages of increased actin assembly. After seeding, NIH/3T3 fibroblasts were allowed to settle for 15 min, and then we measured the intracellular MMC for 2 hr during dynamic cell spreading (*Figure 4A-i and ii*). MMC decreased gradually with increasing cell spreading area, and the observed decrease in the intracellular MMC was accompanied by increasing cell volume (*Figure 4A-iii*). Thus, we hypothesized that the physiological MMC setpoint might be linked to the spreading area for NIH/3T3 fibroblasts. Within a population of NIH/3T3 seeded on fibronectin-coated glass, the well-spread cells had a lower MMC than the rounded, less-spread cells (Pearson's correlation coefficient, $r=-0.42$) (*Figure 4B*). To investigate differential MMC between well-spread and rounded cells with FRAP, we simultaneously seeded NIH/3T3 cells on fibronectin or PEG-400-coated glass for 2 hr. The hydrophobic PEG coating arrested cell spreading but maintained a stable cell attachment to facilitate FRAP. The translational diffusion rate of EGFP was substantially lower in the spreading-arrested cells on PEG in comparison to the well-spread cells on fibronectin (*Figure 4C-i*), demonstrating that the extent of increased microviscosity in spreading-arrested cells is high enough to be detectable by FRAP. The elevated microviscosity expectedly correlated with the $r_{EGFP}$ measurements (*Figure 4C-ii*). The spreading-arrested cells did not show the spatial variability of MMC akin to well-spread cells (*Figure 4—figure supplement 1A*), so to track the loss of spatial variability, we induced cell rounding by trypsin treatment. Trypsin disrupts integrin-fibronectin bonds, causing cells to detach from the adhesion substrate. Loss of cell adhesion abolished the spatial variability of $r_{EGFP}$ and increased the intracellular MMC (*Figure 4D-i*). Surprisingly, the $r_{EGFP}$ levels of trypsinized cells and actin-depolymerized cells were comparable, and depolymerizing actin before trypsinization caused a non-significant change in MMC compared to trypsinization alone (*Figure 4D-ii*, *Figure 4—figure supplement 1B*). Thus, cellular F-actin levels are crucial in maintaining the MMC setpoint. Cell detachment by trypsinization induced rapid depolymerization of both actin and microtubules (*Figure 4—figure supplement 1C*), which has also been shown previously (*Celik et al., 2013*). Thus, we concluded that the increased pool of monomeric cytoskeletal

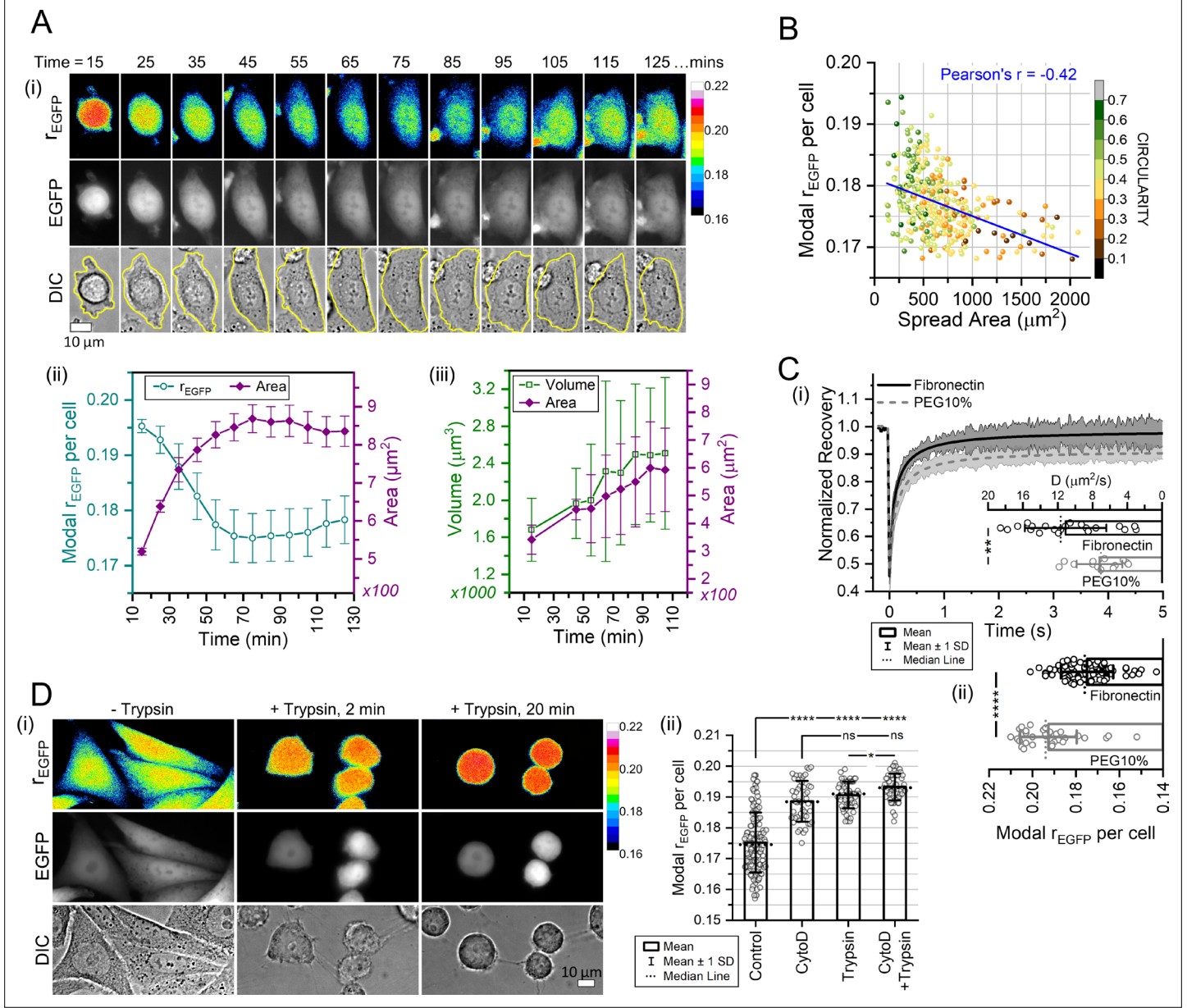

**Figure 4.** The characteristic cellular macromolecular crowding (MMC) is linked to cell spreading and adhesion. (**A-i**) $r_{EGFP}$ maps (top row), EGFP total intensity maps (middle row), and DIC images (bottom row) of NIH/3T3-EGFP during spreading on fibronectin-coated glass. (**A-ii**) Modal $r_{EGFP}$ values (cyan) and spread area (magenta) of NIH/3T3-EGFP averaged over $n=109$ cells, $N=4$. Error bars represent standard deviation. (**A-iii**) Average cell volume (green, open squares) and spread area (magenta, filled diamonds) of NIH/3T3 cells after seeding on fibronectin-coated glass ($n=11$ cells; $N=4$). Error bars show SD. (**B**) Modal $r_{EGFP}$ of NIH/3T3-EGFP cells vs their morphological spread area on fibronectin-coated glass, with the blue line indicating the negative linear correlation and the associated color bar denoting the shape circularity ($4\pi Area/Perimeter^2$) ($n=201$ cells; $N=3$). (**Ci**) Fluorescence recovery after photobleaching (FRAP) analysis of EGFP in NIH/3T3 cells seeded on fibronectin (50 µg/mL) or 10% polyethylene glycol (PEG)-400-coated glass for 2 hr. The average recovery curves and diffusion rates of EGFP are shown with the error bars representing the SD ($n=21$, 14 cells; $N=2$). Statistical significance was evaluated by unpaired t-test. ** indicates p<0.01. (**C-ii**) Modal $r_{EGFP}$ values of NIH/3T3-EGFP cells seeded on fibronectin or 10% PEG for 2 hr ($n=87$, 35 cells; $N=2$). Statistical analysis was performed using Mann-Whitney test. **** indicates p<0.0001. (**D-i**) $r_{EGFP}$ maps (top row), EGFP total intensity maps (middle row), and DIC images (bottom row) of NIH/3T3-EGFP undergoing substrate detachment due to trypsinization. (**D-ii**) Comparison of the modal $r_{EGFP}$ for untreated controls, cytochalasin D (2 µM, 1 hr) treated, trypsinized (20 min), and cytochalasin D pre-treatment (2 µM, 1 hr) then trypsinized (20 min) in NIH/3T3 ($n=131$, 49, 57, 57 cells; $N=3$). Statistical analysis was performed using Mann-Whitney test for every group pair.

The online version of this article includes the following source data and figure supplement(s) for figure 4:

**Source data 1.** Data tables for *Figure 4B, C-ii, and D-ii*.

**Figure supplement 1.** The characteristic cellular macromolecular crowding (MMC) is linked to cell spreading and adhesion.

proteins generated during cell detachment increases the MMC, and the characteristic MMC of a cell is linked to its spreading state.

## Proteostasis disruption alters cellular MMC setpoint

We next explored the cell volume-MMC interplay during hypertonic stress in different cell lines by tracking the percentage change in the modal $r_{EGFP}$ of individual cells. Upon independently measuring cell volume and $r_{EGFP}$ during moderate hypertonic stress (150 mM mannitol) in NIH/3T3 and HeLa cells (*Figure 5A*), we found that both cell lines showed RVI post 10 min of hypertonicity induction, and the cellular MMC levels scaled with the RVI. The MMC and volume recovery of HeLa was slower than NIH/3T3 cells, and HeLa showed a larger change in MMC upon hypertonicity induction even though the average volume shrinkage was similar for both cell lines. Despite partial volume recovery, the MMC recovery of HeLa cells was almost complete in 60 min, probably due to other osmoadaptive mechanisms that change the total intracellular crowder numbers or excluded volume (*Brocker et al., 2012*). The average volume and MMC of HeLa cells also did not recover to its initial state, implying partial RVI. Our observation of the partial RVI in HeLa cells aligned with a previous report (*Tivey et al., 1985*). To verify whether the gradual decrease of cellular MMC post 10 min is due to RVI, we pre-treated NIH/3T3 and HeLa cells with flufenamic acid and subjected them to 150 mM hypertonic shock (*Figure 5B*). Flufenamic acid blocks RVI by inhibiting HICCs (hypertonicity-induced cation channels) (*Numata et al., 2007*; *Wehner et al., 2003*). Pre-treatment with 700 µM flufenamic acid failed to stop the early RVI (0–30 min), but its effects were visible after 30 min, consistent with the previous reports (*Liu et al., 2013a*). The failure of flufenamic acid to stop the early RVI implies the activity of other ion channels during the initial stages of hypertonicity induction (*Jentsch, 2016*; *Okada et al., 2020*). Thus, we were convinced that the gradual decrease of MMC at the later stages of hypertonicity induction is RVI-mediated.

We then subjected NIH/3T3, HeLa, MDA-MB-231, and RAW 264.7 cells to the excess osmolarities of 50 mM mannitol (low hypertonicity) (*Figure 5C-i*) and 150 mM mannitol (moderate hypertonicity) (*Figure 5C-ii*). The intracellular MMC rose rapidly within 5 min of exposure to hypertonic media and NIH/3T3 fibroblasts had the fastest recovery. Contrariwise, HeLa cells showed partial recovery, MDA-MB-231 cells recovered in 50 mM hypertonicity but not in 150 mM hypertonicity, and RAW 264.7 macrophages failed to recover in any degree of hypertonicity within 30 min. Interestingly, for different cell lines, the response to hypertonicity ($\Delta r_{EGFP}\%$ at 10 min) scaled differently with the applied dose of hypertonicity (*Figure 5C*). The $\Delta r_{EGFP}\%$ response of MDA-MB-231 was greater than that of HeLa at 50 mM hypertonicity but smaller at 150 mM hypertonicity. The $\Delta r_{EGFP}\%$ response was higher in 150 mM hypertonicity for each cell line, suggesting a dose-based response to hypertonicity. Further exposing NIH/3T3-EGFP to varying degrees of osmotic imbalance caused a gradually larger increase in $r_{EGFP}$ (*Figure 5D*). NIH/3T3 could restore the MMC rise for ≤150 mM hypertonicity (low-to-moderate levels) within 30 min but failed when the hypertonicity exceeded 200 mM (severe hypertonicities). NIH/3T3 cells failed to recover their MMC for at least 2 hr for 600 mM hypertonicity (data not shown), and the same was true for HeLa cells (for at least 1 hr) (*Figure 5—figure supplement 1C*). We also observed the dilution of MMC when NIH/3T3 cells were exposed to 50% hypotonicity, and the MMC rose briefly at 5 min but plunged until 20 min, and then gradually rose to near-isotonic levels after 2 hr (*Figure 5—figure supplement 1A and B*). The average area of the cell outlines expanded at the onset of hypotonicity and scaled in accordance with the $r_{EGFP}$ values. The cell outlines also showed considerable shrinkage after 2 hr in hypotonic media when the $r_{EGFP}$ values approached near-isotonic levels (*Figure 5—figure supplement 1B*). The response of NIH/3T3 fibroblasts to different degrees of hypertonicity mediated by dextrose (*Figure 5—figure supplement 1D*) was comparable to mannitol (*Figure 5D*). However, for the equivalent osmolarities of 100 mOsm and 600 mOsm, NaCl-mediated hypertonicity induced a lesser $\Delta r_{EGFP}\%$ response when compared to mannitol or dextrose (*Figure 5—figure supplement 1E*). Comparing dextrose, mannitol, and NaCl-mediated hypertonicities, the MMC recovery was faster in the case of 100 mOsm NaCl. Surprisingly, even though 600 mOsm NaCl induced a smaller $\Delta r_{EGFP}\%$, cells did not recover their MMC just like 600 mOsm mannitol/dextrose. The lesser rise in NaCl-mediated hypertonicity and the faster RVI could be attributed to the differences in cellular ion fluxes due to the excess chloride ions in the culture media (*Yurinskaya and Vereninov, 2021*).

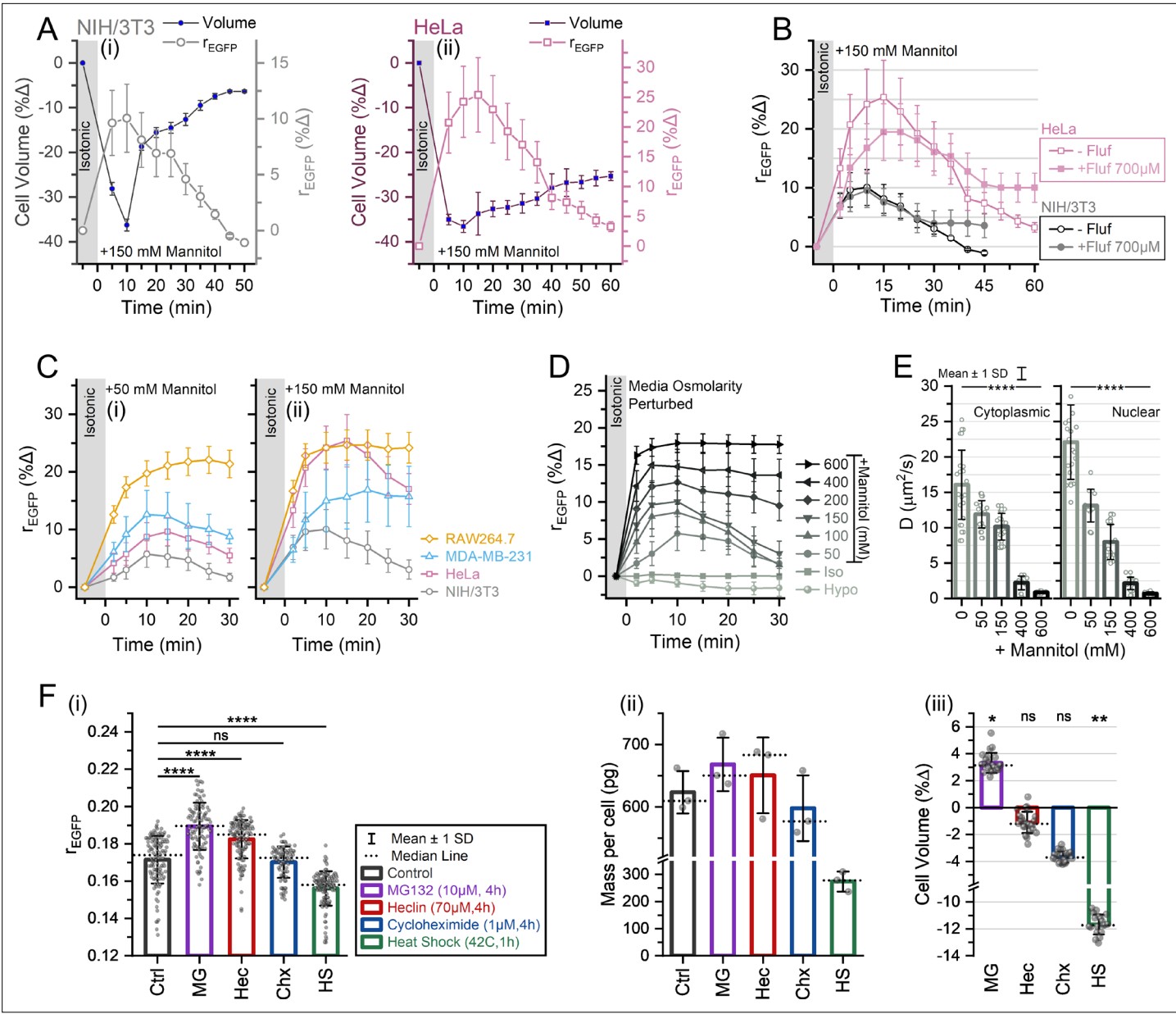

**Figure 5.** Proteostasis disruption alters cellular macromolecular crowding (MMC) setpoint. (**A**) Average percentage changes in cell volume (filled symbols) and modal $r_{EGFP}$ (open symbols) for NIH/3T3 in (**i**) and HeLa in (**ii**) upon exposure to hypertonic mannitol (150 mM) (n>40 cells; N=4 for modal $r_{EGFP}$ data, n>10 cells; N=3 for cell volume data). (**B**) Testing if the $r_{EGFP}$ recovery is mediated by hypertonicity-induced cation channels (HICCs) using flufenamic acid. Closed symbols represent cells pre-treated with 700 μM flufenamic acid for 1 hr; open symbols represent untreated cells. Average percentage change in modal $r_{EGFP}$ (n>40 cells; N=2 for each case) plotted with the standard deviation (SD) error bars. (**C**) Hypertonic shock response of different cell lines estimated through $r_{EGFP}$ measurements. Cells were subjected to an additional 50 mM or 150 mM hypertonicity using mannitol, and the average percentage change in modal $r_{EGFP}$ and its SD is depicted (n>40 cells, N>3 for each case). (**D**) Response of NIH/3T3 fibroblasts to different strengths of hypertonicity (by mannitol) and 50% hypotonicity. The ability of cells to recover $r_{EGFP}$ within 30 min decreases with increasing hypertonicity (n>50 cells, N≥3). (**E**) Diffusion rates of cytoplasmic and nuclear EGFP population estimated through fluorescence recovery after photobleaching (FRAP). The average and SD are shown (n>12 cells for each case). Statistical significance was evaluated by unpaired t-test against the isotonic condition. **** indicates p<0.0001. (**F**) Isotonic perturbations of intracellular MMC by proteostasis disruption - blocking protein degradation (MG 132 and heclin), protein translation (cycloheximide), or inducing widespread protein degradation using heat shock. The mean and SD of the modal $r_{EGFP}$ values are represented in (**i**) (n=132, 98, 138, 90, 130 cells; N=3). Statistical analysis was performed using Kruskal-Wallis ANOVA after Bonferroni correction, followed by Mann-Whitney test for every group pair. **** indicates p<0.00002. Corresponding to (**i**), the intracellular protein mass under each condition is illustrated in (**ii**) (N=3), and the percentage change in the average volume before and after treatment in (**iii**) (n>20 cells; N>2 for each case). Statistical significance was evaluated by paired sample t-test. ** indicates p<0.01, * indicates p<0.05.

*Figure 5 continued on next page*

*Figure 5 continued*

The online version of this article includes the following source data and figure supplement(s) for figure 5:

**Source data 1.** Data tables for *Figure 5F*.

**Figure supplement 1.** Proteostasis disruption alters cellular macromolecular crowding (MMC) setpoint.

We additionally used FRAP to measure the translational mobility of EGFP in the cytoplasm and nucleus of NIH/3T3 cells, within a time window of 10–15 min after introducing various strengths of extracellular hypertonicities (*Figure 5E*). With increasing extracellular hypertonicity, the increase of the average modal $r_{EGFP}$ correlated with a decrease in the average translational diffusion rates of cytoplasmic EGFP (*Figure 5—figure supplement 1F*). The diffusion rate of EGFP was faster in the nucleus than in the cytoplasm during isotonic conditions, but the mobility of EGFP in the nucleus and cytoplasm became similar during hypertonic conditions and decreased with increasing hypertonicity. EGFP was nearly immobile at 600 mM hypertonicity. Thus, the MMC-mediated elevated microviscosity during hypertonic conditions decreases the mobility of both cytoplasmic and nucleoplasmic proteins, enough to be resolvable by FRAP. Comparing Extracellular osmotic imbalances change the cellular MMC through water efflux/influx, but the total number of intracellular proteins (the most abundant macromolecules) can be assumed to be constant during the first 10 min of osmotic stress. To directly alter the number of macromolecules in the cell, we disrupted cellular proteostasis in NIH/3T3 cells by: (i) increasing MMC through protein degradation inhibition via treatments with MG132 (proteasome inhibitor) or heclin (HECT E3 ubiquitin ligase inhibitor) (*Mund et al., 2014*), and (ii) decreasing MMC through protein translation inhibition via cycloheximide treatment (*Siegel and Sisler, 1963*), or widespread protein degradation via heat shock (*Parag et al., 1987*). We estimated the intracellular MMC using $r_{EGFP}$ (*Figure 5F-i*), the average protein mass per cell using the Bradford assay (*Figure 5F-ii*; *Guo et al., 2017*), and cell volume using 3D confocal scans (*Figure 5F-iii*). Cycloheximide treatment caused non-significant changes in MMC and cell volume after 4 hr of treatment. MG132, heclin, and heat shock treatments altered the intracellular MMC, which qualitatively scaled with the changes in cell volume and protein mass per cell. However, despite the changes in cell volume and protein mass, cells failed to achieve the MMC levels of the untreated condition. Therefore, we concluded that cells cannot maintain MMC homeostasis when the general cellular proteostasis is disrupted, and thus the MMC setpoint is altered.

## Hypertonic stress-induced NFkB activation is mediated by TNFR1

Hypertonic stress disrupts numerous physiological functions in a cell which might eventually lead to apoptosis (*Maeno et al., 2000*; *Kültz, 2004*; *Maeno et al., 2006*; *Burg et al., 2007*; *Kwon et al., 2009*). The transcription factor NFkB (nuclear factor kappa beta) plays a major role in protecting cells from apoptosis (*Taniguchi and Karin, 2018*), and has been shown to upregulate osmoprotective genes that promote cell survival during hypertonic stress (*Casali et al., 2018*; *Eisner et al., 2006*; *Farabaugh et al., 2017*; *Németh et al., 2002*; *Roth et al., 2010*). Particularly, hypertonic stress-induced NFkB activity leads to the downregulation of aquaporin 2 (*Hasler et al., 2008*), implying the involvement of NFkB in the cellular RVI mechanism. NFkB is a transcription factor family comprising the p65 (Rel A), p50 (p105), p52 (p100), p68 (Rel B), and p75 (c-Rel) subunits, and in the absence of cellular stresses, the inactive p65-p50 heterodimers are sequestered in the cytoplasm by IkB (inhibitor of kappa beta) (*Sun and Carpenter, 1998*; *Sung et al., 2009*; *Inoue et al., 2016*). Stress induction leads to phosphorylation and proteasome-mediated degradation of IkB, leading to the subsequent release and nuclear translocation of the p65 subunit, which activates the NFkB-mediated cell survival pathways. We had observed that during low-to-moderate levels of hypertonic stress (50–150 mM), NIH/3T3 and HeLa successfully reverted their intracellular MMC through RVI, but at severe hypertonicities (600 mM), neither cell line could recover their MMC (*Figure 5D* and *Figure 5—figure supplement 1C*). To gain a mechanistic insight behind the failure of RVI at severe hypertonic stresses, we investigated NFkB activity by quantifying the fraction of the total cellular p65 content inside the nucleus (identified by Hoechst co-staining) from immunofluorescence images (*Figure 6A and B*). We compared the p65 nuclear fraction in HeLa cells during moderate hypertonic stress (150 mM mannitol), where cells shrunk appreciably and demonstrated MMC recovery through RVI (*Figure 5A-ii*), and during severe hypertonic stress (600 mM mannitol), where cells do not recover their isotonic MMC (*Figure 5—figure*

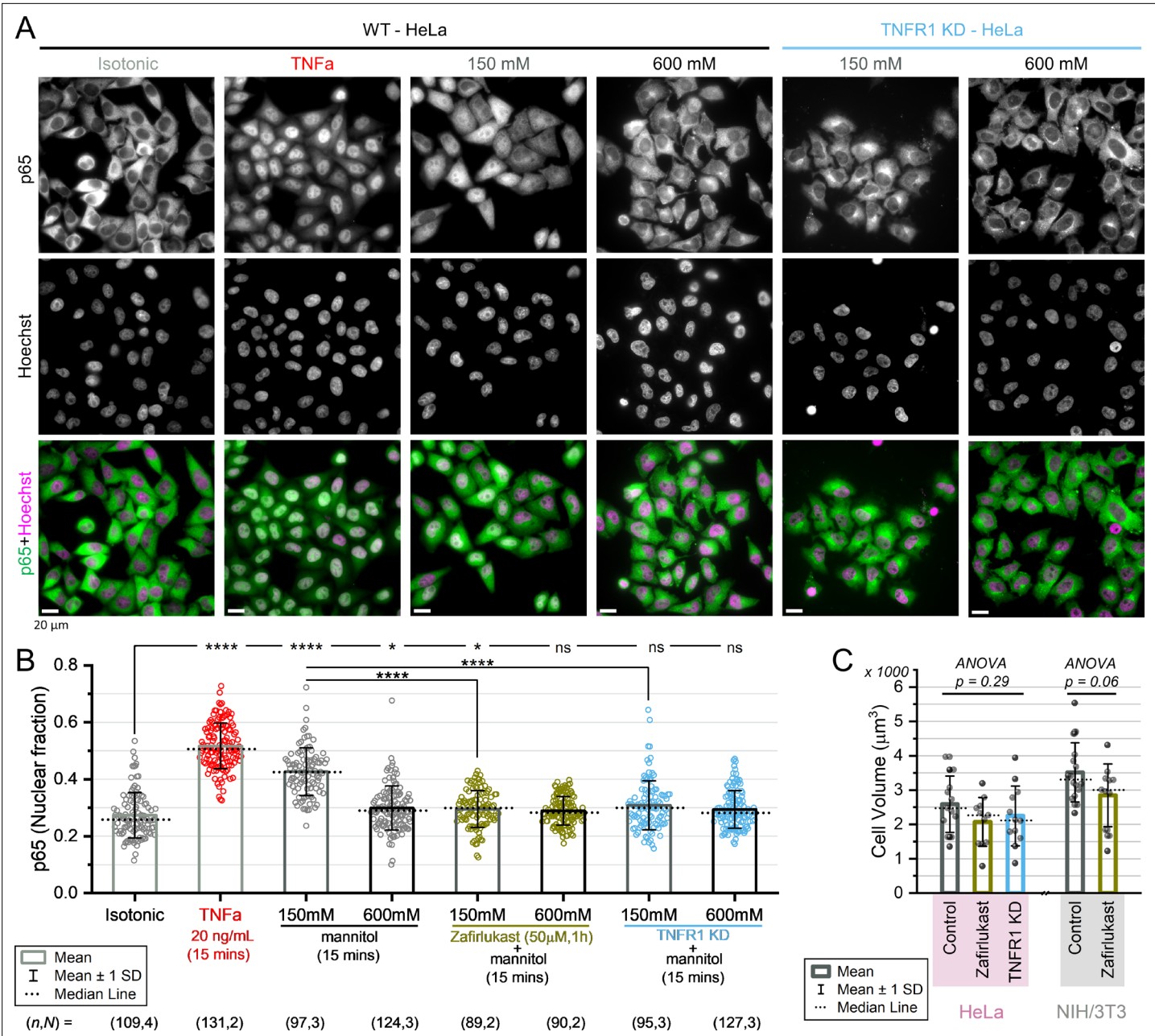

**Figure 6.** Hypertonic stress-induced NFkB activation is mediated by TNFR1. (**A**) Nuclear translocation of p65 visualized through immunofluorescence and Hoechst co-staining for wild-type and TNFR1-knockdown (TNFR1-KD) HeLa nuclei. Nuclear translocation of p65 indicates NFkB pathway activation upon 15 min of treatment with soluble human TNFa (20 ng/mL) or hypertonic mannitol (150 mM and 600 mM). All scale bars represent 20 μm. (**B**) Quantification of p65 nuclear translocation from immunofluorescence images of HeLa cells under indicated conditions. Statistical analysis was performed using Kruskal-Wallis ANOVA after Bonferroni alpha-correction, followed by Mann-Whitney test for the indicated pairs. * indicates p<0.00625, **** indicates p<0.0000125. (**C**) Quantification of cell volume under indicated conditions (*n*=13, 11, 12 cells; *N*=2 for HeLa, and *n*=16, 11 cells; *N*=3 for NIH/3T3). TNFR1 inactivation leads to a decrease in cell volume in both HeLa and NIH/3T3, although the volume changes are statistically insignificant.

The online version of this article includes the following source data and figure supplement(s) for figure 6:

**Source data 1.** Data tables for *Figure 6B and C* and *Figure 6—figure supplement 1C*, and raw images for the immunofluorescence panel in *Figure 6A* and immunoblots in *Figure 6—figure supplement 1B*.

**Figure supplement 1.** Hypertonic stress-induced NFkB activation is mediated by TNFR1.

*supplement 1C*). As a positive control (*Figure 6A*), we treated cells with soluble human TNFa (tumor necrosis factor-alpha), a pro-inflammatory cytokine and known activator of NFkB (*Hayden and Ghosh, 2014*; *Liu et al., 2017*). 150 mM mannitol activated a higher level of p65 than 600 mM mannitol (*Figure 6A and B*), but the level of nuclear p65 in HeLa exposed to 150 mM mannitol was distinguishably less than that induced by TNFa, indicating partial activation of the NFkB pathway. Additionally, the levels of nuclear p65 varied with time under both TNFa and 150 mM hypertonicity compared to the isotonic baseline, but there was no nuclear shuttling of p65 in cells exposed to 600 mM hypertonicity (*Figure 6—figure supplement 1A*). Then again, the TNF receptor-1 (TNFR1) complex, the primary receptor of soluble TNFa, has been shown to cluster and internalize during hypertonic stresses even without the presence of ligands, which might also lead to NFkB activation (*Rosette and Karin, 1996*; *Lo et al., 2020*; *Kucka and Wajant, 2020*; *Su et al., 2022*). Hence, to distinguish between the TNFR1-mediated and hypertonicity-mediated NFkB activity, we blocked TNFR1 clustering using zafirlukast, a pharmacological inhibitor of TNFR1 oligomerization (*Weinelt et al., 2021*). Surprisingly, we found a significantly less nuclear fraction of p65 upon exposure to 150 mM mannitol in zafirlukast-treated cells. Moreover, siRNA-mediated TNFR1-knockdown (TNFR1-KD) HeLa elicited similar results (*Figure 6A and B*, knockdown estimation in *Figure 6—figure supplement 1B*). Thus, we concluded that NFkB activity is correlated with cellular capacity for RVI, and hypertonicity-induced NFkB activation is mediated by TNFR1. Additionally, we observed that both zafirlukast-treated cells and TNFR1-KD cells had smaller volumes on average (*Figure 6C*), and the corresponding MMC levels in the TNFR1-incapacitated cells were significantly higher (*Figure 6—figure supplement 1C*). Therefore, we speculated that TNFR1 activity might also be involved in regulating the cell volume and MMC setpoints.

## TNFR1 activity is essential for RVI

The effect of TNFR1 inactivation on cell volume and MMC inspired us to probe the RVI in TNFR1-KD and zafirlukast-treated cells. RVI was drastically hindered in HeLa and NIH/3T3 for the moderate hypertonic stress of 150 mM mannitol (*Figure 7A*). Surprisingly, the cell volume shrinkage at 10 min post hypertonicity induction decreased upon both TNFR1 knockdown (for HeLa) and zafirlukast treatment (for HeLa and NIH/3T3) (*Figure 7—figure supplement 1A*). Furthermore, using CAY10512 to block NFkB activation (*Heynekamp et al., 2006*), we found a significant reduction in cellular RVI although the cell volume shrinkage at 10 min was comparable with the control cells. Additionally, we used the difference in cell volumes 10 min after exposure to 150 mM mannitol and our final measurement time point to calculate the volume recovery index (*Figure 7A*, insets). We found that in HeLa, the control cells (no pre-treatment) recovered ~11% of their volume in 70 min after their initial shrinkage at 10 min, while CAY10512-treated cells recovered only to ~6%, and TNFR1-KD or zafirlukast-treated cells lost their volume by ~1% and~8%, respectively (*Figure 7A-i*, inset). For NIH/3T3, the control cells (no pre-treatment) recovered up to ~30% of their volume within 50 min after their initial shrinkage at 10 min, while zafirlukast-treated cells and CAY10512-treated cells recovered ~3% and ~9% of their volumes, respectively (*Figure 7A-ii*, inset). Akin to volume recovery, TNFR1-KD, or zafirlukast-treated HeLa and NIH/3T3 had impeded MMC recovery in 150 mM hypertonic stress, as revealed through $r_{EGFP}$ measurements (*Figure 7B*). As TNFR1 inactivation had deleterious effect on hypertonicity-mediated cell volume shrinkage (*Figure 7—figure supplement 1A*) as well as volume recovery (*Figure 7A*), we further probed the effects of TNFR1 inactivation on cell volume changes during the isotonic physiological condition of cells spreading on fibronectin-coated glass. Interestingly, zafirlukast-mediated TNFR1 inactivation also severely decelerated the increase in the average area and volume of NIH/3T3 cells during spreading (post 15 min of settlement) (*Figure 7C*). Thus, TNFR1 activity was important for cell volume control not only during hypertonic conditions, but also isotonic physiological conditions that involved dynamic cell volume changes.

Cellular RVI was dependent on the dose of hypertonicity and correlated with TNFR1-NFkB activity, so we sought to understand the molecular mechanism behind the lack of RVI at severe hypertonicities. Upon ligand-induced activation, TNFR1 molecules trimerize and the oligomeric clusters recruit TRADD (TNFR1-associated death domain), TRAF2 (TNFR-associated factor 2), and RIPK1 (receptor-interacting serine/threonine-protein kinase 1) at the plasma membrane to form the TNFR1 signaling complex. Linear ubiquitination of RIPK1 facilitates the formation of a scaffolding-like architecture that promotes enhanced phosphorylation of the IKK protein family, which subsequently phosphorylates

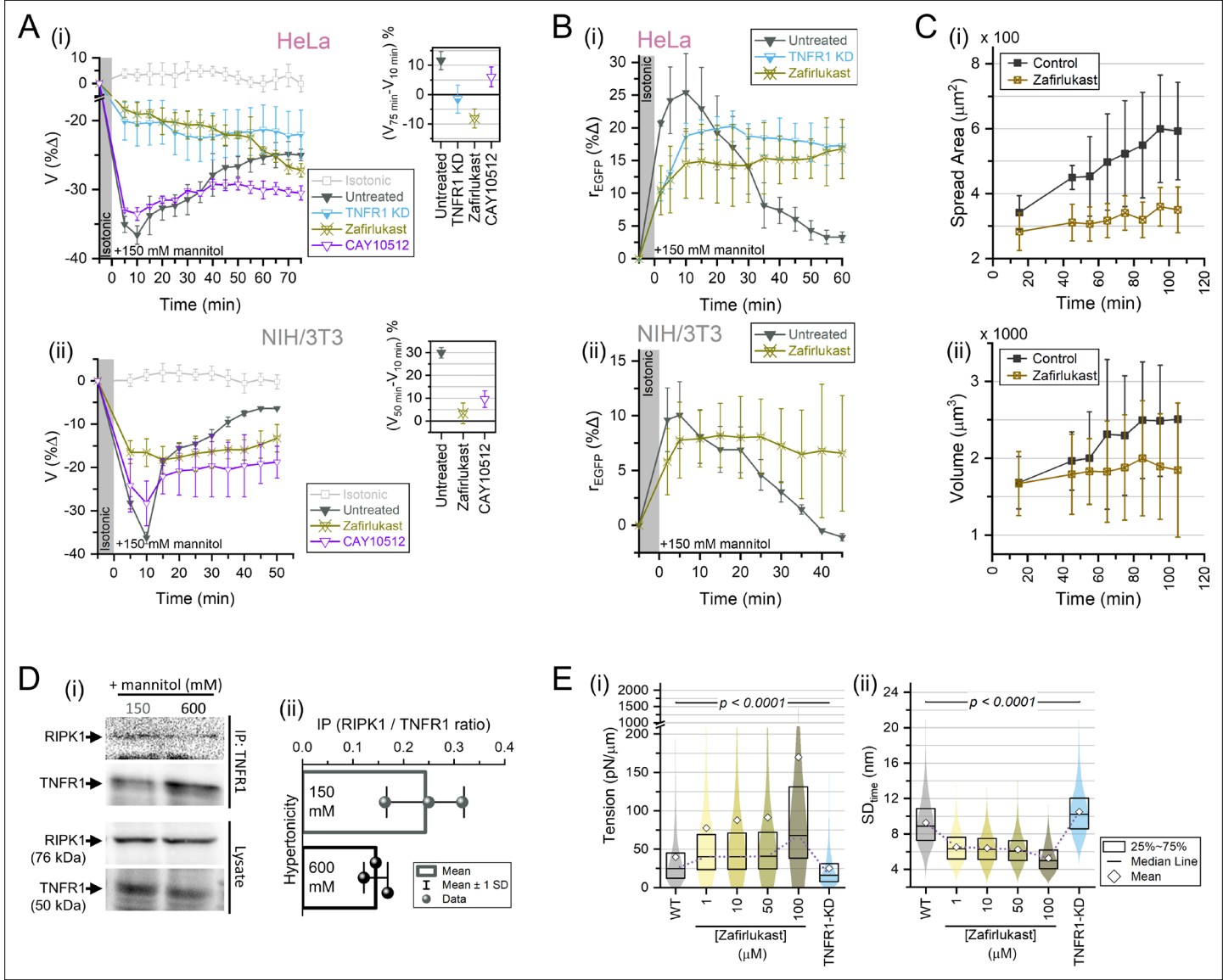

**Figure 7.** TNFR1 activity is essential for regulatory volume increase (RVI). (**A**) Percentage change in the volume of HeLa (**i**) and NIH/3T3 (**ii**) cells while exposed to 150 mM hypertonicity under no pre-treatment, TNFR1-knockdown (TNFR1-KD) condition (HeLa only), zafirlukast pre-treatment (50 µM, 1 hr), and CAY10512 pre-treatment (250 nM, 1 hr), compared to cell volume fluctuations in isotonic conditions. Mean volume and SD plotted for $n \geq 10$ cells, $N \geq 2$ in each case. Insets show the RVI index (percentage change between cell volumes at the final time point of measurement vs at 10 min post hypertonicity induction) for each condition. (**B**) Percentage change in modal $r_{EGFP}$ for HeLa (**i**) and NIH/3T3 (**ii**) cells during hypertonic stress - under no pre-treatment, TNFR1-KD condition (HeLa only), and zafirlukast pre-treatment (50 µM, 1 hr). Mean percentage change and SD plotted for $n \geq 40$ cells, $N \geq 2$ in each case (**C**) Cell spread area trajectory (**i**) and corresponding cell volume trajectory (**ii**) for vehicle control and zafirlukast (50 µM,1 hr) treated NIH/3T3 cells spreading on fibronectin-coated glass. (**D-i**) Immunoprecipitated endogenous TNFR1 and associated RIPK1 under indicated conditions and their expression levels in the whole-cell lysate of wild-type (WT) HeLa cells visualized through immunoblotting; (**D-ii**) is the quantification of RIPK1 content normalized by immunoprecipitated TNFR1 content during hypertonic stress. (**E**) Comparison of the membrane tension in (**i**) and corresponding $SD_{time}$ of membrane fluctuations in (**ii**) of HeLa cells for WT controls, different doses of zafirlukast, and TNFR1-KD. The 25th and 75th percentiles, medians, and means are shown for $N \geq 2$, WT: 51402 FBRs, 46 cells; zafirlukast - 1 µM: 9723 FBRs, 13 cells; 10 µM: 9357 FBRs, 14 cells; 50 µM: 14093 FBRs, 14 cells; 100 µM: 11690 FBRs, 16 cells; TNFR1-KD: 24273 FBRs, 33 cells. Statistical analysis was performed by Mann-Whitney test for every distribution against the WT control.

The online version of this article includes the following source data and figure supplement(s) for figure 7:

**Source data 1.** Data tables for *Figure 7E* (.opj file format) and *Figure 7—figure supplement 1A*, and raw images for the immunoblots in *Figure 7D-i*.

**Figure supplement 1.** TNFR1 activity is essential for regulatory volume increase (RVI).

IkB and initiates the nuclear translocation of p65 (*He and Wang, 2018*; *Mihaly et al., 2014*; *Shi and Sun, 2018*; *Ting and Bertrand, 2016*; *Tu et al., 2021*; *Webster and Vucic, 2020*). RIPK1 recruitment and function is one of the pivotal determinants of the pro-survival TNFR1-NFkB signaling pathway (*Mifflin et al., 2020*), so we probed the levels of RIPK1 recruitment to the TNFR1 complex under different hypertonic stress using immunoprecipitation assays. In HeLa cells, severe hypertonic stresses (600 mM mannitol) had reduced TNFR1-associated RIPK1 than moderate hypertonic stresses (150 mM mannitol) (*Figure 7D*), suggesting that TNFR1 signaling was incapacitated at severe hypertonic stresses. We hypothesized that the impaired recruitment of RIPK1 at the TNFR1 complex during severe hypertonic stresses was due to the MMC-mediated rise in cytoplasmic microviscosity (*Figure 5D*). The absence of TNFR1 signaling further impeded NFkB activity, delaying the onset of RVI and establishing the pivotal role of TNFR1 in modulating RVI. While the impaired mobility of RIPK1 explained the lack of TNFR1-NFkB signaling, the physicochemical reason behind the reduced hypertonic volume shrinkage in zafirlukast and TNFR1-KD cells remained elusive (*Figure 7—figure supplement 1A*). Additionally, zafirlukast or TNFR1-KD reduced the average cell volume (*Figure 6C*) and increased the cellular MMC (*Figure 6—figure supplement 1C*). Since extracellular hypertonicity reduces cortical shear modulus (*Guo et al., 2017*) and membrane tension (*Roffay et al., 2021*), we hypothesized that zafirlukast or TNFR1-KD could alter the cortex or membrane tension, enabling cells to mechanically resist the hypertonic volume deformations (*Venkova et al., 2022*). Membrane tension could also implicitly reduce the hypertonic volume shrinkage by altering the functionality of different membrane proteins, like aquaporins (*Soveral et al., 2008*; *Ozu et al., 2013*; *Jiang et al., 2021*). Using interference reflection microscopy (IRM), we measured the shape fluctuation autocorrelations of the basolateral membrane, allowing membrane tension estimation (*Biswas et al., 2017*). We found that zafirlukast treatment increased tension in a dose-dependent manner, but surprisingly, TNFR1-KD reduced the tension (*Figure 7E* and *Figure 7—figure supplement 1B*). Therefore, we eliminated the causal role of membrane tension in reducing hypertonic cell shrinkage, and we could only speculate that the altered setpoints of cell volume and MMC upon TNFR1 inhibition might be connected to the cellular resistance to hypertonic volume shrinkage.

## Intracellular MMC deviates from the concentration-dilution law under hypertonic stress

The cellular macromolecule concentration should be inversely proportional to the cell volume if the number of macromolecules remains unchanged. Challenged by 150 mM hypertonicity, the MMC ($r_{EGFP}$) of NIH/3T3 and HeLa cells scaled in proportion with the cell volume (*Figure 5A*). For both NIH/3T3 and HeLa cells, the MMC peaked 10 min after hypertonicity induction, indicating equilibration of the intra- and extracellular osmolarities. However, NIH/3T3 showed an ~35% volume shrinkage and ~10% MMC elevation, while HeLa showed a similar ~35% volume shrinkage but an ~25% MMC elevation. We could attribute the difference of hypertonic stress response in the cellular MMC to the observed variability in different cell lines (*Figure 5C*), but the discrepancy in the volume shrinkage vs MMC elevation challenged the concentration-dilution law: $N_i \cdot V_i = constant$, where $N_i$ is the solute concentration and $V_i$ is the solvent volume for the solution $i$. Since $r_{EGFP}$ scaled linearly with macromolecule concentration (*Figure 1A*), we expressed the macromolecule concentration ($[MMC]$) as a linear function of $r_{EGFP}$ as: $[MMC] = m \cdot r_{EGFP} - m \cdot \alpha$, where $1/m$ is the slope in *Figure 1A* and $\alpha$ is the $r_{EGFP}$ value at zero crowder concentration. Thus, according to the concentration-dilution law, $(r_{EGFP,1} - \alpha) \cdot V_1 = (r_{EGFP,2} - \alpha) \cdot V_2$ at any condition, provided the number of macromolecules in the cell or the total excluded volume is constant. Post hypertonicity induction, the MMC of NIH/3T3 cells equilibrated at 10 min for every dose of hypertonicity tested by us (*Figure 5D*). So, we used the Boyle-van't Hoff (VBH) relation to model the equilibrium cell volume compression at different hypertonicities and find the osmotically inactive cell volume at infinite hypertonicity (*Katkov, 2011*; *Roffay et al., 2021*; *Venkova et al., 2022*). At equilibrium, the average cell volumes scaled with the average modal $r_{EGFP}$ values in accordance with the applied hypertonicity (*Figure 8A-i*). Using the VBH relation, we computed the osmotically inactive cell volume ($\cong 284$ μm³) and the limiting $r_{EGFP}$ ($\cong 0.23$) for NIH/3T3 cells (*Figure 8A-ii*) (normalized according to the relations: *Cell volume* $\propto 1/Osmotic\ Pressure$ and $1/MMC \propto 1/Osmotic\ Pressure$). We related the pair of $(r_{EGFP}, V)$ points at isotonic and infinite hypertonicity using the concentration-dilution law and extracted $\alpha = 0.169$ for intracellular EGFP. The intracellular $r_{EGFP}$ vs $V$ of NIH/3T3 populations at different hypertonic strengths deviated from the

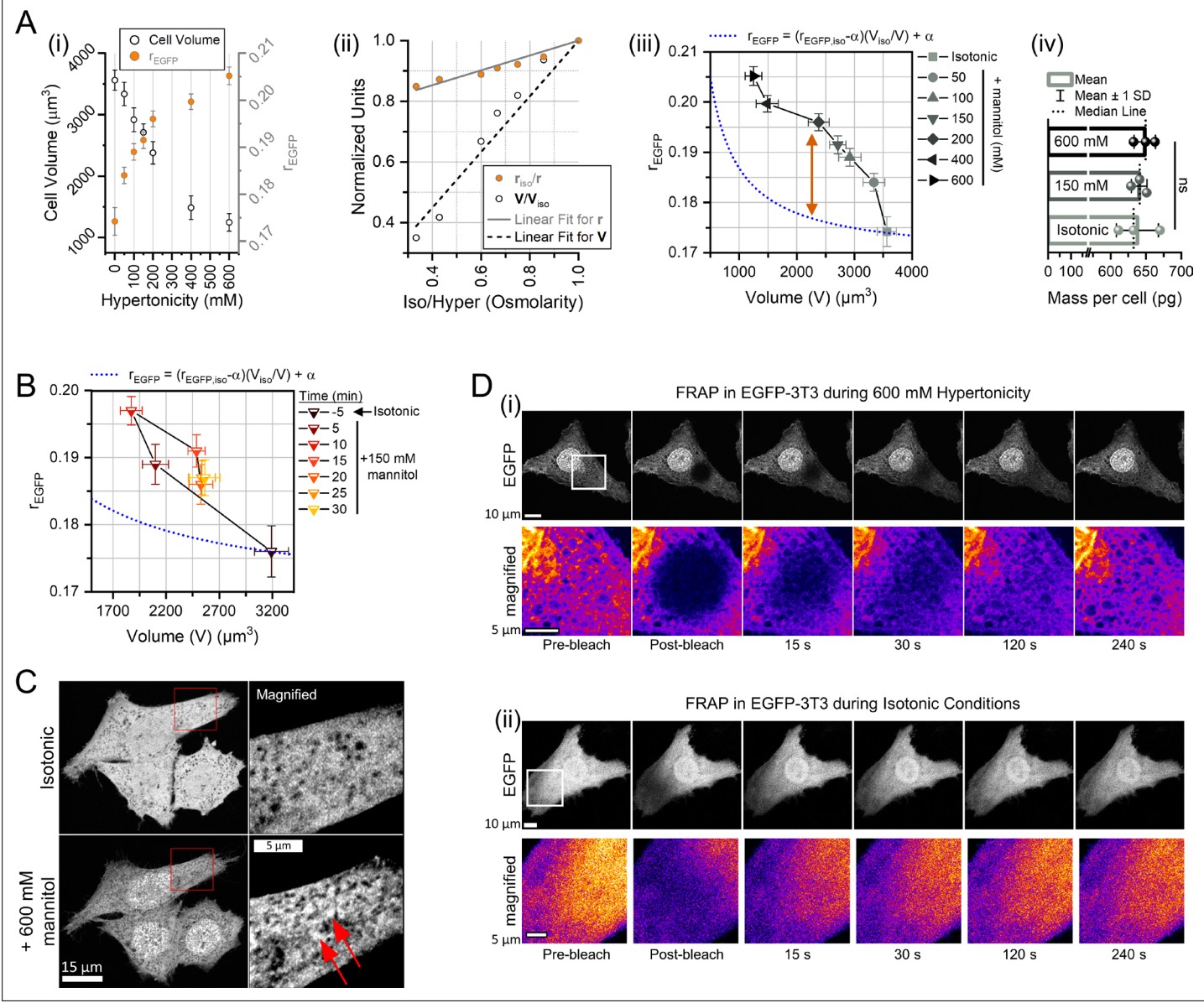

**Figure 8.** Intracellular macromolecular crowding (MMC) deviates from the concentration-dilution law under hypertonic stress. (**A**) The average and standard deviation of cell volume and $r_{EGFP}$ of NIH/3T3 cells at different hypertonicities measured 10 min post hypertonic stress induction, shown in (**i**) (0 indicates isotonic), were normalized to fit the Boyle-van't Hoff relation, as shown in (**ii**). The resultant points were fit to a straight line forced to pass through (1,1) - the isotonic condition, such that the y-intercept gives the limiting cell volume (osmotically inactive cell volume) and limiting $r_{EGFP}$. Using the $r_{EGFP}$ and cell volume values at the limiting and isotonic conditions, the $r_{EGFP}$ vs cell volume trendline representing the concentration-dilution law was calculated (blue dashed line, formula in legend) in (**iii**). The expected trendline of $r_{EGFP}$ vs cell volume deviates from the measured values (gray symbols), indicated by the double-headed arrow, even though the total protein mass per cell at different hypertonic conditions do not change at different hypertonicities, as shown in (**iv**). (**B**) Trajectory of $r_{EGFP}$ vs cell volume at different time points after inducing hypertonic shock (150 mM mannitol) (n=7 cells; N=2). The blue dashed line denotes the theoretical estimate of the trajectory, as indicated in the legend. (**C**) AiryScan super-resolution imaging of NIH/3T3-EGFP cells reveals submicron-sized cluster-like appearance of EGFP under severe hypertonic stress (600 mM mannitol). The brightness-contrast in the magnified insets was individually adjusted for better visualization. (**D**) Time-lapse of EGFP intensity after photobleaching under hypertonic stress of 600 mM mannitol (**i**) and isotonic conditions (**ii**) in NIH/3T3. Pseudocolored bottom panels show the magnified photobleaching area (white squares).

The online version of this article includes the following source data and figure supplement(s) for figure 8:

**Figure supplement 1.** Intracellular macromolecular crowding (MMC) deviates from the concentration-dilution law under hypertonic stress.

**Figure supplement 1—source data 1.** Raw images for *Figure 8—figure supplement 1A*.

**Figure supplement 2.** Intracellular macromolecular crowding (MMC) deviates from the concentration-dilution law under hypertonic stress.

expected relation: $r_{EGFP} = (r_{EGFP,iso} - 0.169) \frac{V_{iso}}{V} + 0.169$, ('iso' → isotonic condition) (*Figure 8A-iii*). The deviation of the measured $r_{EGFP}$ vs $V$ curve in *Figure 8A-iii* suggested the presence of other physical processes that alter $r_{EGFP}$ besides hypertonic volume change. We speculated that the hypertonic conditions either increased the total number of macromolecular crowders, or caused associative reorganization of the crowders that increased the total excluded volume and amplified the refractive index of the protoplasm, elevating $r_{EGFP}$ beyond the expected value. The total protein mass per cell (*Figure 8A-iv*) did not noticeably differ between 150 mM and 600 mM hypertonicities, indicating no measurable change between intracellular crowder numbers. Simultaneous measurements of $r_{EGFP}$ and volume of NIH/3T3 cells under hypertonic stress (150 mM mannitol) also showed a similar deviation from the expected behavior (*Figure 8B*). Therefore, we speculated that hypertonicity-induced reorganization of the intracellular crowders could enforce the deviation of $r_{EGFP}$ from the concentration-dilution law.

In AiryScan super-resolution images of NIH/3T3-EGFP, we found that the intensity profile of EGFP was more non-uniform and puncta-like at severe hypertonicities compared to isotonic conditions (*Figure 8C*). Photobleaching studies on such cells revealed that the translational mobility of EGFP was severely compromised compared to isotonic conditions (*Figure 8D*). Moreover, we found an increase of intracellular subspace devoid of EGFP (EGFP excluded cytoplasmic volume [EECV]). The structure of EECVs inside the cell did not change even within a span of 4 min, and EGFP molecules diffused through the interstitial spaces around the EECVs (*Figure 8D*). Hypertonic stress also caused severe DNA condensation in the nucleus, which confirmed previous reports (*Irianto et al., 2013*). Literature suggests that extracellular hypertonicities can trigger condensation of multivalent proteins and mRNA through liquid-liquid phase separation (LLPS) (*André and Spruijt, 2020*; *Bounedjah et al., 2012*; *Carrettiero et al., 2022*; *Delarue et al., 2018*; *Keber et al., 2021*; *Shin and Brangwynne, 2017*; *Watanabe et al., 2021*; *Watson et al., 2023*; *Yasuda et al., 2020*). The subunits of the NFkB family are multimeric, possess intrinsically disordered domains, and have DNA binding motifs (*Riedlinger et al., 2019*; *Baughman et al., 2022*; *Komives, 2023*), the essential characteristics of a protein that could condensate via LLPS. In NIH/3T3-EGFP cells immunostained for p65 under 150 mM hypertonicity, we found condensate-like granular structures of p65 in the cytoplasm that exclude EGFP from within (*Figure 8—figure supplement 1A*). In live NIH/3T3 cells overexpressing p65-GFP, we observed the granular structures even at low hypertonicities (50 mM mannitol), which disappeared immediately as the cells were rescued to isotonic culture media (*Figure 8—figure supplement 1B*), and an identical effect was observed in severe hypertonic conditions (*Figure 8—figure supplement 2A*). The propensity of granule formation increased with the p65-GFP expression levels, and only ~20% of the cells showed p65 granules. However, at severe hypertonicities (600 mM mannitol), 100% of the transfected cells showed p65 granules (*Figure 8—figure supplement 1C*). The p65 granules at 600 mM hypertonicity appeared smaller and uniformly spherical, while those at 50 mM hypertonicity were larger and more non-uniformly shaped. Photobleaching individual p65-GFP granules showed slow fluorescence recovery of the complete structure, confirming material exchange with the cytosol and the fluid nature of the granules (*Figure 8—figure supplement 2B*). Therefore, we speculated that the EECVs were created by hypertonicity-induced condensation of multivalent proteins like WNK kinases (*Boyd-Shiwarski et al., 2022*), DCP1A (*Jalihal et al., 2020*), YAP (*Cai et al., 2019*), ASK3 (*Watanabe et al., 2021*), and additionally, NFkB family proteins, which enforced the deviation of intracellular MMC from the concentration-dilution law.

## Discussion

Our in vitro and in-cell fluorescence anisotropy measurements of EGFP establish $r_{EGFP}$ as a robust technique to quantitate intracellular MMC. *Figure 1A* shows that among different biomolecules, protein crowding imparts the maximum effect on $r_{EGFP}$, presumably because proteins have the highest average molecular weight among biomolecules, and therefore, the highest molecular polarizability (*Booth et al., 2022*). Thus, proteins have the most significant effect on the protoplasmic refractive index, and thereby, $r_{EGFP}$. Protein crowding mediated changes in solution viscosity ($\eta$), fluorescence lifetime ($\tau$), and the intrinsic anisotropy ($r_0$) of EGFP describes the observed changes in $r_{EGFP}$ in vitro (*Figure 1C and D*) and in cells (*Figure 2A*). Variability in pH (*Figure 1G*) or EGFP concentration below the homo-FRET regime (*Figure 1F*) does not affect $r_{EGFP}$, demonstrating its reliability under different

physiological conditions. However, homo-FRET can potentially affect intracellular $r_{EGFP}$ values if a cell is under severe osmotic compression where local EGFP concentrations exceed the threshold of 10 µM. The linear scaling of $r_{EGFP}$ against protein concentration validates its suitability as a quantitative indicator of crowder concentration (*Figure 1A*). However, the presence of uncharacterizable cellular scattering agents depolarizes the fluorescence emission of EGFP, thus finding appropriate parameters to relate in vitro and in-cell $r_{EGFP}$ is non-trivial. In addition, variability in cell height and focusing uncertainties cause media autofluorescence to further affect intracellular $r_{EGFP}$ values in wide-field microscopes, enhancing the spatial heterogeneity of intracellular $r_{EGFP}$. maps (*Figure 3C*). Despite these potential artifacts, we can exploit $r_{EGFP}$ as a reliable probe by considering the modal value of the $r_{EGFP}$ distribution in a cell, which represents the ubiquitous protein crowding levels and neglects the outliers (*Figure 2B*, *Figure 2—figure supplement 1A*). Intracellular $r_{EGFP}$ maps show the lamellar cytoplasm to be less crowded than the perinuclear cell body, and F-actin structures demarcate regions of variable crowding in the cytoplasm (*Figure 3A*). However, since the lamellar cytoplasm is optically thin, we validated its lower crowding using alternate established methods free of cell height artifacts: FRAP to measure translational diffusion kinetics of EGFP (*Figure 3B-i*, *Figure 3—figure supplement 1A-i*, *Figure 4C*; *Bulthuis et al., 2023*), single-particle tracking microrheology (*Figure 3B-ii*; *Delarue et al., 2018*), and fluorescence lifetime imaging (*Figure 3C*; *Levchenko et al., 2018*; *Pliss et al., 2012*). The different modes of probing MMC confirm that the cytoplasm indeed has differential MMC levels. Additionally, a previous fluorescence anisotropy study of intracellular EGFP using selective plane illumination also shows spatial variability, although the authors chose to ignore it (*Hedde et al., 2015*).

For NIH/3T3 fibroblasts, the cell-to-cell variability of $r_{EGFP}$ is free of homo-FRET artifacts (*Figure 2—figure supplement 1C*), thus establishing that the observed population heterogeneity of intracellular MMC is uncorrelated to the cell cycle stage (G1, S, or G2) (*Figure 2C*). If the spread area does not change significantly, NIH/3T3 fibroblasts tightly maintain the intracellular MMC levels for relatively long timescales (at least 8 hr) (*Video 1*). Although the median levels of intracellular MMC may not vary among cell lines (*Figure 2E*), the hypertonic stress response varies (*Figure 5C*). A reduction in cell spread area upon substrate detachment leads to increased intracellular protein crowding (*Figure 4C and D*) and cytoskeletal depolymerization (*Figure 4—figure supplement 1C*). Contrarily, increasing cell spread area gradually reduces intracellular MMC and increases cell volume (*Figure 4A*). Our observations seemingly conflict with previous reports that measure cell volume during spreading (*Guo et al., 2017*; *Xie et al., 2018*) possibly because of differing experimental conditions, one of which is using a cell-permeable dye to visualize the whole cell compared to EGFP expression used by us. In alignment with our data, other investigations show that the cell volume initially decreases up to 20 min and then starts rising (*Venkova et al., 2022*). Cell spread area has also been shown to positively correlate with cell volume (*Perez Gonzalez et al., 2018*; *Kumar et al., 2019*), hence we believe our observations to be correct. Interestingly, cytochalasin D-mediated actin depolymerization increases MMC without affecting cell volume, while microtubule and vimentin depolymerization does not affect cellular MMC but increases cell volume (*Figure 3D*), indicating that cytoskeletal polymers may regulate the cell volume-MMC setpoint. Moreover, increased MMC due to cell detachment or actin depolymerization is comparable, and promoting deadhesion in actin-depolymerized cells does not increase cellular MMC substantially (*Figure 4D-ii*, *Figure 4—figure supplement 1B*). Previous reports show that increased MMC accelerates actin polymerization (*Rashid et al., 2015*), and actin polymerization is upregulated during the initial hours of cell spreading (*Reinhart-King et al., 2005*). Whether the elevated MMC in substrate-detached cells drives actin polymerization and increases cell spreading area/volume would be an interesting study since actin cytoskeletal proteins have been implicated in regulating cell volume (*Papakonstanti et al., 2000*; *Hoffmann et al., 2009*; *Mills et al., 2020*).

Proteostasis disruption by MG132, heclin, and heat shock alter the intracellular protein crowding without a substantial change in cell volume, implying that NIH/3T3 fibroblasts tolerate at least ~12% change in the MMC setpoint for 4 hr when under isotonic conditions (*Figure 5F*). Contrarily, even an ~5% change in cellular MMC due to extracellular hypertonicity is rectified by RVI (*Figure 5A and D*), implying that cellular osmosensing mechanism is different from MMC-sensing and probably involves cell volume sensing machinery. The recovery of intracellular MMC after hypertonicity induction varies among different cell lines (*Figure 5C*), and HeLa cells recover their MMC even without volume recovery (*Figure 5A*), possibly using alternate mechanisms like osmolyte accumulation (*Burger-Kentischer*

*et al., 1999*; *Burg and Ferraris, 2008*). Remarkably, while moderate hypertonicities (150 mM) elicit RVI in NIH/3T3 and HeLa (*Figure 5A*), both cell lines lose their ability to recover their MMC at severe hypertonicities (600 mM) (*Figure 5D*, *Figure 5—figure supplement 1C–E*). Cellular RVI at moderate hypertonicities correlates with NFkB activity, and surprisingly, knockdown, or pharmacological inhibition of TNFR1, an upstream effector of NFkB signaling, prevents its activation and thwarts RVI (*Figure 6A and B*, *Figure 7A*), indicating an osmosensing activity of TNFR1. Furthermore, at severe hypertonicities, the cytoplasmic viscosity increases 15-fold (*Figure 5E*) and significantly delays the recruitment of RIPK1 at the TNFR1 complex, culminating in the failure of timely TNFR1 activation and RVI. Interestingly, TNFR1 inhibition or knockdown reduces the average cell volume and slows down the hypertonic cell shrinkage (*Figure 7—figure supplement 1A*), but the cause for the slower cell shrinkage rate is unknown and not related to cell membrane tension (*Figure 7E*). We speculate that TNFR1 activity is interconnected with aquaporin levels in the plasma membrane, since aquaporin inhibition also restricts hypertonic cell volume shrinkage (*Krane et al., 2001*; *Hansen and Galtung, 2007*; *Akai et al., 2012*).

Cell volume-MMC kinetics are synchronized during RVI in NIH/3T3 and HeLa cells (*Figure 5A*), yet hypertonicity-induced changes in cell volume and $r_{EGFP}$ do not follow the concentration-dilution law (*Figure 8A and B*). Extracellular hypertonicity creates subspaces within the cytoplasm inaccessible to EGFP molecules (EECV) (*Figure 8C and D*, *Figure 8—figure supplement 1*), conceivably due to LLPS of multiple proteins (*André and Spruijt, 2020*; *Boyd-Shiwarski et al., 2022*; *Cai et al., 2019*; *Carrettiero et al., 2022*; *Jalihal et al., 2020*; *Shin and Brangwynne, 2017*; *Watanabe et al., 2021*; *Yasuda et al., 2020*). The EECV fraction in the cytoplasm increases with the applied hypertonicity, and the intracellular MMC deviates from the concentration-dilution law presumably because of the factors that increase the EECV fraction, which can have aberrant effects on $r_{EGFP}$. In conclusion, our explorations of the cellular MMC-volume interplay illuminate the effects of MMC on cellular biochemical signaling, and we unveil the involvement of TNFR1-NFkB signaling in the cellular RVI process. However, the exact mechanism of hypertonicity-induced TNFR1 activation is still elusive and requires further studies.

## Materials and methods
### Cell culture and pharmacological studies
NIH/3T3 cell line was procured from NCCS (National Center for Cell Science, Pune, India). RAW 264.7 cell line was a generous gift from Dr. Sanjay Dutta (CSIR-Indian Institute of Chemical Biology, Kolkata), while HeLa and MDA-MB-231 cell lines were kindly gifted by Dr. Prosenjit Sen (Indian Association for the Cultivation of Science, Kolkata). FuGENE (Promega, #E2311) was used to transfect cells with the following plasmids: pCAG-mGFP, a gift from Connie Cepko (Addgene plasmid # 14757); 2GFP (GFP-GFP dimer), a very kind gift from Maria Vartiainen (University of Helsinki, Finland) (*Dopie et al., 2012*; *Koskinen and Hotulainen, 2014*); pEGFP-C1 LifeAct-EGFP, a gift from Dyche Mullins (Addgene plasmid # 58470); EGFP-p65, a gift from Johannes A Schmid (Addgene plasmid # 111190); mCherry-Tubulin-6, a gift from Michael Davidson (Addgene plasmid # 55147), and TNFRSF1A DsiRNA (IDT, #hs.Ri.TNFRSF1A.13.1), following standard protocol. Cells cultured in DMEM (Himedia, #Al007G) at 37°C, 5% $CO_2$ in a humidified incubator, were seeded on custom-made glass-bottom 35 mm Petri dishes. The glass was coated with 50 µg/mL of fibronectin (Sigma, #F1141) to promote rapid adhesion and proper spreading or with 10% PEG (PEG-400, Sigma, #CAS: 25322-68-3) to prevent spreading in the appropriate cases. Before microscopy, cells were gently washed with 1× PBS twice, and culture media was replaced with phenol red-free DMEM (Gibco, #21063029), which would be supplemented with the required drug when necessary. For all pharmacological treatments, cytochalasin D (Merck, #C8273), nocodazole (Merck, #487928), withaferin A (Merck, #W4394), heclin (Tocris, #5433), cycloheximide (Sigma, #18079), and zafirlukast (Merck, #Z4152) were dissolved in DMSO, and working concentrations were reconstituted as indicated in appropriate places. For applying heat shock, cells were incubated at 42°C for 1 hr in the presence of 5% $CO_2$. Osmotic imbalances were created by replacing the isotonic complete media with hypertonic or hypotonic complete media using a custom-made flow system. Hypertonic media was prepared by adding mannitol, dextrose, or NaCl (Merck Empura) to phenol red-free DMEM (Gibco, #21063029) at indicated concentrations and filtered for

decontamination. 50% hypotonic media was prepared by adding autoclaved Milli-Q water to equal volumes of phenol red-free DMEM.

## EGFP purification

BL21 (DE3) *Escherichia coli* variant, transformed to express monomeric EGFP, was grown to log phase (OD600 ≅ 0.7) in a 500 mL culture by 12 hr incubation at 37°C. Then, EGFP expression was maximized through isopropyl β-D-1-thiogalactopyranoside induction (40 mg/mL, 37°C, 4 hr). The bacteria were harvested by centrifugation (6500×$g$, at 4°C for 5 min), and the cell pellet was resuspended in 5 mL lysis buffer containing 50 mM Tris-HCl, 150 mM NaCl, 0.1X protease inhibitor, and 1 mg/ml lysozyme. The bacterial cells were then mechanically lysed using a probe sonicator (cycle: 0.5, amplitude: 30%) in an ice bath for 30 min, the debris was separated by centrifugation (10,000×$g$, at 4°C for 40 min), and the supernatant was collected. Proteins heavier than EGFP (MW: 27 kDa) in the supernatant were salted out by the slow addition of 80% ammonium sulfate solution (wt/vol) (up to a final concentration of 20%). The precipitate was centrifuged for removal (13,500×$g$, at 4°C for 45 min), and the remnant proteins in the supernatant, including EGFP, were salted out using 40% ammonium sulfate solution (final concentration). The precipitate was resuspended in 3 mL 50 mM Tris-HCl buffer and was dialyzed against the same buffer overnight with mild stirring at 4°C. The dialyzed solution was subjected to anion exchange chromatography using standard protocols, and the purified EGFP was lyophilized and reconstituted in HEPES (SRL, #63732) buffer of pH (7.2–7.6). The concentration of the reconstituted EGFP was estimated from UV absorbance and FCS. Subsequently, the reconstituted EGFP was diluted to ~50 nM for all experiments (except $r_{EGFP}$ vs EGFP concentration).

## FCS measurements

FCS measurements were performed in solutions diluted from our purified EGFP stock solution using a ×40/1.2 NA water immersion objective on a confocal microscope (Zeiss LSM 780) at 20°C. EGFP was diluted from the stock at the indicated volume fractions in 100 mM HEPES (pH 7.4), then 100 µL of each solution was sandwiched between glass coverslips with ~1 mm space in-between, then sealed airtight and bubble-free. Fluorescence fluctuations were measured for 2 s at a height of 200 µm from the basal coverslip glass, and the averaged autocorrelation data of 200 repetitions was plotted for each solution prepared in triplicate groups. The autocorrelation curve $G(\tau)$ was fit by the built-in curve fitting system to the analytical function for 3D anomalous diffusion: $G(\tau) = 1 + \frac{1}{N} \frac{1-F+F_e^{\frac{-\tau}{\tau_f}}}{1-F} \frac{1}{1+\left(\frac{\tau}{\tau_D}\right)^\alpha \left[\left(1+\frac{1}{S^2}\right)\left(\frac{\tau}{\tau_D}\right)^\alpha\right]^{\frac{1}{2}}}$,

where $N$ is the number of fluorophores in the confocal volume, $\tau$ is the lag time, $F$ is the fraction of fluorophores in the triplet state, and $S$ is the structure parameter ($S = \frac{\omega_Z}{\omega_{XY}}$, with $\omega_Z$ being the axial radius and $\omega_{XY}$ being the lateral radius). The value of the diffusion anomaly parameter, $\alpha$, was fixed at 1 during fitting for simplicity. The triplet state fraction was not accounted for while fitting $G(\tau)$ of the EGFP solutions, and the average diffusion time of EGFP in buffer solutions was measured to be 163±74 µs, while cytoplasmic EGFP had an average diffusion time of 338±103 µs. The number density of fluorophores in the confocal volume $N = 1/(G(0) - 1)$ is independent of fitting parameters, and thus, fitting artifacts can be disregarded. The theoretical values of $\omega_Z$ and $\omega_{XY}$ (for 488 nm light, 1.2 NA objective, and 1.33 refractive index for HEPES buffer) are 901 nm and 248 nm, respectively. The ellipsoidal confocal volume ($V$) thus amounts to ~0.109 fL. Since $G(0) = 1.028$ for the 0.1% dilution (vol/vol), the concentration of EGFP was calculated using $[EGFP] = \frac{N}{N_A}\frac{1}{V}$, which amounted to ~540 nM. Thus, our stock solution of purified EGFP had a concentration of ~540 µM, the maximum [EGFP] depicted in *Figure 1F*. We then measured the fluorescence anisotropy of the same EGFP dilutions in our $r_{EGFP}$ setup. The total intensity values obtained for the different dilutions were plotted against the $1/(G(0) - 1)$ values in *Figure 1—figure supplement 1A*, inset. The corresponding total intensities of the solutions scaled linearly with the prepared dilutions of the EGFP solutions.

## Rationale behind $r_{EGFP}$ as a probe for intracellular MMC

Fluorescence anisotropy requires exciting fluorophores with plane-polarized light, which selectively excites fluorophores aligned more parallel to the polarization plane of the excitation light. The resultant fluorescence emission is also polarized along the excitation plane, ensuing anisotropic intensities of the emitted light when observed through two orthogonally oriented polarizers. The normalized difference between the fluorescence intensities along the parallel ($I_\parallel$) and perpendicular ($I_\perp$) directions

is defined as fluorescence anisotropy: $r = \left(I_{\parallel} - I_{\perp}\right) / \left(I_{\parallel} + 2I_{\perp}\right)$ (*Lakowicz, 2006*; *Ghosh, 2012*). This anisotropy of fluorescence polarization is gradually lost when the fluorophores undergo rapid Brownian rotation in the excited state or due to other non-trivial causes, like homo-FRET (*Bojarski et al., 1991*; *Clayton et al., 2002*; *Tramier and Coppey-Moisan, 2008*) and light scattering (*Bigelow and Foster, 2004*). The extent of a fluorophore's rotation in the excited state determines the loss in anisotropy and depends on the solution viscosity ($\eta$), temperature ($T$), the fluorophore's size ($V$), and the fluorescence lifetime ($\tau$). The Perrin equation describes the measured fluorescence anisotropy as: $r = r_0 / \left(1 + \tau/\theta_C\right)$, where $\theta_C = \eta V / k_B T$ is the rotational correlation time, and $r_0$ is the intrinsic anisotropy in the absence of rotation. $r_0$ is determined by the intrinsic angle between the absorption and emission dipole moments within the fluorophore. Palpably, the value of $\tau/\theta_C$ determines the sensitivity of a fluorophore's $r$ to changes in $\eta$. As seen in comparatively large molecules like EGFP, the value of $\tau/\theta_C$ is <1, implying that an increase in $\theta_C$ (and thus solution $\eta$) has a negligible effect on the measured $r$ of EGFP (*Swaminathan et al., 1997*; *Novikov et al., 2017*). In comparison, for a smaller molecule like fluorescein, the value of $\tau/\theta_C$ is >1, meaning that increases in $\eta$ strongly affect the measured $r$ (*Devauges et al., 2012*). The $r$ of large fluorophores like EGFP (having long $\theta_C$) is still prone to be affected by the solution refractive index ($n$). This is because $n^2 \propto 1/\tau$ according to the Strickler-Berg relation (*Strickler and Berg, 1962*; *Tregidgo et al., 2008*). Thus, an increase in $n$ can also increase the measured $r$ because $n^2 \propto 1/\tau \propto r$. Importantly, as the effect of $n$ on $\tau$ is short range (*Suhling et al., 2002*), one can use $r$ to probe the local $n$ of the protoplasm, and, in turn, the local MMC.

### $r_{EGFP}$ measurement

The $r_{EGFP}$ measurement setup is described in *Figure 1—figure supplement 1B*. Cells seeded on glass-bottom Petri dishes were imaged with a ×40 (NA 0.75) or a ×10 (NA 0.45) air immersion objective using the Zeiss AxioObserver Z1 epifluorescence microscope. Light from a mercury arc lamp (HXP 150) was passed through a linear polarizer (Thorlabs) to create horizontally polarized light. The resulting polarized fluorescence signal from the cells passes through a polarizing beam splitter (DV2, Photometrics) to divide the emission light into parallel and perpendicular polarizations. The light is then collected by a CMOS camera (Hamamatsu Orca Flash 4.0 C13440), and the polarized fluorescence signal appears as an image having 2048×2048 pixels, with each half (1024×2048 pixels) representing the parallel and perpendicularly polarized emission, respectively. Due to misalignment in the optical path, the two halves don't completely overlap. To resolve the misalignment, fluorescent polystyrene microspheres of 200 nm diameter were dried on a glass coverslip and imaged in the same arrangement as $r_{EGFP}$ measurement, such that the images of the beads may serve as fiduciary markers to register the pixels in the two halves of the image. Using the Descriptor-based Registration plugin of Fiji (ImageJ) (*Schindelin et al., 2012*) and a custom Fiji Macro, the left half (perpendicular channel, $I_{\perp}$) and right half (parallel channel, $I_{\parallel}$) of the 2048×2048 image were registered to create the best possible overlap of the corresponding pixels in both channels. Thence, $r_{EGFP}$ was calculated for each pixel using the relation:

$$r_{EGFP} = \frac{I_{\parallel} - gI_{\perp}}{I_{\parallel} + 2gI_{\perp}}$$

where $g$ refers to the instrumental correction factor or $G$-factor, calculated for each pixel from images of 100 nM fluorescein solution. To correct for background fluorescence, a 2048×2048 pixel image of the phenol red-free DMEM, having no cells and illuminated by similar conditions as the experimental subjects, was subtracted from each 2048×2048 image. This process eliminated the background fluorescence of both the parallel and perpendicular channels in the correct ratio. The resultant $r_{EGFP}$ image was saved as a 32-bit TIFF file, thresholded based on intensity (15,000–50,000 count for 16-bit image), and further analyzed using a custom-written code in Fiji (ImageJ). Photobleaching to evaluate homo-FRET was performed at 100% lamp intensity for 30 s, and the same cells were imaged pre and post bleaching.

## Fluorescence lifetime imaging microscopy

In vitro fluorescence lifetime and time-resolved anisotropy decay measurements were done using the DeltaFlex system (Horiba) using four-sided transparent UV quartz cuvettes. FLIM was carried

out using a pico-second 470 nm laser (PicoQuant) and a ×60 water immersion objective (NA 1.2), and fluorescence lifetime data for individual pixels were fitted to mono-exponential decay using the SymPhoTime64 software. The resultant 32-bit TIFF image was analyzed in a similar way as in $r_{EGFP}$ measurements with Fiji (ImageJ) (*Schindelin et al., 2012*). Alternatively, the raw .BIN files of FLIM data were analyzed using custom MATLAB codes for fitting and phasor analysis. The phasor plot is a graphical way to display all the fluorescence lifetime data from a FLIM image in frequency space (*Digman et al., 2008*; *Ranjit et al., 2018*). Each phasor point represents a single fluorescence lifetime and its amplitude in the FLIM image without making any assumptions about the number of decay rates or the specific decay model, thus freeing the need for curve fitting. As a result, pixels having similar fluorescence lifetimes occur in the same spot in phasor maps and can be easily differentiated. The FLIM images were processed using a custom MATLAB code to create phasor maps based on user-defined regions of interest. Cells were seeded on glass-bottom 35 mm Petri dishes, and hypertonic stress was applied following the same protocol as in $r_{EGFP}$ experiments.

## Fluorescence recovery after photobleaching

Photobleaching and recovery were imaged with a 488 nm laser (Coherent OBIS 1185053) through the ×63 oil immersion objective of Zeiss AxioObserver Z1 using a home-built FRAP setup. Briefly, the source laser beam was split in a 90:10 ratio. The resultant beams were collimated using a lens system to be incident parallelly on the back focal plane of the microscope objective. The beams were aligned to illuminate the same spot (of 2 µm diameter) when imaged with the ×63 objective. The low-intensity beam was further dimmed using neutral density filters to minimize photobleaching and image the circular spot. The circular spot was continuously imaged at 50–100 frames per second to perform FRAP with only the low-intensity beam. After 70–100 frames, the high-intensity beam was exposed for 10 ms using a programmable shutter (Thorlabs, SC10) to achieve fast photobleaching. Imaging is continued for a total of 2000 frames, by which time the spot intensity becomes constant, indicating completion of recovery. The fluorophore's diffusion rate and mobile fractions are calculated by fitting the intensity recovery data from the spot with a custom-written MATLAB code, as explained in *Kang et al., 2010*. Before studying live cells, the FRAP setup was calibrated using a glycerol-water mix of known viscosity containing 100 nM fluorescein (data not shown). The FRAP in *Figure 8D* and *Figure 8—figure supplement 2B* was performed using the Zeiss LSM 780 laser scanning confocal system to bleach a larger spot (~10 µm diameter).

## Single-particle tracking

Fluorescent polystyrene beads of diameter 200 nm (Invitrogen, #F8888) were imaged with a ×63 oil immersion objective at 100 frames per second to capture the thermal motion. For in vitro measurement, beads were suspended in BSA solutions at previously indicated concentrations. The beads were ballistically injected with the Helios Gene Gun (Bio-Rad) delivery system for intracellular measurement. Cells were 'shot' with a pressure of 100 PSI at 3–4 cm from the Petri dish. The cells were then gently washed with serum-free media thrice to remove beads stuck on the plasma membrane or glass and incubated in phenol red-free DMEM at 37°C, 5% $CO_2$ for 2 hr to allow them to recuperate. The trajectories of the fluorescent beads were extracted using the Mosaic plugin (Particle Tracking 2D/3D) of ImageJ. The following relation was used for MSD computation of a bead with trajectory $(x_t, y_t)$: $MSD\left(\tau\right) = \left\langle \left(x_{t+\tau} - x_t\right)^2 + \left(y_{t+\tau} - y_t\right)^2 \right\rangle$, where $\tau$ is the lag time. MSD computation was performed using a custom-written MATLAB code.

## Cell volume measurement

Cells were imaged using the Zeiss LSM 780 laser scanning confocal system using a ×63 oil immersion objective. Z-stack images of 0.4 µm step size were acquired in AiryScan super-resolution mode to measure the whole cell volume. While AiryScan imaging improves the spatial *XY* resolution but not the *Z*-resolution, AiryScan processed images have comparatively lesser pixel noise, providing a uniform parameter for image thresholding. An appropriate intensity threshold was used to binarize the Z-stacks, and then the volume of the cells was calculated by counting the number of white pixels and multiplying the resultant with the voxel dimensions.

## Spatial heterogeneity estimation and height map generation

*Figure 3—figure supplement 1D* details the GDM creation using cell and nuclear boundary ROIs. For cell height map generation, the 'royal' LUT of ImageJ was modified to generate colors specific to height range of 0–12 µm. The starting color was black for the base, and the next color was assigned white to create maximum contrast so that individual cell boundaries could be identified. The Z-stacks used to measure cell volume were thresholded and the pixel values were changed to the voxel depth, then the sum of each Z-slice created the local height map, which was color-coded. *Video 2* was generated using ImageJ's '3D Project', and here, individual Z-slices were color-coded according to their vertical height using the same modified LUT.

## Cell extracts, immunoprecipitation, and immunoblotting

HeLa cells (~8 × 10⁶) were plated overnight in 10 cm dishes and treated with mannitol for 15 min, then lysed in 50 mM Tris-HCl (pH 7.4), 150 mM NaCl, 1% NP40, 0.1% SDS, 0.5% Na-deoxycholate supplemented with protease and phosphatase inhibitor cocktail (Sigma, #PPC1010) for 15 min on ice. The cell lysates were centrifuged at 15,000×$g$, 40°C for 20 min, and supernatants were collected. Protein concentration was determined by the Bradford assay, and the lysates were pre-cleared with 50 µL of protein A/G-PLUS agarose beads (Santa Cruz, CA, USA). About 3 mg of pre-cleared lysate was incubated overnight at 4°C with 10 µL of TNFR1/TNFRSF1A Rabbit pAb (ABclonal, A1540) and 50 µL of protein A/G-PLUS agarose beads (Santa Cruz, #sc-2003). The immune complexes were recovered by centrifugation, washed thrice with lysis buffer, and subjected to electrophoresis on 10% Tris-glycine gels. Proteins were then transferred to the PVDF membrane (Millipore), and non-specific binding sites were blocked by incubation in TBS containing 0.1% Tween-20 and 5% BSA. The membrane was probed with primary antibodies - anti-RIPK1/RIP rabbit mAb (ABclonal, #A19580) or anti-TNFR1/TNFRSF1A rabbit pAb (ABclonal, #A1540), in 1:1000 dilution at 4°C overnight, washed with TBS-T and subsequently incubated with secondary antibody (1:10,000 horseradish peroxidase-conjugated goat anti-rabbit IgG, Sigma) for 1 hr. Immunoblotting was done following standard chemiluminescence procedure, and densitometric analysis was performed using ImageJ. For TNFR1 knockdown, cells were incubated with TNFRSF1A DsiRNA or scrambled siRNA in the presence of FuGENE for 48 hr per the manufacturer's recommendations before evaluation by immunoblotting.

## Immunofluorescence imaging and quantification

HeLa or NIH/3T3 cells were plated on glass-bottom dishes and fixed with 4% PFA (paraformaldehyde) in 1× PBS for 15 min at room temperature. Cells to be assessed for hypertonic stress response were fixed with 4% PFA dissolved in mannitol-supplemented hypertonic PBS as per treatment to preserve macromolecular condensation. The cells were then permeabilized with 0.1% Triton X-100 in 1× PBS for 7–8 min, and blocking was performed with 5% BSA solution for an hour at room temperature. The cells were incubated with 1:250 anti-NF-kB p65 antibody (Abcam, #ab16502) at 4°C overnight. The cells were then gently washed three times with 1× PBS and incubated with Alexa Fluor 546-conjugated secondary antibody (1:200 dilution) for 1 hr at room temperature. The nuclei were counterstained with Hoechst (0.5 µg/mL). Imaging was performed using Zeiss AxioObserver Z1 (×63 oil immersion objective), and Z-stacks of randomly selected cell populations were obtained. The Z-plane of a cell having the largest nucleus area was considered for obtaining the nuclear fraction of p65. The total intensity values ('RawIntDen' in ImageJ) of p65 fluorescence were used to quantify the nucleus/whole-cell p65 fraction for individual cells. For the Alexa Fluor 546 phalloidin (Thermo Fisher, #A22283) staining in *Figure 3A*, live NIH/3T3-EGFP cells were fixed on the microscope stage with 4% PFA in 1× PBS for 15 min after imaging for $r_{EGFP}$ measurement, and then co-stained with Alexa Fluor 546 phalloidin and Hoechst.

## Protein mass per cell estimation with Bradford assay

NIH/3T3 cells were serum-starved for 24 hr and then 1×10⁶ cells were seeded on 60 mm dishes in complete medium after counting with a hemocytometer. Cells were allowed to spread overnight for a maximum of 12 hr, and all treatments (described in *Figures 5F-ii and 8A-iv*) were performed on the following morning, such that there are equal number of cells in each plate for every treatment. Post treatment, cells were immediately placed on ice and scraped with 200 µL RIPA lysis buffer. After 15 min of incubation on ice in the lysis buffer, the cell lysates were centrifuged at 15,000×$g$ for 10 min

at 4°C, and the supernatant was collected for protein density measurement using the standard Bradford assay protocol. The total protein density in 200 µL of solution allows the calculation of the total protein mass of $1 \times 10^6$ cells, and thus, protein mass in one cell. Using 8 M urea lysis buffer instead of RIPA buffer yielded no significant difference in the total protein mass content post the treatments indicated in *Figure 5F-ii*, thus confirming no loss of protein in the centrifuge precipitate.

## IRM and membrane tension estimation

An inverted microscope (Nikon, Tokyo, Japan) with adjustable field and aperture diaphragms, ×60 Plan Apo (NA 1.22, water immersion) with ×1.5 external magnification, 100 W mercury arc lamp, (546±12 nm) interference filter, 50:50 beam splitter, and CMOS (ORCA Flash 4.0 Hamamatsu, Japan) camera were used for IRM. Fast time-lapse images of cells were taken at 20 frames per second, and 2048 frames were captured. Membrane fluctuations are quantified for regions within ~100 nm of the coverslip and termed First Branch Regions (FBRs). Calibration, identification of FBRs, and quantification of fluctuation amplitude ($SD_{time}$) and tension were done as previously reported (*Biswas et al., 2017*).

## Statistical analysis

Technical replicates (*N*) of single-cell measurements within the same treatment group were combined to form a single group of biological replicates (*n*) for a given dataset. Normally distributed datasets were analyzed with ANOVA, while non-normal distributions were compared using the non-parametric Kruskal-Wallis ANOVA after alpha-correction by the Bonferroni method, followed by the Mann-Whitney test for every group pair. Differences between the population averages before and after treatment for same-cell measurements were assessed by the paired sample t-test, and for measurements in different cell groups, the unpaired t-test was used, assuming that a large enough sample size would follow the normal distribution. All statistical analyses and data plotting were performed using Origin 2019b.

## Acknowledgements

We are grateful to the CSS and TRC confocal facility of the Indian Association for the Cultivation of Science for providing the confocal microscopy and TCSPC system. We are especially grateful to Prof. Maria Vartiainen (University of Helsinki, Finland) for providing us with the GFP-GFP plasmid, Dr. Aprotim Mazumdar and Dr. PS Kesavan of TIFR Hyderabad for assistance with FLIM measurements, Mrs Debapriya Ghatak for assistance with confocal microscopy, Mr Subrata Das for assistance with TCSPC, and Dr. Sanjay Dutta (CSIR-IICB, Kolkata) and Dr. Prosenjit Sen (IACS, Kolkata) for providing us with essential cell lines. Finally, special thanks to Dr. Shrikant Kokate, Dr. Sarah Körber, Mr. Xiang Le Chua, and Prof. Pekka Lappalainen (University of Helsinki, Finland) for critical reading and assessment of the manuscript. Funding: DKS was supported by the SERB Core Research Grant, Department of Science and Technology, Ministry of Science and Technology (CRG/2022/005356), and the Department of Biotechnology, Ministry of Science and Technology (BT/PR6995/BRB/10/1140/2012). BS acknowledges support from the Wellcome Trust/DBT India Alliance fellowship (grant number IA/I/13/1/500885). PB, SJ, and RRB were supported by fellowships from the Indian Association for the Cultivation of Science. PR and DR were supported by a fellowship from the Council of Scientific and Industrial Research. JD and BC were supported by a fellowship from the University Grants Commission.

## Additional information

### Funding

| Funder | Grant reference number | Author |
| --- | --- | --- |
| Department of Science and Technology, Ministry of Science and Technology, India | CRG/2022/005356 | Deepak Kumar Sinha |

| Funder | Grant reference number | Author |
|---|---|---|
| Department of Biotechnology, Ministry of Science and Technology, India | BT/PR6995/BRB/10/1140/2012 | Deepak Kumar Sinha |
| Wellcome Trust DBt India Alliance | IA/I/13/1/500885 | Bidisha Sinha |
| Indian Association for the Cultivation of Science | | Parijat Biswas Subhamoy Jana Ridita Ray Basunia |
| Council of Scientific and Industrial Research, India | | Priyanka Roy Dipanjan Ray |
| University Grants Commission | | Jibitesh Das Bipasa Chaudhuri |

The funders had no role in study design, data collection and interpretation, or the decision to submit the work for publication.

### Author contributions

Parijat Biswas, Conceptualization, Data curation, Software, Formal analysis, Validation, Investigation, Visualization, Methodology, Writing – original draft, Writing – review and editing, Wrote the paper, performed and analyzed the experimental data of r_EGFP, FRAP, in-vitro single particle tracking, cell volume measurements, and wrote all the codes for image analysis; Priyanka Roy, Data curation, Formal analysis, Validation, Investigation, Visualization, Methodology, Writing – original draft, Performed the immunoprecipitation, immunoblotting, and immunostaining experiments, and edited the manuscript; Subhamoy Jana, Data curation, Software, Formal analysis, Validation, Investigation, Methodology, Did the calibrations and experiments on the TR-FA setup, and immunoblotting for TNFR1 knock-down, and edited the manuscript; Dipanjan Ray, Data curation, Formal analysis, Validation, Investigation, Methodology, Did the intracellular single particle tracking experiments and assisted in manuscript preparation; Jibitesh Das, Data curation, Formal analysis, Validation, Investigation, Visualization, Methodology, Did the membrane tension measurements using IRM and analyzed the data; Bipasa Chaudhuri, Data curation, Formal analysis, Validation, Investigation, Methodology, Did cell volume measurements with NFkB inhibition, critically evaluated and edited the manuscript; Ridita Ray Basunia, Methodology, Standardized and performed the EGFP purification, and documented the methodology; Bidisha Sinha, Conceptualization, Resources, Software, Formal analysis, Supervision, Funding acquisition, Validation, Investigation, Visualization, Methodology, Writing – original draft, Project administration; Deepak Kumar Sinha, Conceptualization, Resources, Data curation, Supervision, Funding acquisition, Validation, Investigation, Visualization, Methodology, Writing – original draft, Project administration, Writing – review and editing

### Author ORCIDs

Parijat Biswas ⓘ https://orcid.org/0000-0003-2235-2384
Priyanka Roy ⓘ https://orcid.org/0000-0002-4197-8211
Dipanjan Ray ⓘ https://orcid.org/0000-0002-2819-3284
Jibitesh Das ⓘ https://orcid.org/0000-0003-3611-5902
Bipasa Chaudhuri ⓘ https://orcid.org/0009-0001-3652-2922
Bidisha Sinha ⓘ https://orcid.org/0000-0001-6449-8205
Deepak Kumar Sinha ⓘ https://orcid.org/0000-0002-8303-5035

### Decision letter and Author response

Decision letter https://doi.org/10.7554/eLife.92719.sa1
Author response https://doi.org/10.7554/eLife.92719.sa2

## Additional files

### Supplementary files
• MDAR checklist

## Data availability

No new datasets were generated by this manuscript. The codes used in the manuscript for analyzing images, FRAP data, and single particle tracking are freely available online in GitHub: https://github.com/bparijat/ImageJ-Macros__MatLab-codes/tree/main/MMC-TNFR1_in_CellVolumeControl (copy archived at *Biswas, 2024*). Descriptions of the codes are provided in a README file along with the codes. Any queries regarding operational details of the codes can be forwarded to the owner of the GitHub repository via direct messaging. Source data for western blotting, immunofluorescence images, and histograms are provided with figures, and further queries can be forwarded to the authors.

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
