## [Editor Report]

This study provides a useful real time technique utilising fluorescence emission anisotropy of cytoplasmically expressed mEGFP to measure macromolecular crowding in living cells. The authors use this technique to provide solid evidence for the role of macromolecular crowding in cell volume control in mammalian cells under different conditions and perturbations. This method is likely to be of general interest to cell biologists and biophysicists since macromolecular crowding has broad implications for cell biological phenomena such as in osmotic stress response, cell cycle, cell death, and phase separation to cite only a few.

---

## [Decision Letter]

**Decision letter after peer review:**

[Editors’ note: the authors submitted for reconsideration following the decision after peer review. What follows is the decision letter after the first round of review.]

Thank you for submitting the paper "Probing the role of macromolecular crowding in cell volume regulation using fluorescence anisotropy" for consideration by *eLife*. Your article has been reviewed by 3 peer reviewers, and the evaluation has been overseen by a Reviewing Editor and a Senior Editor. The reviewers have opted to remain anonymous.

Comments to the Authors:

We are sorry to say that, after consultation with the reviewers, we have decided that this work will not be considered further for publication by *eLife*.

Specifically, all the reviewers felt that there were serious concerns regarding both technical and methodological aspects of the work that seriously limited the conclusions about the method as well as the relationship between MMC and cell volume regulation, that could be drawn from the data. In addition, some of the experiments and their conclusions are not consistent, as detailed in the extensive reviews from the reviewers provided. Any revision would require a lot more time and extensive additional experimentation whose outcomes are likely uncertain, and, inconsistent with the policies of a revised manuscript at *eLife*. While some of these weaknesses have been highlighted in the public review, detailed reviewer reports are also provided.

Consensus Public Review:

In this manuscript, the authors attempt to establish steady-state fluorescence anisotropy of eGFP as a sensor for macromolecular crowding (MMC). This is a valuable tool if it can be fully validated. Traditionally this is a method used to assay the fractions of bound versus unbound in protein-protein and protein-ligand interactions due to an increase in anisotropy as a consequence of an increase in molecular mass. It has also been used to assess molecular proximity since this causes a decrease in anisotropy due to homo-FRET; it has not been used to estimate crowding. The results of in vitro experiments suggest that this method is practically insensitive to pH, ion concentration, small molecules, and probe (i.e., EGFP) concentration, and thus robust enough to evaluate the effects of MMC in cells. The results of in vivo experiments "probably" suggest many interesting findings of the associations between intracellular MMC and cell volume.

Strengths:

Given that MMC has been usually studied using in vitro and in silico experiments thus far, this quantitative method using fluorescence anisotropy has a great potential to bridge the gaps in our understanding of MMC between in vitro/theory and physiological conditions. In addition, the authors applied this method to investigate the relationship between intracellular MMC and cell volume regulation, which helps readers to understand the application utility of this method in cell experiments. Moreover, it should be noted that their idea about applying this method to the context of cell volume regulation is great because one of the challenges in this context is distinguishing the actual changes in intracellular events (i.e., MMC in this case) from probe-derived changes (i.e., EGFP concentration in this case) for any detected differences of a probe measure, but this challenge could be resolved in this method.

Weaknesses:

Unfortunately, the current manuscript is immature as a scientific article, because of lacking sufficient information for readers to understand what their results indicate objectively. The data is not presented with adequate care with regards to a description of methodology and statistics. The method needs to be better benchmarked before conclusions can be reached about the meaning of the observed changes in anisotropy, especially in the comparison between the in vitro and in vivo changes observed. As this is the only method used by the authors to estimate crowding many of the conclusions depend on the validation of the methodology. The authors also fail to put their observation in the context of the available literature on MMC cell volume regulation and cell volume changes, making it harder for the reader to evaluate. This study contains a critical concern in the overall logic.

To evaluate the relationships between intracellular MMC and cell volume "regulation" by utilizing their method, scientists have to recognize that the detected change of fluorescence anisotropy in cells under osmotic perturbation reflects at least four-level of changes: (1) change in a potential molecule specifically affecting excitation/emission process of probe itself, (2) change in the absolute amount of intracellular macromolecules, (3) passive change in cell volume, which is caused by osmotic water flux, and (4) regulatory change in cell volume, which is positively induced by cells using intracellular energy like ATP and chemical potential (i.e., regulation, such as regulatory volume increase (RVI) and regulatory volume decrease (RVD) under osmotic stress).

In this study, the authors address points (1) and (2) through a series of in vitro experiments (Figure 1), however, the data do not provide a clear understanding of the origin of the changes in GFP anisotropy in cells. For example, GFP shows a change in a lifetime in cells of 0.1 ns (from 2.5 ns in isotonic conditions to 2.4 ns) when shifted to hypertonic conditions, and correspondingly, the anisotropy changes by 0.032 from 0.172 to 0.204 (see Figure 2B). This is inconsistent with a 1 ns decrease in GFP-lifetime in solution in vitro (Figure 1C) that leads to a difference of 0.016 in anisotropy (Figure 1A). This discrepancy indicates that there may be other processes happening in the cell apart from changes in GFP lifetime (related to refractive index) that are dictating corresponding changes in anisotropy. Therefore the authors cannot simply consider that the detected change of fluorescence anisotropy in cells relates to the change in intracellular MMC. This study also did not distinguish points (3) and (4) at all, implying that the authors can infer the associations only between intracellular MMC and cell volume, but not between intracellular MMC and cell volume "regulation".

The authors seemed to interpret that the detected changes in cell volume/area correspond to RVI or RVD. At least, they should evaluate cell volume regulation using well-known inhibitors, which have been characterized in the field of electrophysiology (e.g., flufenamate for RVI, NPPB, or DCPIB for RVD; see such as Wehner, F. et al. FEBS Lett. 2003 and Ponce, A. et al. Cell Physiol. Biochem. 2012), to support their interpretations. In addition, they should present a quantitative index for RVI/RVD, because several figures clearly show a difference in the enforced cell shrinkage/swelling (i.e., the difference between isotonic condition and the first time point after osmotic treatment). One of the commonly used indexes for RVI/RVD is the recovery rate from the first time point after osmotic treatment. Moreover (possibly related to the aforementioned major point 1), the authors' usage of RVI and RVD (e.g., in line 307) raises a concern about their understanding of the terms of RVI and RVD. In general, RVI and RVD indicate the cell volume recovery followed by hypertonic stress and hypotonic stress, respectively. And, the cell volume change free from osmotic stress is not referred to as RVI or RVD. For instance, in the case of cell death, the accompanying cell volume change is differently termed as apoptosis volume decrease (AVD) even though the same effector molecule is involved in RVD and AVD (cf. Okada, Y. et al. J. Physiol. 2001). Many aspects of the paper about the relationships between intracellular MMC and cell volume "regulation" seem NOT to be logically supported by their results.

Finally, in terms of data presentation, the figures and figure legends are far from complete regarding the reporting of data. For example, the figure legends do not report any sample size (number of cells, number of repetitions, etc…).

Judging from the plot in Figure 3 (because that information is missing from the methods) the FRAP experiment was performed at 2 frames per second. The resulting recovery curves do not have enough points to properly recover the exponential phase of the recovery. In fact, judging by the plot in Figure 3B, the authors have only 1 data point before the plateau. How can a rate be accurately recovered from these data? A similar concern is raised in Figure 5E. Multiple plots are poorly described. For example, in Figures 2B and C, there is nothing on the figure or in the legend indicating which curves are the area and which are the anisotropy values. In Figure 3 D there is no information about the statistical test used. In Figure 5, the scatter plots have what we may assume are linear regressions, yet nothing is said about how the lines were obtained and if statistical tests were used.

*Reviewer #1:*

In this manuscript, the authors attempt to establish steady-state fluorescence anisotropy of eGFP as a sensor for macromolecular crowding. This is a valuable tool if it can be fully validated. They then explore various means of perturbation of MMC as a way to determine if it acts as a homeostat for cell volume regulation. They conclude that MMC is unlikely to be a control parameter in cell volume regulation since changes in MMC do not influence regulated volume changes, whereas cell volume changes do affect the MMC, in a predictable manner. This is potentially overturning a long-held view regarding the role of MMC in cell volume regulation. They suggest that volume regulation is affected by the homeostatic control of plasma membrane tension. However, there are several issues both technical and conceptual that need to be addressed before the conclusions put forward are convincing:

1. Technical: The authors first test the dependence of GFP steady-state anisotropy on various environmental factors such as pH, viscosity, and concentration. However, there are several points that need to be clarified before the use of emission anisotropy values of EGFP may be reliably taken as a measure of MMC in the cell, as reported.

a) The authors show in Figure 1E that the anisotropy of GFP is independent of concentration, by showing GFP concentrations ranging from 1-9mM (milliMolar). This range of concentrations should result in significant Homo-FRET (FRET between GFP-GFP molecules). Concentration-dependent depolarization of the GFP emission due to homo-FRET has been shown in many studies (for example for a detailed analysis see-https://doi.org/10.1016/S0006-3495(02)73932-5). Ranging from a few μm to 0.6mM of purified GFP, a clear concentration dependence of GFP on anisotropy is seen. A similar study with VFP (DOI: 10.1042/bst0311020) also finds the same effect. It is puzzling to see this apparent independence of eGFP anisotropy reported by the authors. It is likely that eGFP has already entered a state of saturating FRET at these concentrations. It would be very useful to have a larger range of concentrations sampled here, starting from as little as 250nM going into the milliMolar regime.

b) Another discrepancy is the value of anisotropy. The authors do not mention the concentration of GFP used to perform the experiments in Figure1 (A, B, D, F). The methods only specify that they used a 1:100 dilution of the purified GFP. The concentrations may help to clarify the disparities the authors see in the starting values of GFP anisotropy between Figure1(A, B, E, F) and Figure1(D). The anisotropy in Figure1 (A, B, E, F) starts consistently at a value of 0.20, and only in 1D, the value is at 0.24. An anisotropy difference of 0.04 between different experiments is a really large change (bigger than the largest differences observed with cells or with BSA in the paper) and the reasons for such variability need to be explicitly addressed.

c) The differences in eGFP lifetime shown here in Figure1C have been reported before (in the papers of Suhling et al. in 2002, cited also by the authors) and it is understood that these differences are due to a change in the refractive index of the medium. Suhling et al. also show that the lifetime of the fluorophore is agnostic to the nature of the "macromolecule" itself and that various concentrations of glycerol, PEG, NaCl, glucose, or fructose only affect the lifetime by changing the refractive index of the solution. So it is confusing in Figure 1C to plot the dependence of lifetime on the concentrations of BSA and HiSep as two separate graphs when we know that both can collapse on the same curve if the refractive indices of these solutions are plotted on the x-axis instead of the concentrations. Hence it will be good to rework the plots in Figure1 replacing the concentration axes with refractive index instead, Especially in Figure 1C it will be useful to see if the data match the predictions of the Strickler-Berg equation as shown earlier by Suhling.

d) The authors report a decrease in the lifetime of GFP with increasing refractive indices, which is then being indirectly read out by a change in its steady state anisotropy. As the authors also mention, the Perrin equation governs the relationship between anisotropy and lifetime. This needs to be verified for the system in question with an Anisotropy vs Fluorescence Lifetime graph. Between Figures 1A, 1C, and S1B, the authors already have the data to do so. An agreement with the predictions of the Perrin equation will be useful for the authors' efforts to validate the use of anisotropy as a readout of refractive index-mediated fluorescence lifetime changes.

e) It must be mentioned that in a series of papers by the group of Paras Prasad (doi: 10.1021/cb300065w., https://doi.org/10.1038/s41467-019-08354-3, doi: 10.1088/2050-6120/ab8571), they establish fluorescence lifetime of eGFP as a sensor for the local refractive index. They also make the claim that the refractive index map of a cell could serve as a measure of local macromolecular concentration. In the supplementary of the Nature Comms paper linked above, they clearly show the effect of BSA concentration in increasing the refractive index of the buffer and show a decrease in eGFP lifetime in accordance with the relationship predicted by the Strickler-Berg equation. The authors of this submission should cite the work mentioned above and situate the relevance of the current study in the light of what is already known.

2. Experiments with changes in reGFP in cells: These experiments try to understand the effect of changing the crowding state of the cell (by osmotic shock or cell spreading) and try to connect it to the steady state anisotropy, and subsequently cell volume. However, a few confounding factors need to be addressed to allow a meaningful interpretation of the results observed.

a) The authors write in line 99 – "The observed increase of *r*EGFP due to increased cytoplasmic MMC is caused primarily by the decrement of its fluorescence lifetime". Figure 2D shows a lifetime change in cells of 0.1 ns (from 2.5ns in isotonic conditions to 2.4ns) in hypertonic and correspondingly, the anisotropy changes by 0.032 units from 0.172 to 0.204(Figure2B). This is inconsistent with a 1 ns decrease in a lifetime in Figure 1C which leads to a difference of 0.016 units in Figure 1A. This discrepancy indicates that there may be other processes happening in the cell apart from changes in GFP lifetime (or refractive index) that are dictating corresponding changes in anisotropy.

b) An important aspect of the data acquired from the cell experiments is the GFP concentration itself. The GFP concentration which can be read out by the mean GFP intensity per voxel should ideally respond to the changes in cell volume perturbation. GFP concentration will also be useful to understand the effect of various treatments on the cell area and volume. One way to represent this information is to plot Anisotropy as a function of Total Intensity. The GFP concentration in cells should also ideally correspond to the range of concentrations sampled in Figure 1 to decouple the concentration dependence of GFP anisotropy (as indicated above).

c) Using the lifetime of GFP as a censor for MMC has been done before. However steady-state anisotropy is simpler and faster to measure on a conventional microscope than doing FLIM. If the above concerns are addressed then this may prove to be a more accessible way to map local refractive indices in cells. All the other results of the paper can only be interpreted once there is a clearer understanding of what is driving the anisotropy differences seen in cells.

3. MMC, cell volume homeostasis, and plasma membrane tension. This part of the research – both underlying the relationship of MMC and cell volume regulation may only be understood objectively if the above technical issues are completely clarified. At this point, it is only worth noting that changes in MMC as reported by changes in rEGFP do not correlate with what is expected of a sensor for cell volume homeostasis.

The experiments with changes in plasma membrane tension, cell spreading, and cell volume regulation, including the technique of fluctuation spectroscopy to infer plasma membrane tension changes, are interesting but too preliminary to provide an alternative homeostat sensor for cell volume regulation.

*Reviewer #2:*

The authors present a method to quantitatively investigate the effects of macromolecular crowding (MMC) in vitro and in cells, using fluorescence anisotropy of EGFP. Their results of in vitro experiments "probably" suggest that this method is practically insensitive to pH, ion concentration, small molecules, and probe (i.e., EGFP) concentration, and thus robust enough to evaluate the effects of MMC in cells. Their results of in-cell experiments "probably" suggest many interesting findings of the associations between intracellular MMC and cell volume. However, the current manuscript is immature as a scientific article, because of lacking sufficient information for readers to understand what their results indicate objectively. Moreover, even if a reader makes the maximum effort to guess what the authors really wanted to describe, their statements about the relationships between intracellular MMC and cell volume "regulation" seem NOT logically supported by their results.

Strengths:

Given that MMC has been usually studied using in vitro and in silico experiments thus far, this quantitative method using fluorescence anisotropy has a great potential to bridge the gaps in our understanding of MMC between in vitro/theory and physiological conditions. In addition, the authors applied this method to investigate the relationship between intracellular MMC and cell volume regulation, which helps readers to understand the application utility of this method in cell experiments. Moreover, it should be noted that their idea about applying this method to the context of cell volume regulation is great because one of the challenges in this context is distinguishing the actual changes in intracellular events (i.e., MMC in this case) from probe-derived changes (i.e., EGFP concentration in this case) for any detected differences of a probe measure, but this challenge is resolved in this method.

Weaknesses:

Unfortunately, the current manuscript is immature as a scientific article, because of lacking sufficient information for readers to understand what their results indicate objectively. Critical points include: (i) there is no description of what the variance ranges in sentences (e.g., 4.2 {plus minus} 1.1 cP in line 163) and the error bars in all figures indicate (i.e., standard deviation (SD), standard error (SE), confidence interval (CI), or something), (ii) some figure legends (e.g., as to Figure 2B) have insufficient information about the annotations (even within the main text), (iii) there are methods whose details are not described in the Method section (e.g., Bradford assay used in Figure 2H and 4B, cell cycle synchronization used in Figure 5 and S5), and (iv) most sentences are ambiguous for readers to understand whether their statements are objectively/logically deduced by their results or subjectively described as their interpretations. All these points are essential in science (e.g., points (i)-(iii) are important from the perspective of reproducibility). Of note for point (iv), some journals allow authors to describe "Ideas and Speculation" in the Discussion section, but this doesn't mean that the authors can mix up the objective findings with their subjective interpretations/ideas/speculations.

Even after a reviewer made the maximum effort to try to understand what the authors really wanted to describe, some statements seemed contradictory to what their results actually indicate. For instance, in lines 97-99, the authors probably wanted to describe that the black rectangle markers in Figure 2C corresponded to the right axis and indicated the trajectory of the cell area under hypotonic stress (this is unclear due to point (ii)) and that the cell area was decreased during hypotonic stress based on the comparison of these mean (or certain summary statistic) values at 120 min vs. at the first time point after hypotonic treatment. However, even if assuming the error bars indicated SEMs (this is unclear due to point (i)), no scientist could conclude that there was a difference in these values based on Figure 2C, because the error bars clearly overlapped between these time points and a simple t-test, which was only the information described in the Method section, would not exhibit statistical significance for them (regardless using a statistical model with certain covariates).

Apart from these inconsistencies between their statements and results, this study contains a critical concern in the overall logic. To evaluate the relationships between intracellular MMC and cell volume "regulation" by utilizing their method, scientists have to recognize that the detected change of fluorescence anisotropy in cells under osmotic perturbation reflects at least four-level of changes: (1) change in a potential molecule specifically affecting excitation/emission process of probe itself, (2) change in the absolute amount of intracellular macromolecules, (3) passive change in cell volume, which is caused by osmotic water flux, and (4) regulatory change in cell volume, which is positively induced by cells using intracellular energy like ATP and chemical potential (i.e., regulation, such as RVI and RVD under osmotic stress). In this study, points (1) and (2) seemed to be addressed by the series of in vitro experiments, and scientists can consider that the detected change of fluorescence anisotropy in cells basically indicates the change in intracellular MMC. However, this study did not distinguish points (3) and (4) at all. Therefore, scientists can infer the associations only between intracellular MMC and cell volume, but not between intracellular MMC and cell volume "regulation".

Furthermore, there are scientific concerns in this study. Other than osmotic perturbations, the authors utilized alternative methods to operate intracellular MMC. However, their assumptions would be regarded as too bold assumptions in modern cell biology. For instance, they utilized Heclin to increase intracellular MMC under the assumption that Heclin reduces protein degradation via HECT inhibition (lines 130-131). However, HECT is just one family of the E3 ligases in a cell, and the other ubiquitin-proteasome system is still active under the Heclin treatment. Although Figure 2H demonstrated the significant (slight) effect of Heclin on the overall protein amount, this result can imply only the association and cannot deny the possibility that HECT affected the EGFP fluorescence anisotropy independently through its effect on the global protein degradation. In fact, if agreeing with their interpretations that the detected signals in Figure S4 indicated "aggregation", Figure S4 exhibited that the Heclin treatment affected the "aggregation" more strongly than hypertonic treatment, while Figure 2H suggested the weaker effect of Heclin treatment on the fluorescence anisotropy, rather in line with the above concerning possibility. Likewise, the authors utilized heat shock to decrease intracellular MMC under the assumption that heat shock promotes global protein degradation (lines 133-134). However, the Perrin equation (line 65) indicates that fluorescence anisotropy is negatively related to temperature, and it can be assumed that heat shock resulted in the decrease in the EGFP fluorescence anisotropy independently through its effect on the protein degradation.

Recommendations for the authors:

1. Although I have great interest in the authors' results, I strongly recommend that the authors should drastically revise the current manuscript to be understood by standard scientists. I mention concrete examples in the below minor points, but I cannot cover all points within this comment. If the author who mainly prepared the manuscript is unfamiliar with scientific writing, I recommend he/she first learns the general manner of scientific writing.

2. In this manuscript, Figure 1 is important to support that the EGFP fluorescence anisotropy indicates the effect of MMC. Given that the method itself is one of the conceptual advances, the authors should also evaluate the relationship between EGFP fluorescence anisotropy and MMC using other crowding agents as well as BSA (e.g., Ficoll, polyethylene glycol). The expected results would be comparable to the results of BSA.

3. In this study, the authors mainly used dextrose (i.e., D-glucose) as a means to prepare a hypertonic medium. Although they also demonstrated mannitol-based hypertonic treatment in Figure S2F, the main usage of glucose-based hypertonic treatment would raise eyebrows of most scientists studying cell volume regulation. It's primarily because the glucose concentration is approximately from 5 mM to 50 mM in most cell culture media, and it is not difficult for scientists to assume the potential effects of extremely high glucose on intracellular signaling, for example, via glucose transporters (Note that glucose transporters were reported to be involved in cell volume regulation too). Therefore, the authors should perform the additional experiments using mannitol-based hypertonic treatment, at least, in Figure 2G to support their interpretations. Related to this point about osmotic treatments and the aforementioned major point 1, the current Method section does not include information, but the authors may need to use "mM" instead of "mOsm" in this manuscript if they prepared hypertonic solutions based on the calculation of the osmolyte amount, not based on the direct measurement of osmolality using osmometers. Additionally, whether they maintained CO2 level after the replacement with hypertonic solution or not should be clearly described because it is known that bicarbonate (HCO3) is critical for RVI (in contrast to RVD) due to the function of an RVI activator, NHE (sodium-hydrogen exchanger), which would affect the interpretation about cell line comparison (Figure 2G).

4. In the current manuscript, the authors seemed to interpret that the detected changes in cell volume/area correspond to RVI or RVD. As described in the public review comment, this logical jump would not be acceptable. At least, they should evaluate cell volume regulation using well-known inhibitors, which have been characterized in the field of electrophysiology (e.g., flufenamate for RVI, NPPB or DCPIB for RVD; see such as Wehner, F. et al. FEBS Lett. 2003 and Ponce, A. et al. Cell Physiol. Biochem. 2012), to support their interpretations. In addition, they should present a quantitative index for RVI/RVD, because several figures clearly show a difference in the enforced cell shrinkage/swelling (i.e., the difference between isotonic condition and the first time point after osmotic treatment). One of the commonly used indexes for RVI/RVD is the recovery rate from the first time point after osmotic treatment. Moreover (possibly related to the aforementioned major point 1), the authors' usage of RVI and RVD (e.g., in line 307) raises a concern about their understanding of the terms of RVI and RVD. In general, RVI and RVD indicate the cell volume recovery followed by hypertonic stress and hypotonic stress, respectively. And, the cell volume change free from osmotic stress is not referred to as RVI or RVD. For instance, in the case of cell death, the accompanying cell volume change is differently termed as apoptosis volume decrease (AVD) even though the same effector molecule is involved in RVD and AVD (cf. Okada, Y. et al. J. Physiol. 2001).

5. In addition to osmotic perturbation, the authors utilized alternative methods to operate intracellular MMC. However, the usage of Heclin would raise many concerns, as described in the public review comment. Better experimental design would be using a proteasome inhibitor (e.g., MG132, bortezomib) instead of Heclin and adding another sample group under both cycloheximide and proteasome inhibitor treatment. As to heat shock, in addition to the potential concern described in the public review comment, the authors should deny/mention/discuss the possibility that the drastic decrease in protein mass by heat shock (Figure 2H) was caused not by the decrease in intracellular MMC but by the experimental procedures (e.g., the movement of aggregated proteins into an insoluble fraction) if they lysed cells according to the standard Bradford assay (line 137).

6. In the Result section of Figure 3, the authors described region-dependent findings from representative images. However, they should describe them with quantified results. In particular, given the association between fluorescence anisotropy and cell area (e.g., Figure 5A), there is a concern that their observations (e.g., Figure 3A) are just led by not region-specific feature but cell size-dependent heterogeneity (e.g., the stage of the cell cycle). Hence, the quantified index should be also compared under the same cell size. Moreover, the quantified evaluation would potentially affect their speculation (lines 191-193); their speculation may be a possible explanation, but the above concern (i.e., cell size-derived result) seems more possible because Figure 3D shows a significant "decrease" in nocodazole from control.

7. In lines 218-221, the authors described they utilized paraformaldehyde to increase the effective size of MMC via fixation. First of all, related to the aforementioned major point 1, this assumption sentence must be supported by some results or citations. However, their assumption is likely wrong, because 4% paraformaldehyde is an extremely high hypertonic solution (approximately 1,500 mOsm). Although formaldehyde is a membrane-permeable molecule, they need to confirm that the effect of formaldehyde on cell volume is neglectable for supporting their assumption (as my "subjective" remark, a cell after 15-min formaldehyde fixation tends to be more flatter than a live cell). Moreover, even under their assumption, EGFP is also cross-linked with intracellular biomolecules. Although it is common sense that EGFP keeps the ability to be excited and emit fluorescence even after fixation, they should check that the EGFP fluorescence anisotropy is not affected by the cross-link of EGFP; otherwise, they cannot compare the anisotropy between live cells and fixed cells.

8. In Figure 4D and S4, the authors interpreted that the results indicated "protein aggregation", but they must clearly describe the judgement criteria and reason(s). Although the current manuscript has insufficient information, one of them may be based on the observation of the local increase in fluorescence intensity. If so, however, it is still unclear why they can deny the possibility that the intensity increase just corresponds to the local increase in "free" (i.e., not aggregated) fluorophore concentration. The same thing is applied to "DNA condensation" (line 235).

9. In the Result section about Figure 5 and S5, the authors described "cell spreading", but what this terminology exactly indicates is unclear. Given that they mentioned about cell cycle in lines 245-246, this term may indicate the cells in G2 phase. However, no method for cycle synchronization is described in the Method section.

10. Based on the lines 251-253, 258-259, and others, the authors seem to think that a "sensor" must be regulated to be active after its functioning. However, this thought is not logical at all. Although this kind of feedback mechanism may be good for cellular homeostasis, there is no logical necessity for the "sensor" to be managed even after its functioning (i.e., sensing). In other words, the sensor molecule is not necessarily the same as the effector molecule. Additionally, related to "sensor", the authors should clearly describe the definition of "sensor" terminology, for example, in the Introduction section. Based on lines 292-293, they seem to consider their own definition for "sensor", whose candidate can include osmolyte. This is not so general; e.g., in the case of RVD, VRAC (LRRC8s) is said as a sensing molecule, and small osmolyte (e.g., chloride ion, glutamate) is a kind of the second messenger after VRAC activation (i.e., sensing). Likewise, based on lines 284-285, the authors seem to categorize only two modes for "sensor", while a famous review in this field (Hoffman, E. K. et al. Physiol. Rev. 2009, which is cited even in this manuscript) categorized three modes for the sensing principle. Of course, scientists can define their own terminology, but they must clearly describe it to avoid readers' confusion with the previous papers.

*Reviewer #3:*

Biswas et al., are interested in measuring the level of macromolecular crowding in mammalian cells. Macromolecular crowding has been shown to have a broad impact on various cellular functions, such as enzymatic reaction rates, cytoskeleton dynamics, and phase transition to name a few. Despite these observations, there are a lot of unanswered fundamental questions regarding macromolecular crowding. For example, it is unknown if cells can sense the level of crowding in the cytoplasm or what mechanisms they could use to regulate the crowding level. These are the questions the authors are trying to answer. These questions are of general interest to cell biologists and answering them would represent an advance in the fields of cell biology and biophysics. The main result from the study is that cells lack a homeostasis mechanism for macromolecular crowding however given the technical concerns it is unclear if this conclusion is supported by the data.

Strengths:

- The paper's ambitions are built on unknown from the literature and because macromolecular crowding is such a general concept with broad implications the potential conclusions from the paper are of interest to a large audience.

- The authors assess the universality of their observation by comparing multiple mammalian cell lines. This is an interesting approach as every cell line has a potentially different level of molecular crowding as well as different correction mechanisms.

- The authors try to test their hypothesis by using various methods to affect intracellular crowding levels.

Weaknesses:

- The authors use steady-state fluorescence anisotropy measurement to evaluate the level of macromolecular crowding in cells. Although it is a method used to assay the fractions of bounds and unbound protein it is not traditionally used to estimate crowding. The method needs to be benchmarked before conclusions can be reached about the meaning of the observed changes in anisotropy. As this is the only method used by the authors to estimate crowding level many of the conclusions are pending validation of the method.

- The figures and especially the figure legends suffer from a lack of clarity. In some of them, the reader is left to assume what some of the plots describe.

- The authors fail to put their observation in the context of the available literature making it harder for the reader to evaluate.

Recommendations for the authors:

I have a lot of doubts about the anisotropy method used by the authors.

1) I am not aware of any other paper using steady-state fluorescence anisotropy to measure crowding in cells and the authors make it sound like they are trying to establish the method for that usage. The method is usually used to evaluate binding affinity, etc. Can the authors provide a rationale as to how and why steady-state anisotropy should be reporting on crowding?

2) In figure 1 A, the BSA calibration curve shows an increase in anisotropy of around 0.02 from 0 to 5 mM of BSA. How should a change in anisotropy of 0.02 be interpreted? A solution of 5mM BSA should be very crowded, yet the readout only changes by 0.02 from no crowding to 5mM. Perturbations on cells should change crowding by less than that, meaning that the readout of these perturbations would be less than 0.02, probably even less than 0.01. Can such a small change be reliably detected? What is the error in the measurement? How does it compare to biological noise?

3) In figure 1 the authors use in vitro measurement to try and calibrate the method. In figure 1 A the authors report that anisotropy increases with increasing BSA concentrations in the range from 0 to 5 mM of BSA. In figure 1 E the authors report that anisotropy does not change when eGFP concentration is increased from 0 to 9 mM. Yet, BSA and eGFP are both proteins and similar(ish) in size. So, can the authors explain why eGFP anisotropy is sensitive to BSA concentration but not to its own concentration? Increasing eGFP concentration from 0 to 9 mM should increase the crowding of the solution and be picked up by the method.

4) Do the authors know the concentration of eGFP proteins in the cells they are imaging? Steady-state anisotropy could be sensitive to homo-FRET. Can the authors rule out that this contributes to what they are measuring?

5) Images from figure 2 A, figure 3 A, C and D, figure 4 C, as well as Sup. Figure C and G, Sup. Figure 3 B and Sup. Figure 5 A, B, C shows that the anisotropy signal reported scale with cell thickness. The signal clearly appears higher around the center of the cell, where the cell is the thickest, and lower at the edge. Can the authors explain why this striking correlation seems to be present in all the images? To be useful the method needs to be insensitive to cell thickness. Can the author provide data showing that this is the case?

Other experiments also have problems.

6) Mammalian cell volume is notably hard to measure. Even if the author used a super-resolution microscope this kind of super-resolution mostly increases the XY resolution not much the resolution in the Z-direction and the authors seem to have used a simple binarization method. Can the authors provide a comparison to measurements by others and/or use another method? A comparison of the data in figure 5 on spreading area vs volume with the data from others, such as Venkova et al. 2022, seems necessary. Especially because some of the conclusions drawn by the authors disagree with previous observations.

7) Judging from the plot (because that information is missing from the methods) the FRAP experiment was performed at 2 frames per second. The resulting recovery curves do not have enough points to properly recover the exponential phase of the recovery. In fact, judging by the plot in Figure 3B, the authors have only 1 data point before the plateau. How can a rate be accurately recovered from these data? Same concern in figure 5E.

I also have some more general comments.

8) There are in general a lot of problems with the figures and figure legends. Multiple plots are poorly described. For example, Figures 2 B and C, there is nothing on the figure or in the legend indicating which curves are the area and which are the anisotropy values. In Figure 3 D there is no information about the statistical test used. In Figure 5, the scatters plots have what I have to assume are linear regressions, yet nothing is said about how the lines were obtained and if statistical tests were used.

9) In general, the figures and figure legends are far from what the journal guidelines are regarding the reporting of data. For example, the figure legends do not report any sample size (number of cells, number of repetitions, etc…).

[Editors’ note: further revisions were suggested prior to acceptance, as described below.]

Thank you for resubmitting your work entitled "Exploring the role of macromolecular crowding and TNFR1 in cell volume control" for further consideration by *eLife*. Your revised article has been evaluated by Anna Akhmanova (Senior Editor) and a Reviewing Editor.

The manuscript has been improved but there are some remaining issues that need to be addressed, as outlined below.

*Reviewer #1:*

The authors have substantially improved the characterization of the use of fluorescence anisotropy of cytoplasmic GFP as a measure of macromolecular crowding (MMC). Their analysis of the change in anisotropy now correlates with the changes in refractive index of the medium, consistent with theoretical expectations. Therefore changes in the anisotropy as measured in a homogenous environment will reflect changes attributable to changes in MMC. They have addressed and also extended the comments raised in the initial review of their manuscript to provide adequate experimental and theoretical justification for the use of this tool.

Using this tool, they seek to relate the observations they make with the measure of MMC and its relationship to changes in cell volume and its control. However in this context there are many lacunae which chiefly concern the conflation of the use of the tool as a surrogate measure of cell volume, need to be addressed.

Recommendations for the authors

To relate the observations they make with the measure of MMC and its relationship to changes in cell volume and its control, substantial explanation and additional information is necessary.

Specific concerns:

1) Cell spreading and MMC: lines 202-302: The authors first try to explain the differences between the perinuclear and lamellipodial r EGFP values. They then show that the values that they measure in a widefield microscope are distorted because of the small thickness of the lamellipodia and that a confocal microscope gives less variable measurements between the lamellipodia and the cytosolic anisotropy. This calls into question the utility of this technique to measure anything that is spatially varying across a cell and all the observations about spatial homogenization upon various treatments could simply be an artifact of the measurement. Even comparisons between spread cells and rounded cells are not clearly measuring anisotropy but also will have information about the geometry of the cell.

In line 292: The authors claim that the cell volume increases upon cell spreading. This is contrary to what is seen in recent papers making similar measurements in similar ways with the same cell lines. This should be addressed and acknowledged. Ref: https://doi.org/10.1016/j.bpj.2017.11.3785, https://doi.org/10.1073/pnas.1705179114

2) Regulatory volume increase: lines 303-381: Here the authors use the r EGFP values to infer regulatory volume increase. The r EGFP value as an indicator of cell volume has not been clearly established earlier. The authors have also seen cases where depolymerizing microtubules and vimentin increases volume but does not affect r EGFP. Hence all the results of this section can only be interpreted as a change in the cytoplasmic MMC not as a change in the volume. And there needs to be an independent cell volume measurement for each treatment where regulatory volume increase is inferred.

3) Concentration versus anisotropy – lines 382:435: The authors try to provide an explanation for the deviation of the concentration vs anisotropy curve in differing cellular volume conditions, despite the cells having the same mass. They end up speculating on the role of phase separation as a mechanism for exclusion of the GFP tracer. The relevance of this section to the paper is not clear and only serves to remind the reader that the measured GFP anisotropy value is a complex function of cellular parameters and cannot be explained by simple volume scaling arguments.

4) The role of TFNR1: lines 436-508: Here the authors argue that hypotonicity induced p65 migration to the nucleus is mediated by TFNR1 receptor activation. They show that the effector RIPK1's association with TFNR1 is lesser in hyperosmotic shock conditions. They correlate the loss of volume regulation upon TFNR1 perturbation, with reduced p65 nuclear migration and RIPK1 recruitment to implicate TFNR1 directly in Regulatory Volume Increase. However, these separate events need not be connected and the causal role of TFNR1 in hypertonic stress induced cell volume regulation needs to be established more clearly. The fact that TFNR1 receptor perturbations lead to the complete downregulation of the instantaneous volume reduction in TFNR1kd and Zafirlukast as seen in figure 6C is very surprising. This also indicates that TFNR1 may be doing something very different and changing the set point of the cell even before the osmotic shock is applied. In figure 6C and 6D it is again clear that the cell volume measurements do not correlate with anisotropy values again underlining that r EGFP is not a surrogate measure of cellular volume.

*Reviewer #2:*

The authors drastically restructured their manuscript, which resulted in the great improvement as a scientific paper. Moreover, they performed additional experiments, and extended their findings to more concrete biological context (e.g., the involvement of TNFR1). Overall, I think they addressed most of my concerns, and thus I am positive for their main claims. At the same time, their new results raised additional points to be addressed.

1. I thank the authors for their in vitro experiment with additional crowding agents such as Ficoll and polyethylene glycol (Figure 1A). However, the result was a bit different from what I originally expected, i.e., I originally assumed that these crowding agents would show the similar EGFP fluorescence anisotropy to BSA, but the truth was that "BSA has the most prominent impact" (line 102). And the authors interpret that the EGFP fluorescence anisotropy is primarily for protein crowding (e.g., line 140, line 513). I agree that this new finding is quite interesting. At the same time, my original intention is not resolved yet, i.e., they cannot deny that their findings are specific to BSA (e.g., amino acid sequence), rather than general proteins. Hence, they should add another protein at least. Also, it may be one option to specifically claim protein crowding (instead of general macromolecular crowding) in Title and Abstract.

2. The authors largely improved the usage of regulatory volume increase (RVI) throughout the revised manuscript. However, they still use RVI for the result descriptions of Figure 4A-C ("the best RVI" in line 313, "partial RVI" and "RVI" in line 314, "RVI" in 315, "early RVI" in line 330, "RVI" in line 340, "RVI" in 349, etc). Their main claim is that the EGFP fluorescence anisotropy is not a RVI indicator but a protein crowding indicator. Hence, they should describe exactly what the result indicates (e.g., decrease/recovery from the ∆r_{EGFP} increase). Additionally, based on their introductory narrative (lines 304-310), I think that Figure S4D and its result sentence (lines 351-352) fit better than Figure 4A as the following result. I recommend them to revise the current Figure S4D as the new Figure 4A.

3. In contrast to the previous version, the authors eliminated regulatory volume decrease (RVD). However, they observed "dilution of MMC" (line 338) under hypotonicity (Figure 4C-i). I'm not sure due to the lack of recovery rate presentation here, but the trajectory seems to show that the hypotonicity-induced ∆r_{EGFP} decrease did not recover even after dozen minutes. Because cells efflux osmolytes to achieve RVD, this result may indicate that MMC decreased while cell volume recovered, i.e., the hypotonic version of the deviation from theory (Figure 5A). Hence, as well as the hypertonicity results (Figure S4D, 4B), the authors should compare the changes between ∆r_{EGFP} and cell volume change under hypotonic conditions.

4. The authors describe that the decrease from the ∆r_{EGFP} increase was "faster in the case of 100 mOsm change (comparing 50 mM NaCl and 100 mM mannitol/dextrose)" (lines 346-348). However, this description does not match with the results (Figure S4B, S4C), because we cannot recognize the differences at a glance especially due to the large SDs in Figure S4C. Similar to cell volume change, the authors should present statistical test(s) for recovery rates.

5. In all multiple hypothesis testing results (e.g., Figure 2E), the authors must describe how they adjusted P-values. Nominal P-value may be valid, depending on their null hypothesis. However, based on their approach (line 846), P-value adjustment (e.g., Bonferroni method) would be required for their interpretations.

6. As to Figure 4D, the authors observed that the difference in the EGFP diffusion rate between nucleus and cytoplasm diminished under hypertonic condition (lines 355-357). However, the most important finding of Figure 4D is that "hypertonicity-induced elevation of intracellular MMC concomitantly increases the cellular micro viscosity for both cytoplasmic and nucleoplasmic proteins" (lines 357-359). Hence, the authors should perform statistical tests for the differences not between cytoplasm vs. nucleus per mannitol concentration but between 0 mM vs. each concentration of mannitol per cytoplasm/nucleus. Moreover, Figure 4D seems a different topic from the other Figure 4 panels. Rather, Figure 4D and its description (lines 353-359) would fit better at the point before Figure 5D and its description (line 409).

7. In Figure 5D, the control comparison (i.e., isoosmotic condition) is required.

8. I completely disagree that the authors use "LLPS" from their results (e.g., line 424, line 426, line 428), because they did not investigate the material property of the p65 "condensates". In other words, they cannot deny the possibility that the p65 granules/structures are solid-like condensates or aggregates. At the same time, I believe that they do not intend to claim that p65 granules/structures under hypertonic conditions are novel LLPS-induced liquid-phase droplets, and thus they could resolve this point by wording. Otherwise, they need to perform multiple gold-standard experiments of the LLPS studies (e.g., FRAP, in vitro reconstruction).

*Reviewer #3:*

The manuscript presents a wide range of data, from in vitro characterization of their methods, to in vivo measurement of crowding either via established methods (FRAP, FLIM or FCS) and a comparison to the measurement of fluorescence anisotropy (FA for short) of eGFP.

Strengths:

While FA increases linearly under certain in vitro conditions, it only remains a proxy for crowding in vivo, where the scaling is unsure but with a likely increase with crowding. As such, this method is interesting as it would be easy to implement in a lab and could provide an easy way to estimate changes in crowding under various physico-chemical conditions, or even intracellularly. The method has been carefully established in vitro and compared to well-established methods. in vivo, hyperosmotic stresses, known to increase crowding, have been shown to increase FA in 4 different cell lines.

Weaknesses:

I have noted several weaknesses which I summarize in the five points below:

1. Even if the change of FA was carefully studied in vitro, the choice of chemical conditions can affect the response and can be far from the in vivo reality. In particular, one point that is unclear to me is the conclusion of the authors on the absence of dependency of reGFP on micro-viscosity. The authors based their conclusion on the effect of the variation of glycerol concentration from 80% (v/v) to 90% (v/v) for which they indeed observed only a mild increase in FA. However, what is the rationale for starting at 80% (v/v)? This concentration is already very high, with a solution micro-viscosity that is more than 20 times the one of water, thus larger than the typical micro-viscosity inside a cell. It can thus be possible that there is an effect of micro-viscosity, at lower values. One result that would argue for this is the fact that at 80% (v/v) glycerol, reGFP is already much higher (0.24) than the highest value measured at 4 mM BSA (0.22), or the condition in water (0.20). As such, the data presented by the authors do not convincingly show that reGFP is independent of micro-viscosity. Moreover, based on their rationale of the effect of crowding on reGFP on rotational diffusion of the molecule, I would find it odd that micro-viscosity plays no role at all.

2. The authors could have performed a direct comparison of FA with another method in vivo, to plot in particular the change in reGFP as a function of intracellular translational diffusion of eGFP. This data could help the reader better understand the in vivo link between FA and classical metrics such as diffusion coefficients.

3. The authors show that homoFRET would occur at eGFP concentration of 10 μM or higher, and estimated that in isotonic conditions, the eGFP concentration in NIH/3T3 cells is of ∼ 8 μM. However, when the cells are osmotically compressed, cell volume decreases by up to 40%, raising the concentration above 10 μM. The authors do not discuss this point which is crucial and could limit the use of their methods in this particular case.

4. The link between the cellular response to regulated cell volume increase (RVI) is not well established. In particular, it seems that RVI is not the only route to osmoadaptation in the authors' data: one can clearly observe, e.g. in Figure S4D, in particular for HeLa cells, that, whatever reGFP really measures, its value is almost back to isotonic condition while cell volume is not. Similarly, treatment with Flufemanic acid shows that all cannot be attributed to RVI. Thus, the authors should not consider RVI as the only osmo-adapting mechanism, as their data do not show this.

5. The link of TNFR1 with osmoadaptation is not properly demonstrated. In Figure 6C, RVI is partly abrogated under the inhibition of TNFR1. But what I found odd is the fact that volume decrease is also decreased, for both HeLa and NIH/3T3 cells, which is confirmed by FA increasing less than in control conditions. This is not discussed and is probably associated with other effects of the KD of TNFR1 or Zafirlukast, for instance on cell volume or crowding, or on cytoskeleton tension (as Figure S6Dii shows, even if this data is not discussed in the main text!). As the authors do not present the values of reGFP and cell volume under KD of TNFR1 but only their relative change, it is hard to understand what happens. This makes the interpretation of the authors on the putative role of TNFR1 not convincing.

Recommendations for the authors

Below is a list of recommendations associated with the five aforementioned weaknesses of the manuscript, followed by other points and recommendations on other parts.

1. I suggest the authors redo the in vitro experiments with lower values of glycerol, and possibly another viscogen. It could help the reader better understand how FA scales with crowding. The use of cell extracts could also be interesting to better establish the method in vitro.

2. The link with in vivo is hard to make, as crowding is much more complex. However, the authors should provide data comparing in vivo FA to the measurement of eGFP translational diffusion coefficients, for instance using FRAP.

3. The authors should measure the concentration of eGFP under various osmotic conditions to ensure that part of the FA changes measured in vitro cannot be attributed to other effects such as homoFRET, as their data suggest. This could change their interpretation of the results plotted in figure 5 on the deviation from the concentration-dilution regime (see specific point 2 below).

4. I suggest the authors put less emphasis on RVI as the main osmoadapting mechanism in their experiments. Some other routes can exist and should be discussed. This is fine if crowding is not fully correlated to RVI, as the data of the authors seem to show.

5. More experiments and discussion of the data are needed to reach the conclusions the authors reached on TNFR1. The authors should show the effect of the KD alone on different biophysical parameters (like cell volume and reGFP, as they have for tension), and discuss them. I was, for instance, extremely surprised to see experiments of Figure S6Dii, which seems of primordial importance to understand and interpret the effect, not at all discussed in the main text. In general, and this is true also for CAY10512 treatment, the authors should not only display the change of the mode of the data but the whole distribution. How is it impacted? Does it, in particular, become bimodal, with a proportion of cells responding to osmotic stresses while others do not, partly explaining what the authors show? It is a shame to have access to very rich single-cell data and only use mean or median, while distributions can be extremely informative.

Additionally, even if these points are not weaknesses of the manuscript, these are further questions the authors should address:

1. Figure 2B: I understand the data, but I think that the conclusion is too quickly reached. Cells of different aspect ratios can have different spreading and thus different heterogeneity or crowding. Before using the median as a metric and disregarding potential differences at different aspect ratios, I would appreciate having another measurement of crowding (for instance, by FRAP) of translation diffusion of eGFP in cells or different aspect ratios.

2. We discussed homoFRET in the above points 3. Regarding Figure 5 and the discussion about the deviation from the concentration-dilution curve, why the authors do calibrate this curve using the two extreme values of FA (0mM and 600mM)? Clearly, in the case of homoFRET, the 600mM value could not be used. Maybe if the curve were calibrated with other lower values, you would only find a deviation from the concentration-dilution curve at high osmotic stresses, which could then either be a sign of non-linearity of FA dependency in vivo with crowding or some other processes. The authors should clarify this point before reaching their conclusions about a deviation from the concentration-dilution regime.

---

## [Author Response]

[Editors’ note: the authors resubmitted a revised version of the paper for consideration. What follows is the authors’ response to the first round of review.]

Comments to the Authors:We are sorry to say that, after consultation with the reviewers, we have decided that this work will not be considered further for publication by eLife.Specifically, all the reviewers felt that there were serious concerns regarding both technical and methodological aspects of the work that seriously limited the conclusions about the method as well as the relationship between MMC and cell volume regulation, that could be drawn from the data. In addition, some of the experiments and their conclusions are not consistent, as detailed in the extensive reviews from the reviewers provided. Any revision would require a lot more time and extensive additional experimentation whose outcomes are likely uncertain, and, inconsistent with the policies of a revised manuscript at eLife. While some of these weaknesses have been highlighted in the public review, detailed reviewer reports are also provided.In this manuscript, the authors attempt to establish steady-state fluorescence anisotropy of eGFP as a sensor for macromolecular crowding (MMC). This is a valuable tool if it can be fully validated. Traditionally this is a method used to assay the fractions of bound versus unbound in protein-protein and protein-ligand interactions due to an increase in anisotropy as a consequence of an increase in molecular mass. It has also been used to assess molecular proximity since this causes a decrease in anisotropy due to homo-FRET; it has not been used to estimate crowding. The results of in vitro experiments suggest that this method is practically insensitive to pH, ion concentration, small molecules, and probe (i.e., EGFP) concentration, and thus robust enough to evaluate the effects of MMC in cells. The results of in vivo experiments "probably" suggest many interesting findings of the associations between intracellular MMC and cell volume.Strengths:Given that MMC has been usually studied using in vitro and in silico experiments thus far, this quantitative method using fluorescence anisotropy has a great potential to bridge the gaps in our understanding of MMC between in vitro/theory and physiological conditions. In addition, the authors applied this method to investigate the relationship between intracellular MMC and cell volume regulation, which helps readers to understand the application utility of this method in cell experiments. Moreover, it should be noted that their idea about applying this method to the context of cell volume regulation is great because one of the challenges in this context is distinguishing the actual changes in intracellular events (i.e., MMC in this case) from probe-derived changes (i.e., EGFP concentration in this case) for any detected differences of a probe measure, but this challenge could be resolved in this method.Weaknesses:Unfortunately, the current manuscript is immature as a scientific article, because of lacking sufficient information for readers to understand what their results indicate objectively. The data is not presented with adequate care with regards to a description of methodology and statistics. The method needs to be better benchmarked before conclusions can be reached about the meaning of the observed changes in anisotropy, especially in the comparison between the in vitro and in vivo changes observed. As this is the only method used by the authors to estimate crowding many of the conclusions depend on the validation of the methodology. The authors also fail to put their observation in the context of the available literature on MMC cell volume regulation and cell volume changes, making it harder for the reader to evaluate. This study contains a critical concern in the overall logic.To evaluate the relationships between intracellular MMC and cell volume "regulation" by utilizing their method, scientists have to recognize that the detected change of fluorescence anisotropy in cells under osmotic perturbation reflects at least four-level of changes: (1) change in a potential molecule specifically affecting excitation/emission process of probe itself, (2) change in the absolute amount of intracellular macromolecules, (3) passive change in cell volume, which is caused by osmotic water flux, and (4) regulatory change in cell volume, which is positively induced by cells using intracellular energy like ATP and chemical potential (i.e., regulation, such as regulatory volume increase (RVI) and regulatory volume decrease (RVD) under osmotic stress).In this study, the authors address points (1) and (2) through a series of in vitro experiments (Figure 1), however, the data do not provide a clear understanding of the origin of the changes in GFP anisotropy in cells. For example, GFP shows a change in a lifetime in cells of 0.1 ns (from 2.5 ns in isotonic conditions to 2.4 ns) when shifted to hypertonic conditions, and correspondingly, the anisotropy changes by 0.032 from 0.172 to 0.204 (see Figure 2B). This is inconsistent with a 1 ns decrease in GFP-lifetime in solution in vitro (Figure 1C) that leads to a difference of 0.016 in anisotropy (Figure 1A). This discrepancy indicates that there may be other processes happening in the cell apart from changes in GFP lifetime (related to refractive index) that are dictating corresponding changes in anisotropy. Therefore the authors cannot simply consider that the detected change of fluorescence anisotropy in cells relates to the change in intracellular MMC. This study also did not distinguish points (3) and (4) at all, implying that the authors can infer the associations only between intracellular MMC and cell volume, but not between intracellular MMC and cell volume "regulation".The authors seemed to interpret that the detected changes in cell volume/area correspond to RVI or RVD. At least, they should evaluate cell volume regulation using well-known inhibitors, which have been characterized in the field of electrophysiology (e.g., flufenamate for RVI, NPPB, or DCPIB for RVD; see such as Wehner, F. et al. FEBS Lett. 2003 and Ponce, A. et al. Cell Physiol. Biochem. 2012), to support their interpretations. In addition, they should present a quantitative index for RVI/RVD, because several figures clearly show a difference in the enforced cell shrinkage/swelling (i.e., the difference between isotonic condition and the first time point after osmotic treatment). One of the commonly used indexes for RVI/RVD is the recovery rate from the first time point after osmotic treatment. Moreover (possibly related to the aforementioned major point 1), the authors' usage of RVI and RVD (e.g., in line 307) raises a concern about their understanding of the terms of RVI and RVD. In general, RVI and RVD indicate the cell volume recovery followed by hypertonic stress and hypotonic stress, respectively. And, the cell volume change free from osmotic stress is not referred to as RVI or RVD. For instance, in the case of cell death, the accompanying cell volume change is differently termed as apoptosis volume decrease (AVD) even though the same effector molecule is involved in RVD and AVD (cf. Okada, Y. et al. J. Physiol. 2001). Many aspects of the paper about the relationships between intracellular MMC and cell volume "regulation" seem NOT to be logically supported by their results.Finally, in terms of data presentation, the figures and figure legends are far from complete regarding the reporting of data. For example, the figure legends do not report any sample size (number of cells, number of repetitions, etc…).Judging from the plot in Figure 3 (because that information is missing from the methods) the FRAP experiment was performed at 2 frames per second. The resulting recovery curves do not have enough points to properly recover the exponential phase of the recovery. In fact, judging by the plot in Figure 3B, the authors have only 1 data point before the plateau. How can a rate be accurately recovered from these data? A similar concern is raised in Figure 5E. Multiple plots are poorly described. For example, in Figures 2B and C, there is nothing on the figure or in the legend indicating which curves are the area and which are the anisotropy values. In Figure 3 D there is no information about the statistical test used. In Figure 5, the scatter plots have what we may assume are linear regressions, yet nothing is said about how the lines were obtained and if statistical tests were used.Reviewer #1:In this manuscript, the authors attempt to establish steady-state fluorescence anisotropy of eGFP as a sensor for macromolecular crowding. This is a valuable tool if it can be fully validated. They then explore various means of perturbation of MMC as a way to determine if it acts as a homeostat for cell volume regulation. They conclude that MMC is unlikely to be a control parameter in cell volume regulation since changes in MMC do not influence regulated volume changes, whereas cell volume changes do affect the MMC, in a predictable manner. This is potentially overturning a long-held view regarding the role of MMC in cell volume regulation. They suggest that volume regulation is affected by the homeostatic control of plasma membrane tension. However, there are several issues both technical and conceptual that need to be addressed before the conclusions put forward are convincing:1. Technical: The authors first test the dependence of GFP steady-state anisotropy on various environmental factors such as pH, viscosity, and concentration. However, there are several points that need to be clarified before the use of emission anisotropy values of EGFP may be reliably taken as a measure of MMC in the cell, as reported.a) The authors show in Figure 1E that the anisotropy of GFP is independent of concentration, by showing GFP concentrations ranging from 1-9mM (milliMolar). This range of concentrations should result in significant Homo-FRET (FRET between GFP-GFP molecules). Concentration-dependent depolarization of the GFP emission due to homo-FRET has been shown in many studies (for example for a detailed analysis see-https://doi.org/10.1016/S0006-3495(02)73932-5). Ranging from a few μm to 0.6mM of purified GFP, a clear concentration dependence of GFP on anisotropy is seen. A similar study with VFP (DOI: 10.1042/bst0311020) also finds the same effect. It is puzzling to see this apparent independence of eGFP anisotropy reported by the authors. It is likely that eGFP has already entered a state of saturating FRET at these concentrations. It would be very useful to have a larger range of concentrations sampled here, starting from as little as 250nM going into the milliMolar regime.

The X-axis labels in Figure 1E mentioning the unit of EGFP concentration is a typographical error. We apologize profusely for our mistake and all the confusion. The maximum concentration of EGFP used for experiments described in Figure 1E should be 9 µM, as per our 488 nm absorbance measurements, and not 9 mM as indicated in the current manuscript. The likely cause of the typo was a character conversion failure by our Origin graph plotting software (from English ‘m’ to Greek ‘µ’), and we overlooked the error. Furthermore, we have re-evaluated the concentration of EGFP (denoted as [EGFP]) by FCS (Fluorescence Correlation Spectroscopy), which puts the maximum [EGFP] in Figure 1E at ~540 µM. The new manuscript contains the re-evaluated calculation in Figure 1F. To validate the absence of homo-FRET in our intracellular *r_EGFP_* measurements, we performed photobleaching experiments in EGFP (monomer) and 2GFP (EGFP dimer) expressing NIH/3T3, explained in Figure S2B in the new manuscript.

b) Another discrepancy is the value of anisotropy. The authors do not mention the concentration of GFP used to perform the experiments in Figure1 (A, B, D, F). The methods only specify that they used a 1:100 dilution of the purified GFP.

We used 50 nM EGFP for all our experiments except Figure 1F.

The concentrations may help to clarify the disparities the authors see in the starting values of GFP anisotropy between Figure1(A, B, E, F) and Figure1(D). The anisotropy in Figure1 (A, B, E, F) starts consistently at a value of 0.20, and only in 1D, the value is at 0.24. An anisotropy difference of 0.04 between different experiments is a really large change (bigger than the largest differences observed with cells or with BSA in the paper) and the reasons for such variability need to be explicitly addressed.

In Figure 1 (A, B, E, F), r*_EGFP_* values start at 0.20 because the measurements were made in 1xPBS. In Figure 1D, the first measurement of r*_EGFP_* was made in an 80% glycerol solution (as indicated in the X-axis), which is why the value starts at ~0.24 (because of higher refractive index of the glycerol solution). The indicated range of glycerol concentration was chosen to highlight the insensitivity of r*_EGFP_* to solution viscosity, because for EGFP, *τ*⁄*θ_C_* < 1 in water as well as 80% glycerol, while for fluorescein, *τ*⁄*θ_C_* > 1 in water and ~1 in 80% glycerol. As a result, a change in solution viscosity will have negligible effect on EGFP anisotropy. The result is available in Figure 1E in the revised manuscript. This is an important result because the solution viscosity of cell cytoplasm may change appreciably due to temperature or pH fluctuations, which will have negligible impact on the MMC sensing property of r*_EGFP_*.

c) The differences in eGFP lifetime shown here in Figure1C have been reported before (in the papers of Suhling et al. in 2002, cited also by the authors) and it is understood that these differences are due to a change in the refractive index of the medium. Suhling et al. also show that the lifetime of the fluorophore is agnostic to the nature of the "macromolecule" itself and that various concentrations of glycerol, PEG, NaCl, glucose, or fructose only affect the lifetime by changing the refractive index of the solution. So it is confusing in Figure 1C to plot the dependence of lifetime on the concentrations of BSA and HiSep as two separate graphs when we know that both can collapse on the same curve if the refractive indices of these solutions are plotted on the x-axis instead of the concentrations. Hence it will be good to rework the plots in Figure1 replacing the concentration axes with refractive index instead, Especially in Figure 1C it will be useful to see if the data match the predictions of the Strickler-Berg equation as shown earlier by Suhling.

The change in fluorescence lifetime of EGFP is driven primarily by changes in the solution refractive index. We agree with the reviewer that if we plot refractive index vs lifetime, both the graphs will fit in the same X-axis. We show lifetime vs squared refractive index (Figure S1B-C) in the new manuscript to validate the Strickler Berg equation. The purpose of using concentration in Figure 1C was to help the reader visualize that a similar change in intracellular protein vs polysucrose concentration induces a significantly higher change in fluorescence lifetime in the case of proteins, as compared to smaller molecules like polysucrose. This graph implies that hundreds of mM change in polysucrose concentration will lead to a very small change in the measured fluorescence anisotropy, whereas only a few mM change on protein concentration will introduce a much higher change in anisotropy. Therefore, we propose r*_EGFP_* as a protein crowder sensor.

d) The authors report a decrease in the lifetime of GFP with increasing refractive indices, which is then being indirectly read out by a change in its steady state anisotropy. As the authors also mention, the Perrin equation governs the relationship between anisotropy and lifetime. This needs to be verified for the system in question with an Anisotropy vs Fluorescence Lifetime graph. Between Figures 1A, 1C, and S1B, the authors already have the data to do so. An agreement with the predictions of the Perrin equation will be useful for the authors' efforts to validate the use of anisotropy as a readout of refractive index-mediated fluorescence lifetime changes.

We plotted the directly measured *r_EGFP_* vs. reconstructed the values of steady-state fluorescence anisotropy calculated from the time-resolved data of EGFP’s fluorescence lifetime (*τ*), rotational correlation time (*θ_C_*), and the intrinsic anisotropy *r*_0_ in different BSA concentrations in the new manuscript (Figure 1D). The difference in measured Vs estimated value is due to additional depolarization of fluorescence by the instrument.

e) It must be mentioned that in a series of papers by the group of Paras Prasad (doi: 10.1021/cb300065w., https://doi.org/10.1038/s41467-019-08354-3, doi: 10.1088/2050-6120/ab8571), they establish fluorescence lifetime of eGFP as a sensor for the local refractive index. They also make the claim that the refractive index map of a cell could serve as a measure of local macromolecular concentration. In the supplementary of the Nature Comms paper linked above, they clearly show the effect of BSA concentration in increasing the refractive index of the buffer and show a decrease in eGFP lifetime in accordance with the relationship predicted by the Strickler-Berg equation. The authors of this submission should cite the work mentioned above and situate the relevance of the current study in the light of what is already known.

We have cited the mentioned articles at the appropriate places in the revised manuscript. Indeed, FLIM can reliably map cellular MMC but fluorescence anisotropy is more high-throughput and cheaper to implement.

2. Experiments with changes in reGFP in cells: These experiments try to understand the effect of changing the crowding state of the cell (by osmotic shock or cell spreading) and try to connect it to the steady state anisotropy, and subsequently cell volume. However, a few confounding factors need to be addressed to allow a meaningful interpretation of the results observed.a) The authors write in line 99 – "The observed increase of rEGFP due to increased cytoplasmic MMC is caused primarily by the decrement of its fluorescence lifetime". Figure 2D shows a lifetime change in cells of 0.1 ns (from 2.5ns in isotonic conditions to 2.4ns) in hypertonic and correspondingly, the anisotropy changes by 0.032 units from 0.172 to 0.204(Figure2B). This is inconsistent with a 1 ns decrease in a lifetime in Figure 1C which leads to a difference of 0.016 units in Figure 1A. This discrepancy indicates that there may be other processes happening in the cell apart from changes in GFP lifetime (or refractive index) that are dictating corresponding changes in anisotropy.

We agree with the excellent observation of the reviewer that the change in fluorescence anisotropy (∆*r_EGFP_*) observed in solution due to a change in fluorescence lifetime (∆*τ*) is smaller than the intracellular ∆*r_EGFP_* observed for the same ∆*τ*.There are multiple reasons that we could ascribe to the discrepancies- (I) In addition to fluorescence lifetime, the intrinsic fluorescence anisotropy (*r*_0_) also changes with MMC (revised Figure 1C-ii ,Figure 2A-iv). (II) The *τ_EGFP_* and *r_EGFP_* measurements were performed on multiple cells which show heterogeneous response to hypertonic stress. Therefore, for the same ∆*τ_EGFP_*, the cellular ∆*r_EGFP_* values might not agree with the in vitro values. Upon measuring all the parameters in the same cell, (Figure 2A in the revised manuscript), we find that the Perrin equation is sufficient to describe the observed changes of ∆*r_EGFP_* in cells and there are no unknown processes that causes the observed changes in *r_EGFP_*. (III) Most importantly, we ascribe this inconsistency to the heterogeneity of intracellular crowder molecules creating a much higher net refractive index in the cell than what we can generate in vitro using solutions of single crowder molecules. The phenomenon can be understood by comparing the fluorescence lifetime and anisotropy measurements of EGFP in BSA and Ficoll solutions (Figure 1A-B, revised). Ficoll has a lower refractive index than BSA in solution, so Ficoll needs a significantly larger change in concentration than BSA to impart an appreciable change in both lifetime and anisotropy values. Between the maximum concentration change in EGFP-BSA solutions (comparing Figure 1B and Figure 1A), ∆*τ* = −0.87 ns, while ∆*r_EGFP_* = 0.018. In case of EGFP-Ficoll solutions (comparing Figure 1B and Figure 1A), ∆*τ* = −0.38 ns, while ∆*r_EGFP_* = 0.007. Thus, for a ∆*τ* = −0.1 ns, the corresponding ∆*r_EGFP_* = 0.0021 for EGFP-BSA solutions (higher refractive index) and ∆*r_EGFP_* = 0.0018 for EGFP-Ficoll solutions (lower refractive index). The same result can be extrapolated for the ∆*r_EGFP_* and ∆*τ* comparison between the cytoplasm and BSA solutions. The cytoplasm being a milieu of macromolecules generates higher ∆*r_EGFP_* and ∆*τ* than pure BSA solutions. The additional factor of intrinsic anisotropy *r*_0_ scaling linearly with increased crowding could be due to changes in the angle between the absorption and emission dipole moments of the EGFP chromophore. We are unsure of the mechanism behind this effect and are still characterizing it. Direct measurement of refractive indices of the crowder solutions should strongly establish this observation, and we have included it in the revised manuscript (Figure S1C).

b) An important aspect of the data acquired from the cell experiments is the GFP concentration itself. The GFP concentration which can be read out by the mean GFP intensity per voxel should ideally respond to the changes in cell volume perturbation. GFP concentration will also be useful to understand the effect of various treatments on the cell area and volume. One way to represent this information is to plot Anisotropy as a function of Total Intensity. The GFP concentration in cells should also ideally correspond to the range of concentrations sampled in Figure 1 to decouple the concentration dependence of GFP anisotropy (as indicated above).

Change of fluorescence intensity readout in the cells to understand change in volume is indeed the simplest method. However, we observe that intracellular EGFP undergoes a change in brightness with decreased volume and increased crowding during hypertonic shock. Author response image 1 depicts 16-bit images of NIH/3T3-EGFP in isotonic and hypertonic conditions (additional 600 mM Dextrose). With the same intensity grayscale for both images, the mean intensity of the pixels inside the red square is different under both conditions, as illustrated in Author response image 1. Upon plotting the average total intensity per cell, normalized by the pre-hypertonic shock value, we observe a dextrose concentration dependent reduction in the total intensity change (Author response image 1). The same is observed in purified EGFP with BSA as crowders. Therefore, this makes it difficult to use intensity change as a readout of volume change. Additionally, over long timescales, the cellular fluorescence intensity could also change due to biological causes that directly affect intracellular EGFP concentration without affecting cell volume. Decoupling the effects of EGFP concentration and volume thus becomes difficult in such situations.

**Author response image 1. sa2fig1:** Change in the brightness of EGFP with increased MMC during hypertonic shock. (**A**) Ttotal intensity images of cells in isotonic and hypertonic condition (+600 mM dextrose). (**B**) Total intensity trajectories normalized by the initial total intensity per cell. Error bars indicate standard deviation of the variabilities in the normalized total intensity values. n=128 (isotonic), 83 (+50 mM), 56 (+100 mM), 109 (+150 mM), 66 (+200 mM), 92 (+400 mM) and 112 (+600 mM).

c) Using the lifetime of GFP as a censor for MMC has been done before. However steady-state anisotropy is simpler and faster to measure on a conventional microscope than doing FLIM. If the above concerns are addressed then this may prove to be a more accessible way to map local refractive indices in cells. All the other results of the paper can only be interpreted once there is a clearer understanding of what is driving the anisotropy differences seen in cells.

In light of our justifications given above, we hope the reviewer would agree that the anisotropy change of EGFP is driven primarily by change in the lifetime and the intrinsic anisotropy (r_0_) (via refractive index) (Figure 1C and 2A in the revised manuscript).

3. MMC, cell volume homeostasis, and plasma membrane tension. This part of the research – both underlying the relationship of MMC and cell volume regulation may only be understood objectively if the above technical issues are completely clarified. At this point, it is only worth noting that changes in MMC as reported by changes in rEGFP do not correlate with what is expected of a sensor for cell volume homeostasis.

Since we have ruled out the factors like homo-FRET and viscosity, we hope the reviewer would agree that a change in anisotropy is solely due to a change in macromolecular crowding.

The experiments with changes in plasma membrane tension, cell spreading, and cell volume regulation, including the technique of fluctuation spectroscopy to infer plasma membrane tension changes, are interesting but too preliminary to provide an alternative homeostat sensor for cell volume regulation.

We show that impairment of RIPK1 recruitment in the TNFR1 complex inhibits NFkB signaling and RVI in the new manuscript (Figure S6).

Reviewer #2:The authors present a method to quantitatively investigate the effects of macromolecular crowding (MMC) in vitro and in cells, using fluorescence anisotropy of EGFP. Their results of in vitro experiments "probably" suggest that this method is practically insensitive to pH, ion concentration, small molecules, and probe (i.e., EGFP) concentration, and thus robust enough to evaluate the effects of MMC in cells. Their results of in-cell experiments "probably" suggest many interesting findings of the associations between intracellular MMC and cell volume. However, the current manuscript is immature as a scientific article, because of lacking sufficient information for readers to understand what their results indicate objectively. Moreover, even if a reader makes the maximum effort to guess what the authors really wanted to describe, their statements about the relationships between intracellular MMC and cell volume "regulation" seem NOT logically supported by their results.Strengths:Given that MMC has been usually studied using in vitro and in silico experiments thus far, this quantitative method using fluorescence anisotropy has a great potential to bridge the gaps in our understanding of MMC between in vitro/theory and physiological conditions. In addition, the authors applied this method to investigate the relationship between intracellular MMC and cell volume regulation, which helps readers to understand the application utility of this method in cell experiments. Moreover, it should be noted that their idea about applying this method to the context of cell volume regulation is great because one of the challenges in this context is distinguishing the actual changes in intracellular events (i.e., MMC in this case) from probe-derived changes (i.e., EGFP concentration in this case) for any detected differences of a probe measure, but this challenge is resolved in this method.Weaknesses:Unfortunately, the current manuscript is immature as a scientific article, because of lacking sufficient information for readers to understand what their results indicate objectively. Critical points include: (i) there is no description of what the variance ranges in sentences (e.g., 4.2 {plus minus} 1.1 cP in line 163) and the error bars in all figures indicate (i.e., standard deviation (SD), standard error (SE), confidence interval (CI), or something), (ii) some figure legends (e.g., as to Figure 2B) have insufficient information about the annotations (even within the main text), (iii) there are methods whose details are not described in the Method section (e.g., Bradford assay used in Figure 2H and 4B, cell cycle synchronization used in Figure 5 and S5), and (iv) most sentences are ambiguous for readers to understand whether their statements are objectively/logically deduced by their results or subjectively described as their interpretations. All these points are essential in science (e.g., points (i)-(iii) are important from the perspective of reproducibility). Of note for point (iv), some journals allow authors to describe "Ideas and Speculation" in the Discussion section, but this doesn't mean that the authors can mix up the objective findings with their subjective interpretations/ideas/speculations.

We apologize for the confusion created by the lack of clarity in the data we presented. We have added all the important information to the quantified values we have mentioned in the revised manuscript.

Even after a reviewer made the maximum effort to try to understand what the authors really wanted to describe, some statements seemed contradictory to what their results actually indicate. For instance, in lines 97-99, the authors probably wanted to describe that the black rectangle markers in Figure 2C corresponded to the right axis and indicated the trajectory of the cell area under hypotonic stress (this is unclear due to point (ii)) and that the cell area was decreased during hypotonic stress based on the comparison of these mean (or certain summary statistic) values at 120 min vs. at the first time point after hypotonic treatment.

We apologize profusely for the confusion in Figure 2B and 2C, the black rectangles do indeed indicate cell area and circles indicate anisotropy values. We have discarded the indicated figures in the revised manuscript.

However, even if assuming the error bars indicated SEMs (this is unclear due to point (i)), no scientist could conclude that there was a difference in these values based on Figure 2C, because the error bars clearly overlapped between these time points and a simple t-test, which was only the information described in the Method section, would not exhibit statistical significance for them (regardless using a statistical model with certain covariates).

We have stringently described the error bars and statistical analysis for every figure in the new manuscript.

Apart from these inconsistencies between their statements and results, this study contains a critical concern in the overall logic. To evaluate the relationships between intracellular MMC and cell volume "regulation" by utilizing their method, scientists have to recognize that the detected change of fluorescence anisotropy in cells under osmotic perturbation reflects at least four-level of changes: (1) change in a potential molecule specifically affecting excitation/emission process of probe itself, (2) change in the absolute amount of intracellular macromolecules, (3) passive change in cell volume, which is caused by osmotic water flux, and (4) regulatory change in cell volume, which is positively induced by cells using intracellular energy like ATP and chemical potential (i.e., regulation, such as RVI and RVD under osmotic stress). In this study, points (1) and (2) seemed to be addressed by the series of in vitro experiments, and scientists can consider that the detected change of fluorescence anisotropy in cells basically indicates the change in intracellular MMC. However, this study did not distinguish points (3) and (4) at all. Therefore, scientists can infer the associations only between intracellular MMC and cell volume, but not between intracellular MMC and cell volume "regulation".

We do agree with the reviewer’s comment that cellular *r_EGFP_* measurements can only generate information about the association of intracellular MMC to cell volume, but *r_EGFP_* alone is not enough to distinguish between passive and active cell volume changes. Post induction of hypertonic stress, the passive cell volume shrinkage and the active RVI process could occur simultaneously but at different rates, such that RVI is visible only after the rate of passive cell shrinkage has reduced significantly. Measuring *r_EGFP_* could not decouple the active and passive processes, but led us to the observation that cells lose their RVI ability at severe hypertonic stresses. Our experiments demonstrate that the TNFR1 activity is essential for initiating RVI, and in the revised manuscript, we show that increased MMC at severe hypertonic stresses reduces the efficacy of RIPK1 association with the TNFR1 signaling complex, thus impeding RVI. Through FRAP experiments, we show that cytoplasmic viscosity increases 15-fold during severe hypertonic stresses, and validate the role of MMC in reducing protein mobility and molecular signaling efficiency. While MMC does not appear to have a significant role in affecting RVI at low to medium hypertonic stresses (50-200 mM), the increased MMC at severe hypertonic stresses (>~600 mM) causes NIH/3T3 and HeLa cells to lose their RVI abilities owing to the elevated cytoplasmic viscosity. We speculate that a more in-depth investigation of the TNFR1 complex’s signal initiation would lead to a better understanding of when the molecular switch for RVI is activated.

Furthermore, there are scientific concerns in this study. Other than osmotic perturbations, the authors utilized alternative methods to operate intracellular MMC. However, their assumptions would be regarded as too bold assumptions in modern cell biology. For instance, they utilized Heclin to increase intracellular MMC under the assumption that Heclin reduces protein degradation via HECT inhibition (lines 130-131). However, HECT is just one family of the E3 ligases in a cell, and the other ubiquitin-proteasome system is still active under the Heclin treatment. Although Figure 2H demonstrated the significant (slight) effect of Heclin on the overall protein amount, this result can imply only the association and cannot deny the possibility that HECT affected the EGFP fluorescence anisotropy independently through its effect on the global protein degradation. In fact, if agreeing with their interpretations that the detected signals in Figure S4 indicated "aggregation", Figure S4 exhibited that the Heclin treatment affected the "aggregation" more strongly than hypertonic treatment, while Figure 2H suggested the weaker effect of Heclin treatment on the fluorescence anisotropy, rather in line with the above concerning possibility.

The objective of Heclin treatment was to increase the cytosolic protein content. Though other ubiquitin-proteasome pathways may still be active, we still manage to get a small but significant increase in protein content per cell. Through Figure 4D, we wanted to highlight the observation that localized clustering of EGFP increases the anisotropy values, which occurs during hypertonic treatment (Figure 4A) and cell fixation (Figure 4C). We have included the new explanation in Figure 5, S5 of the revised manuscript.

Likewise, the authors utilized heat shock to decrease intracellular MMC under the assumption that heat shock promotes global protein degradation (lines 133-134). However, the Perrin equation (line 65) indicates that fluorescence anisotropy is negatively related to temperature, and it can be assumed that heat shock resulted in the decrease in the EGFP fluorescence anisotropy independently through its effect on the protein degradation.

The heat shock was not provided to the cells during imaging. The cells were brought back to 37°C prior to fluorescence anisotropy measurements, thus eliminating the effect of ambient temperature on anisotropy measurements. We apologize for not indicating this crucial point in the method section. In the revised manuscript it is indicated in Figure 4E.

Recommendations for the authors:1. Although I have great interest in the authors' results, I strongly recommend that the authors should drastically revise the current manuscript to be understood by standard scientists. I mention concrete examples in the below minor points, but I cannot cover all points within this comment. If the author who mainly prepared the manuscript is unfamiliar with scientific writing, I recommend he/she first learns the general manner of scientific writing.2. In this manuscript, Figure 1 is important to support that the EGFP fluorescence anisotropy indicates the effect of MMC. Given that the method itself is one of the conceptual advances, the authors should also evaluate the relationship between EGFP fluorescence anisotropy and MMC using other crowding agents as well as BSA (e.g., Ficoll, polyethylene glycol). The expected results would be comparable to the results of BSA.

We have employed other polymers as crowding agents in Figure 1A.

3. In this study, the authors mainly used dextrose (i.e., D-glucose) as a means to prepare a hypertonic medium. Although they also demonstrated mannitol-based hypertonic treatment in Figure S2F, the main usage of glucose-based hypertonic treatment would raise eyebrows of most scientists studying cell volume regulation. It's primarily because the glucose concentration is approximately from 5 mM to 50 mM in most cell culture media, and it is not difficult for scientists to assume the potential effects of extremely high glucose on intracellular signaling, for example, via glucose transporters (Note that glucose transporters were reported to be involved in cell volume regulation too). Therefore, the authors should perform the additional experiments using mannitol-based hypertonic treatment, at least, in Figure 2G to support their interpretations.

We have reperformed the osmotic imbalance experiments with mannitol to validate all our data.

Related to this point about osmotic treatments and the aforementioned major point 1, the current Method section does not include information, but the authors may need to use "mM" instead of "mOsm" in this manuscript if they prepared hypertonic solutions based on the calculation of the osmolyte amount, not based on the direct measurement of osmolality using osmometers.

We have changed our terminology from “mOsm” to “mM”.

Additionally, whether they maintained CO2 level after the replacement with hypertonic solution or not should be clearly described because it is known that bicarbonate (HCO3) is critical for RVI (in contrast to RVD) due to the function of an RVI activator, NHE (sodium-hydrogen exchanger), which would affect the interpretation about cell line comparison (Figure 2G).

CO_2_ was reintroduced immediately after any perturbation to culture media while imaging. Care was taken to ensure the pH of the media is constant during all experiments with help on stage CO2 incubator.

4. In the current manuscript, the authors seemed to interpret that the detected changes in cell volume/area correspond to RVI or RVD. As described in the public review comment, this logical jump would not be acceptable. At least, they should evaluate cell volume regulation using well-known inhibitors, which have been characterized in the field of electrophysiology (e.g., flufenamate for RVI, NPPB or DCPIB for RVD; see such as Wehner, F. et al. FEBS Lett. 2003 and Ponce, A. et al. Cell Physiol. Biochem. 2012), to support their interpretations.

We have performed experiments with flufenamic acid (Figure 4B)

In addition, they should present a quantitative index for RVI/RVD, because several figures clearly show a difference in the enforced cell shrinkage/swelling (i.e., the difference between isotonic condition and the first time point after osmotic treatment). One of the commonly used indexes for RVI/RVD is the recovery rate from the first time point after osmotic treatment.

We used new metrics in the new manuscript (Figure 6C).

Moreover (possibly related to the aforementioned major point 1), the authors' usage of RVI and RVD (e.g., in line 307) raises a concern about their understanding of the terms of RVI and RVD. In general, RVI and RVD indicate the cell volume recovery followed by hypertonic stress and hypotonic stress, respectively. And, the cell volume change free from osmotic stress is not referred to as RVI or RVD. For instance, in the case of cell death, the accompanying cell volume change is differently termed as apoptosis volume decrease (AVD) even though the same effector molecule is involved in RVD and AVD (cf. Okada, Y. et al. J. Physiol. 2001).

We have modified the terminology wherever misused.

5. In addition to osmotic perturbation, the authors utilized alternative methods to operate intracellular MMC. However, the usage of Heclin would raise many concerns, as described in the public review comment. Better experimental design would be using a proteasome inhibitor (e.g., MG132, bortezomib) instead of Heclin and adding another sample group under both cycloheximide and proteasome inhibitor treatment.

We have performed the same experiments as Heclin with MG132 (Figure 4E, S4E revised manuscript).

As to heat shock, in addition to the potential concern described in the public review comment, the authors should deny/mention/discuss the possibility that the drastic decrease in protein mass by heat shock (Figure 2H) was caused not by the decrease in intracellular MMC but by the experimental procedures (e.g., the movement of aggregated proteins into an insoluble fraction) if they lysed cells according to the standard Bradford assay (line 137).

We confirmed loss of aggregated proteins during heat shock by using urea in the lysis buffer and find no change with our previous results (Figure 4E-ii revised manuscript).

6. In the Result section of Figure 3, the authors described region-dependent findings from representative images. However, they should describe them with quantified results. In particular, given the association between fluorescence anisotropy and cell area (e.g., Figure 5A), there is a concern that their observations (e.g., Figure 3A) are just led by not region-specific feature but cell size-dependent heterogeneity (e.g., the stage of the cell cycle). Hence, the quantified index should be also compared under the same cell size. Moreover, the quantified evaluation would potentially affect their speculation (lines 191-193); their speculation may be a possible explanation, but the above concern (i.e., cell size-derived result) seems more possible because Figure 3D shows a significant "decrease" in nocodazole from control.

We have investigated intracellular MMC with new metrics (Figure 2B).

7. In lines 218-221, the authors described they utilized paraformaldehyde to increase the effective size of MMC via fixation. First of all, related to the aforementioned major point 1, this assumption sentence must be supported by some results or citations. However, their assumption is likely wrong, because 4% paraformaldehyde is an extremely high hypertonic solution (approximately 1,500 mOsm). Although formaldehyde is a membrane-permeable molecule, they need to confirm that the effect of formaldehyde on cell volume is neglectable for supporting their assumption (as my "subjective" remark, a cell after 15-min formaldehyde fixation tends to be more flatter than a live cell).

We agree with the reviewer, however, we find that 4% PFA fixation does not alter the cell volume beyond ~25%. While a ~57% decrease.in cell volume during hypertonic treatment of 950 mOsm strength (additional 600mM dextrose to culture media) increases anisotropy values by ~20%, cell fixation reduces volume by ~25% but increases anisotropy values by ~22%. As neither treatment visibly alter the cell spread area, the effect of 4% PFA and hypertonic shock on cell volume manifests through cell height and is visible from the orthogonal view of the Z-stack images taken for volume measurement (Author response image 2). We have not used this data in the new manuscript.

**Author response image 2. sa2fig2:** Representative AiryScan images of NIH/3T3-EGFP showing the lateral (XY) view of the maximum intensity projection of Z-stack (step-size 180nm) and orthogonal (YZ) view upon fixation (**A**) and under hypertonic condition (**B**). Scale bars are 10 µm for XY view and 3 µm for YZ view. The cell depicted in (**A**) was imaged in 1xPBS before and after 30 minutes of fixation with 4% PFA (at 4°C). Cell volume measurement was performed, cells were removed from the microscope stage, fixed, and then the same cells were identified using fiduciary markers on the petridish and microscope stage, and cell volume was measured again. The cell depicted in (**B**) was imaged before and after 30 minutes incubation in hypertonic culture media containing an excess of 600 mM dextrose at 37°C, 5% CO2.

Moreover, even under their assumption, EGFP is also cross-linked with intracellular biomolecules. Although it is common sense that EGFP keeps the ability to be excited and emit fluorescence even after fixation, they should check that the EGFP fluorescence anisotropy is not affected by the cross-link of EGFP; otherwise, they cannot compare the anisotropy between live cells and fixed cells.

There is precedence in the observation that fixation alters the fluorescence lifetime and anisotropy of fluorescent proteins (Ganguly et al., 2011). This is in line with our claims of the manuscript. Cross-linking increases local crowding around EGFP, thus increasing the refractive index significantly. We have not used this data in the revised manuscript.

8. In Figure 4D and S4, the authors interpreted that the results indicated "protein aggregation", but they must clearly describe the judgement criteria and reason(s). Although the current manuscript has insufficient information, one of them may be based on the observation of the local increase in fluorescence intensity. If so, however, it is still unclear why they can deny the possibility that the intensity increase just corresponds to the local increase in "free" (i.e., not aggregated) fluorophore concentration. The same thing is applied to "DNA condensation" (line 235).

We performed photobleaching to analyze the EGFP clusters in the new manuscript (Figure 5D). These results indicate that a local increase in "free" (i.e., not aggregated) fluorophore concentration leads to the observation. Thus we have revised our claim.

9. In the Result section about Figure 5 and S5, the authors described "cell spreading", but what this terminology exactly indicates is unclear. Given that they mentioned about cell cycle in lines 245-246, this term may indicate the cells in G2 phase. However, no method for cycle synchronization is described in the Method section.

We have changed the term “Cell spreading activates RVI”. In the old manuscript, “Cell spreading” refers to spreading after seeding (~1-2 hours) and not due to cell cycle dependent spreading (10-12 hours timescale for NIH/3T3). We wanted to describe the simultaneous increase of cell spread area and cell volume post seeding due to adhesion. We have not included the data in the revised manuscript.

10. Based on the lines 251-253, 258-259, and others, the authors seem to think that a "sensor" must be regulated to be active after its functioning. However, this thought is not logical at all. Although this kind of feedback mechanism may be good for cellular homeostasis, there is no logical necessity for the "sensor" to be managed even after its functioning (i.e., sensing). In other words, the sensor molecule is not necessarily the same as the effector molecule. Additionally, related to "sensor", the authors should clearly describe the definition of "sensor" terminology, for example, in the Introduction section. Based on lines 292-293, they seem to consider their own definition for "sensor", whose candidate can include osmolyte. This is not so general; e.g., in the case of RVD, VRAC (LRRC8s) is said as a sensing molecule, and small osmolyte (e.g., chloride ion, glutamate) is a kind of the second messenger after VRAC activation (i.e., sensing). Likewise, based on lines 284-285, the authors seem to categorize only two modes for "sensor", while a famous review in this field (Hoffman, E. K. et al. Physiol. Rev. 2009, which is cited even in this manuscript) categorized three modes for the sensing principle. Of course, scientists can define their own terminology, but they must clearly describe it to avoid readers' confusion with the previous papers.

We agree with the reviewer’s comment. The cell volume sensor need not be necessarily regulated beyond the initial “sensing”. However, since intracellular MMC and cell volume is inversely related, it is difficult to say when the “sensor” that is MMC decouples from the “effectors” whose signaling regulate cell volume during processes such as RVI/RVD. We have rephrased the terminologies to make our manuscript more easily understandable for the readers.

Reviewer #3:Biswas et al., are interested in measuring the level of macromolecular crowding in mammalian cells. Macromolecular crowding has been shown to have a broad impact on various cellular functions, such as enzymatic reaction rates, cytoskeleton dynamics, and phase transition to name a few. Despite these observations, there are a lot of unanswered fundamental questions regarding macromolecular crowding. For example, it is unknown if cells can sense the level of crowding in the cytoplasm or what mechanisms they could use to regulate the crowding level. These are the questions the authors are trying to answer. These questions are of general interest to cell biologists and answering them would represent an advance in the fields of cell biology and biophysics. The main result from the study is that cells lack a homeostasis mechanism for macromolecular crowding however given the technical concerns it is unclear if this conclusion is supported by the data.Strengths:- The paper's ambitions are built on unknown from the literature and because macromolecular crowding is such a general concept with broad implications the potential conclusions from the paper are of interest to a large audience.- The authors assess the universality of their observation by comparing multiple mammalian cell lines. This is an interesting approach as every cell line has a potentially different level of molecular crowding as well as different correction mechanisms.- The authors try to test their hypothesis by using various methods to affect intracellular crowding levels.Weaknesses:- The authors use steady-state fluorescence anisotropy measurement to evaluate the level of macromolecular crowding in cells. Although it is a method used to assay the fractions of bounds and unbound protein it is not traditionally used to estimate crowding. The method needs to be benchmarked before conclusions can be reached about the meaning of the observed changes in anisotropy. As this is the only method used by the authors to estimate crowding level many of the conclusions are pending validation of the method.- The figures and especially the figure legends suffer from a lack of clarity. In some of them, the reader is left to assume what some of the plots describe.- The authors fail to put their observation in the context of the available literature making it harder for the reader to evaluate.Recommendations for the authors:I have a lot of doubts about the anisotropy method used by the authors.1) I am not aware of any other paper using steady-state fluorescence anisotropy to measure crowding in cells and the authors make it sound like they are trying to establish the method for that usage. The method is usually used to evaluate binding affinity, etc. Can the authors provide a rationale as to how and why steady-state anisotropy should be reporting on crowding?

The rationale is explained in the Materials and methods section in the revised manuscript. The intracellular environment is crowded with macromolecules, and the change in crowding of macromolecules changes the intracellular refractive index. The extent of increase of refractive index also depends on the dielectric properties of the crowder molecules. Hence, quantifying the refractive index could enable one to gauge the macromolecular crowding (MMC) level in the cytoplasm. The elevated cytoplasmic refractive index alters the fluorescence lifetime of EGFP from the values measured in salt buffers. The effect of refractive index (*n*) on the intrinsic fluorescence lifetime (τ_0_) of EGFP can be understood from the Strickler-Berg equation (Strickler & Berg, 1962)-1τ0=2.88x10−9n2∫I(ϑ)dϑ∫I(ϑ)v−3dϑ∫ε(ϑ)ϑdϑ

Here, *I*(*ϑ*) and *ε*(*ϑ*) represents the fluorescence emission spectrum and the molar extinction coefficient of the molecule for a wavenumber *ϑ*. Thus essentially, 1τ0∝n2 when all other factors are preserved. Altering the MMC conditions changes the intracellular refractive index and, subsequently, the lifetime of EGFP. Now, since the Perrin equation for fluorescence anisotropy states that r=r01+τθc therefore r∝1τ0∝n2. For EGFP, *τ* is ~2.6 ns (Suhling et al., 2002) and *θC* is ~14 ns in water (Novikov et al., 2017). Thus, 1τ0 for EGFP is 0.19 in water. Increased MMC decreases *τ* and increases *θC*, thus increasing the value of *r*. Further, *r*0 itself could also be affected by changes in *n* due to changes in the angle of the chromophore’s transition dipole moment. Our measurement establishes that crowding alters *r*_0_ too (Figure S1 B) Hence, fluorescence anisotropy of EGFP (*r_EGFP_*) could be used to probe the cytoplasmic MMC. While direct fluorescence lifetime imaging microscopy has been used to study intracellular protein densities (Levchenko et al., 2017; Pliss et al., 2019), we used steady state fluorescence anisotropy imaging, which requires less complicated instrumentation and has a higher throughput than measuring fluorescence lifetime with confocal microscope systems. Through a series of calibration experiments, we show that *r_EGFP_* is a reliable probe for measuring intracellular MMC, particularly by protein crowders. We show that *r_EGFP_* has a high dynamic range, is practically insensitive to intracellular pH, and can rapidly sense the changes in MMC conditions during various cellular processes.

2) In figure 1 A, the BSA calibration curve shows an increase in anisotropy of around 0.02 from 0 to 5 mM of BSA. How should a change in anisotropy of 0.02 be interpreted? A solution of 5mM BSA should be very crowded, yet the readout only changes by 0.02 from no crowding to 5mM. Perturbations on cells should change crowding by less than that, meaning that the readout of these perturbations would be less than 0.02, probably even less than 0.01. Can such a small change be reliably detected? What is the error in the measurement? How does it compare to biological noise?

Error bars in Figure 1A represents the standard deviation (~0.006). We get the standard error of mean to be ~0.002. Thus, the minimum measurable change in anisotropy is ~0.002. Therefore, 0.02 is a significant change in anisotropy values. The significance of measurable change in anisotropy can be verified from literature (Goswami et al., 2008; Koskinen & Hotulainen, 2014; Vishwasrao et al., 2012).

3) In figure 1 the authors use in vitro measurement to try and calibrate the method. In figure 1 A the authors report that anisotropy increases with increasing BSA concentrations in the range from 0 to 5 mM of BSA. In figure 1 E the authors report that anisotropy does not change when eGFP concentration is increased from 0 to 9 mM. Yet, BSA and eGFP are both proteins and similar(ish) in size. So, can the authors explain why eGFP anisotropy is sensitive to BSA concentration but not to its own concentration? Increasing eGFP concentration from 0 to 9 mM should increase the crowding of the solution and be picked up by the method.

Since, this was the first point raised by another esteemed reviewer, we will repeat our statement. The X-axis labels in Figure 1E mentioning the unit of EGFP concentration is a typographical error. We apologize profusely for our mistake and all the confusion. The maximum concentration of EGFP used for experiments described in Figure 1E should be 9 μM, as per our 488 nm absorbance measurements, and not 9 mM as indicated in the current manuscript. The likely cause of the typo was a character conversion failure by our Origin graph plotting software (from English ‘m’ to Greek ‘μ’), and we overlooked the error. We have re-evaluated the concentration of EGFP (denoted as [EGFP]) by FCS (Fluorescence Correlation Spectroscopy), which puts the maximum [EGFP] in Figure 1E at ~540 μM. The new manuscript contains the re-evaluated calculation in Figure 1F. To validate the absence of homo-FRET in our intracellular *r_EGFP_* measurements, we performed photobleaching experiments in EGFP (monomer) and 2GFP (EGFP dimer) expressing NIH/3T3, explained in Figure S2B in the new manuscript.

4) Do the authors know the concentration of eGFP proteins in the cells they are imaging? Steady-state anisotropy could be sensitive to homo-FRET. Can the authors rule out that this contributes to what they are measuring?

We used 50 nM EGFP for in our in vitro experiments. We have measured the distribution of intracellular EGFP concentration using FCS (Figure S2C). The intracellular concentration of cellular EGFP used for MMC estimation is below the homo-FRET condition for 90% of NIH/3T3 cells.

5) Images from figure 2 A, figure 3 A, C and D, figure 4 C, as well as Sup. Figure C and G, Sup. Figure 3 B and Sup. Figure 5 A, B, C shows that the anisotropy signal reported scale with cell thickness. The signal clearly appears higher around the center of the cell, where the cell is the thickest, and lower at the edge. Can the authors explain why this striking correlation seems to be present in all the images? To be useful the method needs to be insensitive to cell thickness. Can the author provide data showing that this is the case?

We agree with the reviewer that the vertical thickness of cells interfere with anisotropy measurements. In the revised manuscript, we have explained in Figure S3. We have further demonstrated that taking the modal value of *rEGFP* provides a reliable indicator of the average intracellular MMC (Figure 2B).

Other experiments also have problems.6) Mammalian cell volume is notably hard to measure. Even if the author used a super-resolution microscope this kind of super-resolution mostly increases the XY resolution not much the resolution in the Z-direction and the authors seem to have used a simple binarization method. Can the authors provide a comparison to measurements by others and/or use another method?

We agree that AiryScan super-resolution does not improve cell volume measurement accuracy. However, AiryScan processing reduces the file sizes of the Z-stack images and also reduces pixel noise in the images which enables easier and more accurate intensity-based binarization, which is useful for the ImageJ macro written for measuring cell volume. In the work of Guo et al., 2017, the authors have compared cell volume measurements by standard confocal microscopy and SIM super-resolution microscopy and have found no discernible difference. Since cell lines used in different labs may vary (depending on the source of acquisition, passage stages, culture media source), it is difficult to compare our data with others’. Exemplarily, the median cell volume of 3T3 fibroblasts measured by Venkova et al. is ~1500 μm3, while the median cell volume for our NIH/3T3 fibroblasts is ~3200 μm3. The Z-resolution of our confocal microscopy is of the order of 1μm (63x/1.4NA). This could overestimate the height by a maximum of 0.5 μm in comparison to Venkova et al. The XY-resolution of our volume measurement method is of the order of 0.35 μm, and thus better than the Fluorescence Exclusion method by Venkova et al., which uses a lower magnification objective. Thus, we assume the difference in cell volumes is not because of measurement error. Given the pyramid-like geometry of adherent cells, with a large base and gradually constricting apex, the error in height estimation will have negligible effect on cell volume estimation as the visible cell area decreases with cell height. Moreover, the median cell spread area also differs significantly from the measurements of Venkova et al., where they ingeniously employed Interference Reflection microscopy to measure cell area, while we used AiryScan imaging of EGFP expressing cells. Such difference can exist only if the cell lines used for volume/area measurement are phenotypically different. We are also attempting to employ cell volume measurement methods alternative to Z-stack imaging as illustrated in the works of Cadart et al., 2017, 2018; and Model, 2015 to verify our data.

A comparison of the data in figure 5 on spreading area vs volume with the data from others, such as Venkova et al. 2022, seems necessary. Especially because some of the conclusions drawn by the authors disagree with previous observations.

Our finding on area volume relation (Figure 5 in manuscript) matches with findings of Venkova et al., 2022, (Figure 1—figure supplement 1 -C). Since we did not explore the cause of correlation, we could not ascribe the correlation to its source. At the time of preparing the old version of the manuscript, we were not aware of Venkova et al. We have not included this data in the revised manuscript.

7) Judging from the plot (because that information is missing from the methods) the FRAP experiment was performed at 2 frames per second. The resulting recovery curves do not have enough points to properly recover the exponential phase of the recovery. In fact, judging by the plot in Figure 3B, the authors have only 1 data point before the plateau. How can a rate be accurately recovered from these data? Same concern in figure 5E.

The FRAP experiments were performed at 2 frames/sec. All the points of FRAP recovery data were not shown by using a function of Ori gin Pro graphing software called “skip points”. The FRAP data was shown in “line+symbol” with skipped points = 50. This was done to reduce clutter in the graphs. We have upgraded our software and we include a reformatted example of Figure 3B (i) from the original manuscript which is now included in the revised manuscript (Figure 3C1).

I also have some more general comments.8) There are in general a lot of problems with the figures and figure legends. Multiple plots are poorly described. For example, Figures 2 B and C, there is nothing on the figure or in the legend indicating which curves are the area and which are the anisotropy values. In Figure 3 D there is no information about the statistical test used. In Figure 5, the scatters plots have what I have to assume are linear regressions, yet nothing is said about how the lines were obtained and if statistical tests were used.

We apologize for these mistakes and we have addressed all the concerns in the revised manuscript.

9) In general, the figures and figure legends are far from what the journal guidelines are regarding the reporting of data. For example, the figure legends do not report any sample size (number of cells, number of repetitions, etc…).

We have addressed all the concerns in the revised manuscript.

[Editors’ note: what follows is the authors’ response to the second round of review.]

The manuscript has been improved but there are some remaining issues that need to be addressed, as outlined below.Reviewer #1:The authors have substantially improved the characterization of the use of fluorescence anisotropy of cytoplasmic GFP as a measure of macromolecular crowding (MMC). Their analysis of the change in anisotropy now correlates with the changes in refractive index of the medium, consistent with theoretical expectations. Therefore changes in the anisotropy as measured in a homogenous environment will reflect changes attributable to changes in MMC. They have addressed and also extended the comments raised in the initial review of their manuscript to provide adequate experimental and theoretical justification for the use of this tool.Using this tool, they seek to relate the observations they make with the measure of MMC and its relationship to changes in cell volume and its control. However in this context there are many lacunae which chiefly concern the conflation of the use of the tool as a surrogate measure of cell volume, need to be addressed.Recommendations for the authorsTo relate the observations they make with the measure of MMC and its relationship to changes in cell volume and its control, substantial explanation and additional information is necessary.Specific concerns:1) Cell spreading and MMC: lines 202-302: The authors first try to explain the differences between the perinuclear and lamellipodial r EGFP values. They then show that the values that they measure in a widefield microscope are distorted because of the small thickness of the lamellipodia and that a confocal microscope gives less variable measurements between the lamellipodia and the cytosolic anisotropy. This calls into question the utility of this technique to measure anything that is spatially varying across a cell and all the observations about spatial homogenization upon various treatments could simply be an artifact of the measurement. Even comparisons between spread cells and rounded cells are not clearly measuring anisotropy but also will have information about the geometry of the cell.

We agree with the reviewer that it is difficult to interpret rEGFP values when the cellular MMC is spatially heterogeneous, and we mentioned the issue in the Discussion section (lines 520-525 in the original manuscript, 536-552 in the revised manuscript). It is non-trivial to decouple the contributions of the characteristic cellular MMC and the media autofluorescence at the varying heights of a cell in the rEGFP maps. This creates problem in finding the characteristic cellular MMC using the arithmetic mean value of intracellular rEGFP maps. The advantage of using the modal rEGFP value (instead of the arithmetic mean rEGFP value) is a more accurate depiction of the characteristic MMC, despite the spatial heterogeneity. To demonstrate, we highlight the pixels corresponding to the modal rEGFP in three different MDA-MB-231 cells which have highly variable cell geometry and intracellular MMC (Author response image 3). In Author response image 3, the pixels corresponding to the modal value of rEGFP in each cell have been displayed in the respective colors of the cell boundaries. In the cell depicted with magenta, the pixels having modal rEGFP values are absent in the lamellar region despite its enormous size. Additionally, the comparatively high rEGFP values in the nuclear regions are also skipped. To strengthen our claim, we compared the mean rEGFP and modal rEGFP of the same cell (in magenta) by arbitrarily cropping out varying amounts of the lamellar region to represent cells of different morphologies (Author response image 3). The associated quantification shows that while the mean rEGFP changes with increasing cropping (panels 1-3), the modal rEGFP is constant despite the cropping, and thus, is insensitive to spatial variabilities of MMC or height artifacts. Further in hypertonic conditions, where the cell geometry does not vary, the modal rEGFP depicts the MMC changes consistently (Figure 2B and revised manuscript’s Figure 2 —figure supplement 1B). Thus, using modal rEGFP of a cell we can compare the characteristic intracellular MMC irrespective of the changes in cell geometry.

**Author response image 3. sa2fig3:** (**A**) MDA-MB-231 cells expressing EGFP. (i) EGFP expression, (ii) rEGFP map, and (iii) highlighted pixels having modal rEGFP values with cell outline in corresponding colours. (**B**) The row of images depicts the cell in magenta with the lamellar region progressively cropped out. The associated plot compares the mean and mode of the rEGFP maps for the cropped cell sections: 1 (uncropped), 2, and 3.

In line 292: The authors claim that the cell volume increases upon cell spreading. This is contrary to what is seen in recent papers making similar measurements in similar ways with the same cell lines. This should be addressed and acknowledged. Ref: https://doi.org/10.1016/j.bpj.2017.11.3785, https://doi.org/10.1073/pnas.1705179114

We are aware that our data in Figure S3G (Figure 4A-ii in revised manuscript) contradicts the mentioned references. Currently, we can only ascribe this contradiction to differing experimental conditions, like using a fluorescent dye to visualize the protoplasm or cell membrane as the previous reports, while we use cellular EGFP expression without externally perturbing the cytosol. In Supplementary Figure 1D of Venkova et al., 2022 (https://doi.org/10.7554/*eLife*.72381), we can see that in HeLa cells, the average volume starts increasing 20 minutes post cell seeding, which agrees with our observations. We have acknowledged and discussed the contradictory observations while citing the references mentioned above in the Discussion section of the revised manuscript (lines 561-567).

2) Regulatory volume increase: lines 303-381: Here the authors use the r EGFP values to infer regulatory volume increase. The r EGFP value as an indicator of cell volume has not been clearly established earlier. The authors have also seen cases where depolymerizing microtubules and vimentin increases volume but does not affect r EGFP. Hence all the results of this section can only be interpreted as a change in the cytoplasmic MMC not as a change in the volume. And there needs to be an independent cell volume measurement for each treatment where regulatory volume increase is inferred.

We apologize for describing cellular MMC-restoration upon hypertonicity-induction as RVI. We did not try to establish rEGFP as a cell volume indicator, instead, we intend to claim that the changes in rEGFP observed are a consequence of RVI. Also complained by our 2nd reviewer, we realized that interchangeably using RVI and MMC-restoration has made the text confusing and scientifically incorrect. Hence, we have replaced “RVI” with “MMC-recovery” or similar phrases wherever we are not measuring cell volume in the revised manuscript. Cell volume and MMC should be qualitatively reciprocal to each other, and we found that cellular MMC and volume scale during RVI (Figure S4D, Figure 5A in revised manuscript), as per expectation. In Figure 5B (revised Figure 8B), we also simultaneously measured rEGFP and cell volume during RVI. We wanted to highlight that in certain cases, like actin depolymerization, cellular MMC increases while volume remains unchanged, while in the cases of microtubule and vimentin depolymerization, cell volume increases while MMC remains unchanged. Therefore, only in conditions where the number of intracellular crowders and the excluded volume fraction of the cytoplasm remains unchanged, rEGFP scales as cell volume. As polymer to monomer transition always leads to a change in crowder number and excluded volume (https://doi.org/10.1016/S1089-3156(98)00056-7, https://doi.org/10.1083/jcb.200609066), rEGFP does not correspond to cell volume during such conditions. We have also independently measured cell volume and MMC in multiple cases where we expected a change in the cytoplasmic crowder numbers (e.g., cytoskeletal depolymerizations, cell spreading, proteostasis disruptions, RVI).

3) Concentration versus anisotropy – lines 382:435: The authors try to provide an explanation for the deviation of the concentration vs anisotropy curve in differing cellular volume conditions, despite the cells having the same mass. They end up speculating on the role of phase separation as a mechanism for exclusion of the GFP tracer. The relevance of this section to the paper is not clear and only serves to remind the reader that the measured GFP anisotropy value is a complex function of cellular parameters and cannot be explained by simple volume scaling arguments.

We agree with the reviewer that rEGFP is a complex function which does not always correspond to the cell volume, as in the cases of cytoskeletal depolymerizations. The primary purpose for exploring the concentration-dilution law was an attempt to answer the question- whether rEGFP reflects changes in cell volume during hypertonic conditions, coincidentally asked by the reviewer in the previous question. We found that rEGFP only scales qualitatively with cell volume during RVI (Figure 8A revised manuscript). When cell mass is unchanging, cell volume changes must correspond to changes in macromolecular concentrations. If rEGFP was an accurate reporter of cellular MMC, rEGFP vs cell volume should follow the concentration dilution law as depicted in Figure 5A-i, 5B (Figure 8A-B revised manuscript). However, we found that rEGFP shows a higher-than-expected value, and thus, either the MMC in the cell does not follow the concentration-dilution law, or rEGFP is not reporting MMC truthfully. Given our in-vitro data, rEGFP scales linearly with crowder concentration (Figure 1A) and follows the Strickler-Berg relation (Figure 1 —figure supplement 1D), so we can presume that rEGFP truthfully reports MMC. Photobleaching studies showed that EGFP diffuses slowly through interstitial spaces that exclude EGFP (Figure 5D, revised Figure 8D). Since Airyscan images of intracellular EGFP showed granular structures, we speculated the existence of the EGFP Excluded Cytoplasmic Volume (EECV) which confines EGFP molecules to narrow spaces. In an attempt to characterize EECVs, we envisaged that hypertonicity-induced phase separation creates EECVs, and the phase separated cytoplasm elevates rEGFP above the expected values, and thus the deviation from the concentration dilution law. We suppose that other methods of measuring crowding like fluorescence lifetime would also suffer from these issues as they originate from the changes in refractive index. We thank the reviewer for asking the question and pointing out the lack of explanation in the text. We have elaborated the relevance in Figure 8 of the revised manuscript.

4) The role of TFNR1: lines 436-508: Here the authors argue that hypotonicity induced p65 migration to the nucleus is mediated by TFNR1 receptor activation. They show that the effector RIPK1's association with TFNR1 is lesser in hyperosmotic shock conditions. They correlate the loss of volume regulation upon TFNR1 perturbation, with reduced p65 nuclear migration and RIPK1 recruitment to implicate TFNR1 directly in Regulatory Volume Increase. However, these separate events need not be connected and the causal role of TFNR1 in hypertonic stress induced cell volume regulation needs to be established more clearly. The fact that TFNR1 receptor perturbations lead to the complete downregulation of the instantaneous volume reduction in TFNR1kd and Zafirlukast as seen in figure 6C is very surprising. This also indicates that TFNR1 may be doing something very different and changing the set point of the cell even before the osmotic shock is applied. In figure 6C and 6D it is again clear that the cell volume measurements do not correlate with anisotropy values again underlining that r EGFP is not a surrogate measure of cellular volume.

We agree with the reviewer that the cell volume-regulatory role of TNFR1 can be elucidated further and is a subject for our future studies. Our present conclusions are based on the observations that [1] cells show RVI during moderate hypertonicity (150 mM) but not severe hypertonicity (600 mM), [2] moderate hypertonicity induces nuclear p65 translocation, but not severe hypertonicity, and [3] downregulating TNFR1 activity via Zafirlukast or knockdown, or NFkB inhibition by CAY10512 prevents hypertonicity-induced p65 nuclear translocation and prevents RVI. Thus, our data show that TNFR1 is necessary for initiating cellular RVI. RIPK1 recruitment and ubiquitination is a critical factor that promotes p65 nuclear translocation and determines the cell fate upon TNFR1 activation (Figure 3 in https://doi.org/10.1038/s41580-023-00623-w). We show that the increased cytoplasmic microviscosity at severe hypertonicities impairs RIPK1 recruitment, thus delaying p65 nuclear translocation, and consequently RVI. We have now included the TNFR1-inhibited cell volume and MMC data in the revised manuscript. Inhibiting TNFR1-signalling resulted in three different phenomena: [1] a reduction in cell volume (Figure 6C) and elevation of MMC (Figure S6C), [2] a reduced rate of hypertonicity induced cell shrinkage (Figure S7A), and [3] a loss of cellular RVI (Figure 7A) and MMC recovery ability (Figure 7B). We suspect that these three phenomena are not interconnected since cellular RVI requires NFkB activation which is absent under TNFR1 inactivation. Using the drug CAY10512 to inhibit NFkB activation, we observe a similar loss of RVI (Figure 7A), although the hypertonic cell shrinkage is analogous to untreated conditions (Figure 7 —figure supplement 1A). Hypertonic cell shrinkage has been shown to be linked to membrane aquaporin levels, and inhibiting aquaporin activity enables cells to resist hypertonic volume shrinkage (https://doi.org/10.1074/jbc.M008760200, https://doi.org/10.1128%2FJB.01665-12). Hence, we are exploring the role of aquaporins in volume shrinkage and RVI as future studies. Furthermore, we would like to draw the reviewer’s attention to Figure 2A of the article Liu et al., 2022 (https://doi.org/10.1038%2Fs41598-022-18630-w), where the authors also attempted to investigate the role of TNFR1 in RVI as an upstream activator of p38 MAPK, a well-known osmosensor connected to the functioning of NKCC and NHE1 ion transporters. The authors surprisingly find that hypertonicity-induced p38 phosphorylation is higher in TNFR1-knockdown cells than wild type cells. As per numerous previous reports, a downregulation of TNFR1 is expected to decrease phosphorylated p38 levels (https://doi.org/10.1038/s41467-018-03640-y). Currently, we can only speculate that the elevated phospho-p38 levels and other proteins described in Liu et al. affect NKCC, NHE1, or similar ion channels that result in a reduction of the isotonic cell volume.

Additionally, p38 phosphorylation is known to amplify NFkB activity (although shown in different cell lines-https://doi.org/10.4049%2Fjimmunol.179.10.7101), hence, according to Liu et al. 2022, hypertonicity induced NFkB activation should be higher in TNFR1-KD cells, which is evidently not true (Figure 6A-B). Thus, although NFkB has numerous upstream activators in the cell for different conditions, it should be safe to assume that p38 is not one of them during hypertonic stress. Therefore, compiling our observations, we believe that the TNFR1-mediated NFkB signaling is directly responsible for activating RVI.

Reviewer #2:The authors drastically restructured their manuscript, which resulted in the great improvement as a scientific paper. Moreover, they performed additional experiments, and extended their findings to more concrete biological context (e.g., the involvement of TNFR1). Overall, I think they addressed most of my concerns, and thus I am positive for their main claims. At the same time, their new results raised additional points to be addressed.1. I thank the authors for their in vitro experiment with additional crowding agents such as Ficoll and polyethylene glycol (Figure 1A). However, the result was a bit different from what I originally expected, i.e., I originally assumed that these crowding agents would show the similar EGFP fluorescence anisotropy to BSA, but the truth was that "BSA has the most prominent impact" (line 102). And the authors interpret that the EGFP fluorescence anisotropy is primarily for protein crowding (e.g., line 140, line 513). I agree that this new finding is quite interesting. At the same time, my original intention is not resolved yet, i.e., they cannot deny that their findings are specific to BSA (e.g., amino acid sequence), rather than general proteins. Hence, they should add another protein at least. Also, it may be one option to specifically claim protein crowding (instead of general macromolecular crowding) in Title and Abstract.

As per the reviewer’s suggestion, we measured rEGFP in different concentrations of another protein: α-Lactalbumin (molecular weight = 14 kDa) (revised Figure 1A). Given the current data, the molar mass of the molecule determines the slope of rEGFP vs concentration. Solution refractive index (n) is the primary factor that determines rEGFP, and according to the Lorentz-Lorentz equation, n is determined by the solute number density and mean polarizability of the solute molecule. The mean polarizability is dependent on the number of atoms in the molecule if the atomic polarizabilities are comparable, so naturally proteins, being the larger macromolecule species, have a greater influence on rEGFP. Even among proteins, the molar mass determines the mean polarizability (https://doi.org/10.1038/s41598-022-05586-0). Thus, although rEGFP is more sensitive to proteins compared to other macromolecules, rEGFP is still a macromolecule concentration sensor. Hence, we believe our terminology of “macromolecular crowding sensor” is justified.

2. The authors largely improved the usage of regulatory volume increase (RVI) throughout the revised manuscript. However, they still use RVI for the result descriptions of Figure 4A-C ("the best RVI" in line 313, "partial RVI" and "RVI" in line 314, "RVI" in 315, "early RVI" in line 330, "RVI" in line 340, "RVI" in 349, etc). Their main claim is that the EGFP fluorescence anisotropy is not a RVI indicator but a protein crowding indicator. Hence, they should describe exactly what the result indicates (e.g., decrease/recovery from the ∆r_{EGFP} increase). Additionally, based on their introductory narrative (lines 304-310), I think that Figure S4D and its result sentence (lines 351-352) fit better than Figure 4A as the following result. I recommend them to revise the current Figure S4D as the new Figure 4A.

We realize our mistake in using the RVI terminology erroneously, and we have revised the text and figures according to the reviewer’s suggestion.

3. In contrast to the previous version, the authors eliminated regulatory volume decrease (RVD). However, they observed "dilution of MMC" (line 338) under hypotonicity (Figure 4C-i). I'm not sure due to the lack of recovery rate presentation here, but the trajectory seems to show that the hypotonicity-induced ∆r_{EGFP} decrease did not recover even after dozen minutes. Because cells efflux osmolytes to achieve RVD, this result may indicate that MMC decreased while cell volume recovered, i.e., the hypotonic version of the deviation from theory (Figure 5A). Hence, as well as the hypertonicity results (Figure S4D, 4B), the authors should compare the changes between ∆r_{EGFP} and cell volume change under hypotonic conditions.

In the revised Figure 5 —figure supplement 1A-B, we have included the RVD data. The ΔrEGFP for hypotonic stress recovers slightly within 2 minutes for NIH/3T3 and then continuously decreases until plateauing from 20-40 mins, and finally rises back to isotonic levels at ~2 h. Since we were unsure how accurate our cell volume estimations would be in the sub-minute timescales of the initial recovery, we did not measure the volume of the cells but quantified the cell outline area (which is an approximate projection of cell swelling and recovery). In the timescales of hours, efflux of osmolytes like taurine could indeed restore the cellular MMC. However, the cell morphology changed drastically in 50% hypotonicity for 2 hours, so we could not confirm RVD with enough accuracy. We could investigate RVD with a faster cell volume measurement technique like fluorescence exclusion microscopy or transmission through dye microscopy techniques (https://doi.org/10.1111/jmi.12929), however, the required microfluidic devices are currently not available to us.

4. The authors describe that the decrease from the ∆r_{EGFP} increase was "faster in the case of 100 mOsm change (comparing 50 mM NaCl and 100 mM mannitol/dextrose)" (lines 346-348). However, this description does not match with the results (Figure S4B, S4C), because we cannot recognize the differences at a glance especially due to the large SDs in Figure S4C. Similar to cell volume change, the authors should present statistical test(s) for recovery rates.

In the revised Figure 5 —figure supplement 1E, we have compared the MMC kinetics under mannitol, dextrose, and NaCl in the same graph for easier visualization of the recovery rates. We claimed a faster rate of MMC recovery since the NaCl^-^mediated MMC elevation is restored within 10 mins of hypertonicity induction while mannitol/dextrose-mediated MMC elevation is restored by NIH/3T3 cells in 30 mins.

5. In all multiple hypothesis testing results (e.g., Figure 2E), the authors must describe how they adjusted P-values. Nominal P-value may be valid, depending on their null hypothesis. However, based on their approach (line 846), P-value adjustment (e.g., Bonferroni method) would be required for their interpretations.

The p-value was calculated based on the null hypothesis that the samples belong to the same population at the α-value of 0.05. We re-adjusted the α-values by Bonferroni method and compared the p-values and have provided the description at individual figures and in the “Statistical Analysis” section.

6. As to Figure 4D, the authors observed that the difference in the EGFP diffusion rate between nucleus and cytoplasm diminished under hypertonic condition (lines 355-357). However, the most important finding of Figure 4D is that "hypertonicity-induced elevation of intracellular MMC concomitantly increases the cellular micro viscosity for both cytoplasmic and nucleoplasmic proteins" (lines 357-359). Hence, the authors should perform statistical tests for the differences not between cytoplasm vs. nucleus per mannitol concentration but between 0 mM vs. each concentration of mannitol per cytoplasm/nucleus.

In the revised Figure 5E, we have made the recommended changes.

Moreover, Figure 4D seems a different topic from the other Figure 4 panels. Rather, Figure 4D and its description (lines 353-359) would fit better at the point before Figure 5D and its description (line 409).

The FRAP analysis (revised Figure 5E) was provided as a visualization tool for MMC-induced microviscosity changes to accompany the MMC changes in revised Figure 5D. As another reviewer has also requested for the microviscosity values at different hypertonicities, we have chosen to keep the FRAP analysis in the same place.

7. In Figure 5D, the control comparison (i.e., isoosmotic condition) is required.

We have provided the isosmotic condition in the revised Figure 8D.

8. I completely disagree that the authors use "LLPS" from their results (e.g., line 424, line 426, line 428), because they did not investigate the material property of the p65 "condensates". In other words, they cannot deny the possibility that the p65 granules/structures are solid-like condensates or aggregates. At the same time, I believe that they do not intend to claim that p65 granules/structures under hypertonic conditions are novel LLPS-induced liquid-phase droplets, and thus they could resolve this point by wording. Otherwise, they need to perform multiple gold-standard experiments of the LLPS studies (e.g., FRAP, in vitro reconstruction).

We agree with the reviewer that without probing the material properties, it is incorrect to assume that the observed p65 granules are “liquid”. We have rephrased the terminology to “granules” or similar terminology in the revised manuscript. We speculated the existence of LLPS as the granular structures of the NFkB subunit p65 appeared with hypertonicity induction and disappeared upon rescue (revised Figure 8 —figure supplement 1B), and because p65 exhibits the typical properties of proteins that undergo LLPS (like multivalency, intrinsically disordered domains, DNA binding capacity:- https://doi.org/10.1016/j.jbc.2022.102349, https://doi.org/10.1096/fj.201801638R, https://doi.org/10.1042%2FBST20230035).

Reviewer #3:The manuscript presents a wide range of data, from in vitro characterization of their methods, to in vivo measurement of crowding either via established methods (FRAP, FLIM or FCS) and a comparison to the measurement of fluorescence anisotropy (FA for short) of eGFP.Strengths:While FA increases linearly under certain in vitro conditions, it only remains a proxy for crowding in vivo, where the scaling is unsure but with a likely increase with crowding. As such, this method is interesting as it would be easy to implement in a lab and could provide an easy way to estimate changes in crowding under various physico-chemical conditions, or even intracellularly. The method has been carefully established in vitro and compared to well-established methods. in vivo, hyperosmotic stresses, known to increase crowding, have been shown to increase FA in 4 different cell lines.Weaknesses:I have noted several weaknesses which I summarize in the five points below:1. Even if the change of FA was carefully studied in vitro, the choice of chemical conditions can affect the response and can be far from the in vivo reality. In particular, one point that is unclear to me is the conclusion of the authors on the absence of dependency of reGFP on micro-viscosity. The authors based their conclusion on the effect of the variation of glycerol concentration from 80% (v/v) to 90% (v/v) for which they indeed observed only a mild increase in FA. However, what is the rationale for starting at 80% (v/v)? This concentration is already very high, with a solution micro-viscosity that is more than 20 times the one of water, thus larger than the typical micro-viscosity inside a cell. It can thus be possible that there is an effect of micro-viscosity, at lower values. One result that would argue for this is the fact that at 80% (v/v) glycerol, reGFP is already much higher (0.24) than the highest value measured at 4 mM BSA (0.22), or the condition in water (0.20). As such, the data presented by the authors do not convincingly show that reGFP is independent of micro-viscosity. Moreover, based on their rationale of the effect of crowding on reGFP on rotational diffusion of the molecule, I would find it odd that micro-viscosity plays no role at all.

We chose the 80-90% regime of glycerol concentrations to highlight the non-linear effect of viscosity and the linear effect of refractive index on FA. Even FA of EGFP has a linear and noticeable increase at lower glycerol concentrations, and we have included the data (revised Figure 1E). We chose to show the 80-90% regime to highlight the glaring insensitivity of FA_EGFP to the exponentially increasing viscosity. We don’t want to state that the solution microviscosity has no effect on FA of EGFP, but rather, has a significantly less effect on EGFP as compared to fluorescein. In room temperature water, EGFP has a rotational correlation time (*θC*) of ~16 ns while fluorescein’s is ~4 ns, while EGFP’s fluorescence lifetime (*τ*) is ~2.6 ns while fluorescein’s is ~4 ns. Thus, for τθc for EGFP is <1 while for fluorescein it is ~1. Viscosity and *θC* increases exponentially with glycerol content, so according to the Perrin equation r=r0/(1+τ/θc), the effect of viscosity is more visible on the FA of fluorescein than EGFP (revised Figure 1E).

2. The authors could have performed a direct comparison of FA with another method in vivo, to plot in particular the change in reGFP as a function of intracellular translational diffusion of eGFP. This data could help the reader better understand the in vivo link between FA and classical metrics such as diffusion coefficients.

We have used single particle tracking of 200 nm beads and FRAP of EGFP to investigate MMC both in-vitro and in-vivo. FRAP of EGFP is insufficient in measuring MMC changes in multiple cases while single particle tracking succeeds. However, both methods have less throughput than rEGFP, and thus strengthens our method. In revised Figure 4C and 5D-E, we have presented a side-by-side comparison of FA and EGFP diffusion rate for different MMC conditions.

3. The authors show that homoFRET would occur at eGFP concentration of 10 μM or higher, and estimated that in isotonic conditions, the eGFP concentration in NIH/3T3 cells is of ~ 8 μM. However, when the cells are osmotically compressed, cell volume decreases by up to 40%, raising the concentration above 10 μM. The authors do not discuss this point which is crucial and could limit the use of their methods in this particular case.

We appreciate the reviewer for this keen observation which we indeed overlooked. To understand the effect of [EGFP]-dependent homo-FRET on our measurements in cases of cell volume compression, we fitted the rEGFP vs [EGFP] curve in Figure 1F to a power law function: y = a + b∙xc (Author response image 4), which gives an estimate of the expected rEGFP at different [EGFP] in crowder-less environments. Next, we used the [EGFP] distribution in revised Figure 2 —figure supplement 1D-inset to compute the cellular [EGFP] increment upon volume compression (Author response image 4), assuming that an x% decrease in volume increases the concentration by x%. Indeed, at a volume compression of 40% (final cell volume = 60% of the initial volume), the EGFP concentration in all cells exceed 10 μM, thus entering the homo-FRET regime. Then, we mapped the individual cellular EGFP concentrations at different levels of volume compression to the expected rEGFP values obtained from the fit (Author response image 4). The rEGFP values do not account for the presence of crowders in Author response image 4 and are based on in-vitro measurements, which is usually higher than intracellular rEGFP values. We find that the lowest value of rEGFP at 40% volume compression equates to ~0.204, which is ~0.015% less than the median value of 0.2074. Therefore, we do not think that homo-FRET has a considerable impact on the cell population we investigated. However, we have added a line of caution in the Discussion section of the revised manuscript (line 488) regarding the measurement of cellular MMC under severe hypertonic stresses. Additionally, we would like to point out that the cells we chose for quantifying MMC passed through an intensity cutoff, which generally eliminated the cells that highly overexpressed EGFP.

**Author response image 4. sa2fig4:** (**A**) Fitting rEGFP vs [EGFP] to y = a + b xc and its residual. (**B**) Estimated cellular [EGFP] distribution at different cell volume compressions. (**C**) Estimated rEGFP at different cell volume compressions in crowder-less condition and using cell-free system rEGFP values.

4. The link between the cellular response to regulated cell volume increase (RVI) is not well established. In particular, it seems that RVI is not the only route to osmoadaptation in the authors' data: one can clearly observe, e.g. in Figure S4D, in particular for HeLa cells, that, whatever reGFP really measures, its value is almost back to isotonic condition while cell volume is not. Similarly, treatment with Flufemanic acid shows that all cannot be attributed to RVI. Thus, the authors should not consider RVI as the only osmo-adapting mechanism, as their data do not show this.

We agree with the reviewer that post-hypertonicity induction, the recovery of rEGFP and cell volume does not correlate in HeLa cells. Given the differences between the RVI of NIH/3T3 and HeLa, there could be a different underlying mechanism present in HeLa that decouples the rEGFP and cell volume during RVI. Furthermore, we are also not aware of any other osmoadaptation mechanism than RVI that can occur in less than an hour, so we can only speculate that cytosolic osmolyte accumulation rescues the MMC in HeLa cells to compensate the lack of volume recovery. Another possible reason could be protein condensation due to hypertonic stress, which elevates rEGFP beyond the expected value as condensates have a higher refractive index than the cytoplasm (https://doi.org/10.1101/2020.10.25.352823).

5. The link of TNFR1 with osmoadaptation is not properly demonstrated. In Figure 6C, RVI is partly abrogated under the inhibition of TNFR1. But what I found odd is the fact that volume decrease is also decreased, for both HeLa and NIH/3T3 cells, which is confirmed by FA increasing less than in control conditions. This is not discussed and is probably associated with other effects of the KD of TNFR1 or Zafirlukast, for instance on cell volume or crowding, or on cytoskeleton tension (as Figure S6Dii shows, even if this data is not discussed in the main text!). As the authors do not present the values of reGFP and cell volume under KD of TNFR1 but only their relative change, it is hard to understand what happens. This makes the interpretation of the authors on the putative role of TNFR1 not convincing.

We apologize for not explaining Figure S6D in the unrevised manuscript and not presenting intrinsic the cell volume of Zafirlukast treated cells and TNFR1-KD cells. The volumes of TNFR1 incapacitated cells are smaller, and they also show less volume change post hypertonicity induction. We have included the data in the revised text (Figure 6C). Inhibiting TNFR1-signalling resulted in three different phenomena: [1] a reduction in cell volume (Figure 6C) and elevation of MMC (Figure 6 —figure supplement 1C), [2] a reduced rate of hypertonicity induced cell shrinkage (Figure 7 —figure supplement 1A), and [3] a loss of cellular RVI (Figure 7A) and MMC recovery ability (Figure 7B). We suspect that these three phenomena are not interconnected since cellular RVI requires NFkB activation which is absent under TNFR1 inactivation. Using the drug CAY10512 to inhibit NFkB activation, we observe a similar loss of RVI (Figure 7A), although the hypertonic cell shrinkage is analogous to untreated conditions (Figure 7 —figure supplement 1A). Hypertonic cell shrinkage has been shown to be linked to membrane aquaporin levels, and inhibiting aquaporin activity hinders hypertonic volume shrinkage (https://doi.org/10.1074/jbc.M008760200, https://doi.org/10.1128%2FJB.01665-12). Hence, we are exploring the role of aquaporins in volume shrinkage and RVI as future studies.

We claim the indispensability of TNFR1-NFkB signaling in RVI because- [1] cells show RVI during moderate hypertonicity (150 mM) but not severe hypertonicity (600 mM), [2] moderate hypertonicity induces nuclear p65 translocation, but not severe hypertonicity, and [3] downregulating TNFR1 activity via Zafirlukast or knockdown, or NFkB inhibition by CAY10512 prevents hypertonicity-induced p65 nuclear translocation and prevents RVI. Thus, our data show that TNFR1 is necessary for initiating cellular RVI. RIPK1 recruitment and ubiquitination is a critical factor that promotes p65 nuclear translocation and determines the cell fate upon TNFR1 activation (Figure 3 in https://doi.org/10.1038/s41580-023-00623-w). We show that the increased cytoplasmic microviscosity at severe hypertonicities impairs RIPK1 recruitment, thus delaying p65 nuclear translocation, and consequently, impairing RVI.

Recommendations for the authorsBelow is a list of recommendations associated with the five aforementioned weaknesses of the manuscript, followed by other points and recommendations on other parts.1. I suggest the authors redo the in vitro experiments with lower values of glycerol, and possibly another viscogen. It could help the reader better understand how FA scales with crowding. The use of cell extracts could also be interesting to better establish the method in vitro.

We have further showed the rEGFP of α-lactalbumin in the revised Figure 1A and showed rEGFP data for lower glycerol content in revised Figure 1E.

2. The link with in vivo is hard to make, as crowding is much more complex. However, the authors should provide data comparing in vivo FA to the measurement of eGFP translational diffusion coefficients, for instance using FRAP.

We have provided FRAP data multiple cases, like revised Figure 4C and 5D-E, and in revised Figure 3B, we show and explain why FRAP of EGFP and similar proteins is not a reliable indicator of protein crowding.

3. The authors should measure the concentration of eGFP under various osmotic conditions to ensure that part of the FA changes measured in vitro cannot be attributed to other effects such as homoFRET, as their data suggest. This could change their interpretation of the results plotted in figure 5 on the deviation from the concentration-dilution regime (see specific point 2 below).

Measuring intracellular [EGFP] accurately using FCS may not yield a reliable value of [EGFP], as certain parameters like diffusion rate of EGFP, confocal volume, or EGFP brightness which are essential for accurate concentration estimations, will have non-trivial changes due to the elevated MMC. However, as addressed in Author response image 4, the issue of homo-FRET due to hypertonic cell volume shrinkage is negligible.

4. I suggest the authors put less emphasis on RVI as the main osmoadapting mechanism in their experiments. Some other routes can exist and should be discussed. This is fine if crowding is not fully correlated to RVI, as the data of the authors seem to show.

We agree that crowding may not be fully correlated to RVI, as our data in revised Figure 8 suggests, however, we can only speculate that alternative adaptive mechanisms like osmolyte accumulation enables MMC recovery. We have mentioned the same in the Discussion section of the revised manuscript in line 584.

5. More experiments and discussion of the data are needed to reach the conclusions the authors reached on TNFR1. The authors should show the effect of the KD alone on different biophysical parameters (like cell volume and reGFP, as they have for tension), and discuss them. I was, for instance, extremely surprised to see experiments of Figure S6Dii, which seems of primordial importance to understand and interpret the effect, not at all discussed in the main text. In general, and this is true also for CAY10512 treatment, the authors should not only display the change of the mode of the data but the whole distribution. How is it impacted? Does it, in particular, become bimodal, with a proportion of cells responding to osmotic stresses while others do not, partly explaining what the authors show? It is a shame to have access to very rich single-cell data and only use mean or median, while distributions can be extremely informative.

We apologize for not discussing the data in the main text. In the revised manuscript, we have plotted the hypertonic shrinkage of single cells with kernel smoothed distribution curves (Figure 7 —figure supplement 1A). We believe our sample size is too small to comment on the modality of distributions. We did not show individual cell volume traces as the graphs are already too cluttered and so we did not want to increase the complexity.

Additionally, even if these points are not weaknesses of the manuscript, these are further questions the authors should address:1. Figure 2B: I understand the data, but I think that the conclusion is too quickly reached. Cells of different aspect ratios can have different spreading and thus different heterogeneity or crowding. Before using the median as a metric and disregarding potential differences at different aspect ratios, I would appreciate having another measurement of crowding (for instance, by FRAP) of translation diffusion of eGFP in cells or different aspect ratios.

In the revised Figure 3B, we explain why FRAP is not a reliable indicator of crowding. While we agree that cells of different morphologies have different crowding levels (revised Figure 4B), FRAP is insufficient to explain the differences and single particle tracking yields too localized data for creating microviscosity maps of individual cells. In revised Figure 2 —figure supplement 1B, our representative FA measurements with a 10x objective provide a large enough sample size for comparing cells of different aspect ratios and spread area. The use of the rEGFP distribution’s modal value bypasses the heterogeneities in the individual protoplasm and provides a more accurate metric to compare cell-to-cell variability (revised Figure 2 —figure supplement 1A).

2. We discussed homoFRET in the above points 3. Regarding Figure 5 and the discussion about the deviation from the concentration-dilution curve, why the authors do calibrate this curve using the two extreme values of FA (0mM and 600mM)? Clearly, in the case of homoFRET, the 600mM value could not be used. Maybe if the curve were calibrated with other lower values, you would only find a deviation from the concentration-dilution curve at high osmotic stresses, which could then either be a sign of non-linearity of FA dependency in vivo with crowding or some other processes. The authors should clarify this point before reaching their conclusions about a deviation from the concentration-dilution regime.

We understand the implication- if the hypertonicity-induced cell volume compression increases homo-FRET in the cells, the rEGFP values would decrease with cell shrinkage and the nature of the rEGFP vs cell volume curve would deviate from the concentration-dilution law, as visible now. We have provided an explanation as to why homo-FRET would not significantly affect the rEGFP vs cell volume curve in Author response image 4. Furthermore, the shift is again evident in revised Figure 8B where we have measured rEGFP and cell volume in the same cells. Comparing the time points of 5- and 15-minutes post hypertonicity induction, the rEGFP values are similar but there is a difference of ~500 μm3 in the cell volumes. We ascribe this behavior to the formation of EECVs (EGFP Excluding Cell Volumes), and subsequently justify EECVs in the following panels. The difference in cell volumes indicates that the formation of EECVs can increase rEGFP for a particular cell volume than what is expected, and perhaps the actual MMC condition. Regarding the endpoints for the concentration-dilution curve, we chose the isotonic condition and the maximally compressed condition of 600 mM hypertonicity to cover the maximum range of the dataset. However, we realize that choosing any of the intermediate points of the rEGFP vs cell volume curve is erroneous, so we fitted the normalized cell volume as a function of the inverse of applied hypertonicity to a Boyle-van’t Hoff line, which yielded the osmotically inactive volume at infinite hypertonicity. We computed the concentration-dilution curve using the new coordinates, as explained in revised Figure 8A.